# Climate and hydraulic traits interact to set thresholds for liana viability

Alyssa M. Willson [1], Anna T. Trugman [2], Jennifer S. Powers [3,4], Chris M. Smith-Martin[5] & David Medvigy [1✉]

Lianas, or woody vines, and trees dominate the canopy of tropical forests and comprise the majority of tropical aboveground carbon storage. These growth forms respond differently to contemporary variation in climate and resource availability, but their responses to future climate change are poorly understood because there are very few predictive ecosystem models representing lianas. We compile a database of liana functional traits (846 species) and use it to parameterize a mechanistic model of liana-tree competition. The substantial difference between liana and tree hydraulic conductivity represents a critical source of inter-growth form variation. Here, we show that lianas are many times more sensitive to drying atmospheric conditions than trees as a result of this trait difference. Further, we use our competition model and projections of tropical hydroclimate based on Representative Concentration Pathway 4.5 to show that lianas are more susceptible to reaching a hydraulic threshold for viability by 2100.

[1] Department of Biological Sciences, University of Notre Dame, 100 Galvin Life Sciences, Notre Dame, IN 46556, USA. [2] Department of Geography, University of California Santa Barbara, Santa Barbara, CA 93106, USA. [3] Department of Ecology, Evolution, and Behavior, University of Minnesota, St. Paul, MN 55108, USA. [4] Department of Plant and Microbial Ecology, University of Minnesota, St. Paul, MN 55108, USA. [5] Department of Ecology, Evolution and Evolutionary Biology, Columbia University, New York, NY 10027, USA. ✉email: dmedvigy@nd.edu

Lianas are the main competitors with trees for light in tropical forests, influencing both the magnitude of carbon (C) storage through replacing larger tree stems with smaller liana stems[1] and C residence time through faster liana woody tissue turnover[2–4]. Because lianas are structural parasites, relying on trees for mechanical support, lianas can afford to construct more leaf area per unit supporting stem area than trees[5,6]. This distinction makes lianas formidable competitors for limited light at the forest canopy and reduces ecosystem C storage via decreased allocation to longer-lived woody stem tissue.

In tropical biomes, dry, moist, and wet forests occur in contrasting precipitation regimes, leading to markedly different plant communities[7], with lianas being more abundant in dry forests[8]. Under current and future climate change, increased temperatures are predicted to intensify water stress, particularly in regions already experiencing periodic dry conditions[9–11]. One metric of atmospheric dryness that increases plant water stress, vapor pressure deficit (VPD), is calculated from air temperature and humidity. VPD describes atmospheric water demand and is strongly negatively correlated with global gross primary production (GPP)[12,13]. Despite this negative correlation, the impact of VPD on growth form-specific abundance has not been established in the tropics[13].

Liana abundance is increasing in tropical forests of the Americas[14–16], with consequences for tropical forest ecosystem function and diversity. An increase in liana abundance can decrease tropical forest carbon storage[17], decrease the commercial value of forests[18,19], and increase the cost of resource extraction[20,21]. Additionally, increasing liana abundance can increase tree mortality[14], decrease tree fruit production[18,22], and alter Neotropical tree community composition via differential tolerance of tree species to liana parasitism[23]. Therefore, understanding the mechanisms underlying liana proliferation, particularly in the contexts of liana-tree canopy competition and climate change, is crucial to improving ecological forecasts and implementing appropriate management practices[24]. Such efforts will aid in maintaining tropical forest diversity, terrestrial C sink strength, and economic sustainability of forest resource extraction.

Here, we capitalize on the increase in liana research over the past two decades[25,26] to compile a pantropical database of liana functional traits. Using this database, we parameterize a liana-tree competition model and use the model to discern how the

identified trait differences influence liana and tree viability under different climate scenarios. We find that sapwood-specific hydraulic conductivity ($K_{s,max}$), a plant hydraulic trait describing the maximum amount of water passing through the xylem and strongly related to C sequestration via leaf-level gas exchange, is significantly higher on average among lianas than trees. We then show that this trait largely determines liana viability in model simulations. Under future climate conditions, our results indicate that the viable range of liana hydraulic conductivities will become smaller than the range observed today.

## Results and discussion

**Functional trait meta-analysis.** To identify systematic differences in functional traits between lianas and trees, we compiled a pantropical database of functional traits from the TRY plant trait database[27] (Methods: TRY meta-analysis). We selected traits to (1) include multiple plant organs (i.e. leaves, stems, roots, and hydraulic architecture), (2) represent tradeoffs in allocation and life history strategy (e.g. high specific leaf area (measuring leaf efficiency) is often correlated with low leaf lifespan[28]), and (3) correspond with standard functional traits in global vegetation models[29,30]. According to our database, containing 846 liana species and over 12,000 tree species, the most striking differences between trees and lianas existed in hydraulic traits (Supplementary Fig. 1).

We used the conclusion from our preliminary analysis of the TRY database, that hydraulic traits systematically differ between tropical trees and lianas, as the foundation for a more comprehensive analysis of hydraulic functional traits between tropical trees and lianas (hereafter "extended meta-analysis;" Methods: Extended meta-analysis). In our extended meta-analysis, on average, liana $K_{s,max}$ was over three times greater than tree $K_{s,max}$ (Glass' $\Delta = 2.69$, Mann–Whitney test statistic = 1452, $n_{tree} = 103$, $n_{liana} = 51$, $p < 1.0 \times 10^{-5}$; Fig. 1, Supplementary Tables 1 and 2). Meanwhile, the pressure at which 50% xylem function is lost (representing hydraulic safety, $P_{50}$) and the slope of the percent loss of conductivity curve (representing the sensitivity of the xylem to changing pressure, Slope) were not statistically or physiologically different ($P_{50}$: tree mean 18% greater than liana mean, Glass' $\Delta = 0.35$, Mann–Whitney test statistic = 984, $n_{tree} = 60$, $n_{liana} = 40$, $p > 0.12$; Slope: liana mean

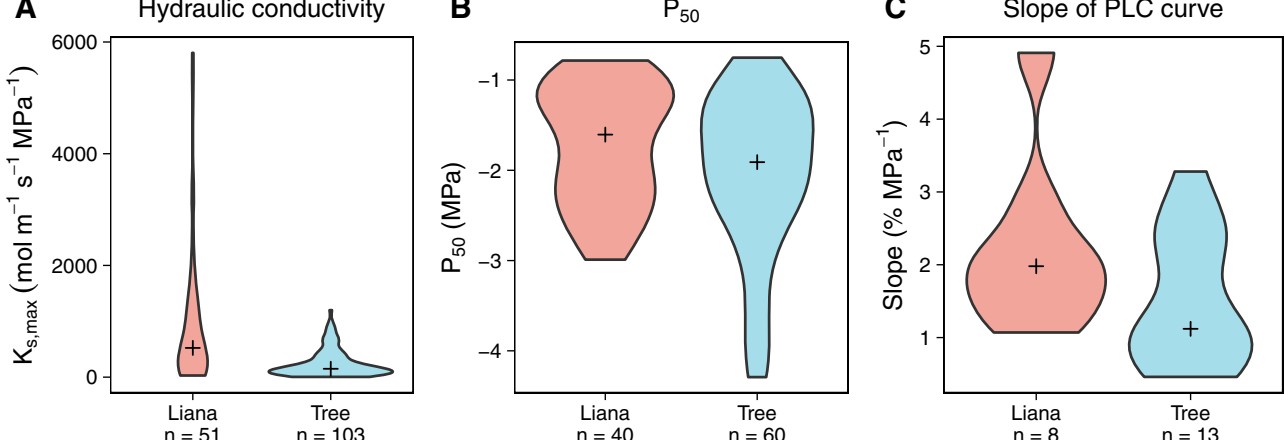

**Fig. 1 Hydraulic trait differences between growth forms. A** Lianas have substantially greater stem-specific hydraulic conductivity ($K_{s,max}$) and **B** marginally less negative pressure at which 50% xylem function is lost ($P_{50}$). **C** Slope of the percent loss of conductivity (PLC) curve does not differ between growth forms. Results derived from our extended meta-analysis, which combines observations from the TRY database with more recent hydraulic trait measurements. Red violins represent lianas, blue violins represent trees. Black crosses represent medians for each growth form. Number of species for which each trait was measured is indicated below the growth form name.

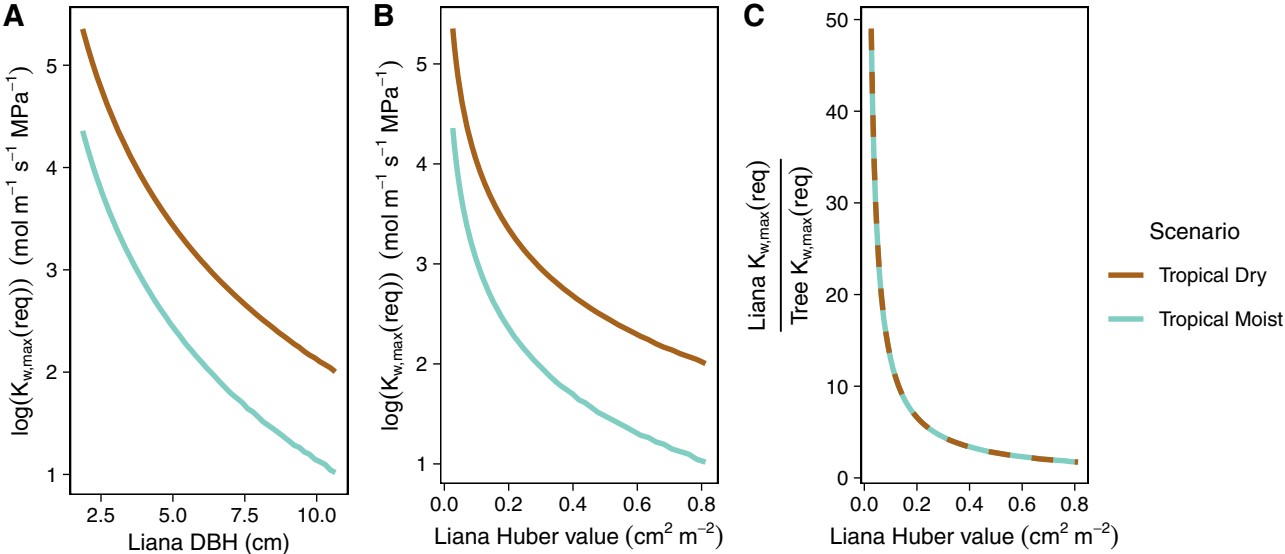

**Fig. 2 Allometry and climate affect required maximum whole-plant hydraulic conductivity.** Required maximum whole-plant hydraulic conductivity ($K_{w,max}$(req)) as a function of diameter at breast height (DBH, **A**) and Huber value (sapwood area [$cm^2$] per unit leaf area [$m^2$]), **B**, **C** and hydroclimate (tropical moist forest or tropical dry forest). Total leaf area = 200 $m^2$, 60% tree leaf area, 40% liana leaf area. In all three panels, colors represent the different hydroclimate scenarios (tropical dry forest = brown; tropical moist forest = blue). **A** Liana log($K_{w,max}$(req)) as a function of liana DBH. **B** Liana log($K_{w,max}$(req)) as a function of liana Huber value. **C** The ratio of liana $K_{w,max}$(req) to tree $K_{w,max}$(req) as a function of liana Huber value. Tree $K_{w,max}$(req) was computed at a reference scenario where tree DBH = 18.2 cm.

50% greater than tree mean, Glass' Δ = 0.78, Mann–Whitney test statistic = 33, $n_{tree}$ = 13, $n_{liana}$ = 8, $p > 0.1$, Fig. 1, Supplementary Tables 1 and 2). These conclusions persist regardless of which growth form is used as the reference group in the calculation of Glass' Δ. For $K_{s,max}$, Glass' Δ is smaller in magnitude when the liana growth form is used as the reference group, reflecting the higher variance within the liana growth form than the tree growth form, but lianas still show significantly higher $K_{s,max}$ on average than trees (Glass' Δ using liana growth form as reference = −0.55). Both $P_{50}$ and Slope remain non-significant when using lianas as the reference group.

By contrast, we found only weak differences in leaf and stem anatomy traits and no differences in root traits (Supplementary Discussion: TRY meta-analysis & Extended meta-analysis). Tropical ecologists have regarded lianas as having more acquisitive traits (i.e., traits yielding a quick return on resource investment, e.g., high photosynthesis rate) than trees[31–35]. Our results suggest that lianas are not systematically more acquisitive than trees across plant organs. However, the relatively few observations of root traits for both tree and liana growth forms precludes a definitive conclusion. Root trait measurements should be a priority moving forward to accurately characterize the differences between trees and lianas.

While regional studies have previously identified the difference in hydraulic traits between trees and lianas[36–38], our results are unique in three ways. First, we find that differences in hydraulic traits are not accompanied by differences in root traits. Second, our analysis represents the most comprehensive pantropical meta-analysis of liana hydraulic traits, demonstrating the pervasiveness of differences in hydraulic traits between growth forms. Third, our results were performed on a database of liana hydraulic traits that was compiled with an explicit consideration of the unique liana xylem anatomy, making the estimates of liana $K_{s,max}$ and the difference between growth forms in $K_{s,max}$ more reliable.

The large difference in $K_{s,max}$ between groups suggests that $K_{s,max}$ represents a substantial source of variation between growth forms; therefore, we sought to identify thresholds of liana and tree

viability, defined as the minimum conditions under which annual net primary production (NPP) is greater than zero, under different hydroclimate scenarios.

**Hydraulic traits influence viability.** To evaluate how $K_{s,max}$ influences liana-tree competition, we parameterized a plant model[39] coupling Farquhar photosynthesis[40], Shinozaki water transport[41], and Ball-Berry stomatal conductance[42] to estimate annual net primary production (NPP) for a liana-tree pair sharing a single canopy (Methods: Competition Model). We restricted growth form-specific parameterization to whole-plant hydraulic conductivity, allometry, and woody turnover rate (Methods: Parameterization). We conducted an extensive sensitivity analysis to ensure that parameters for which tropical data are sparse would not strongly influence simulation outcomes (Methods: Sensitivity analysis).

We forced the model with average monthly soil water potential (Ψ) and average hourly vapor pressure deficit (VPD) characteristic of Central American sites representing contrasting hydroclimates: Barro Colorado Island, Panama ("tropical moist forest") and Horizontes, Costa Rica ("tropical dry forest") (Methods: Climate Data). All other parameters remained constant between runs. For each scenario, we identified the minimum maximum whole-plant hydraulic conductivity required ($K_{w,max}$(req), Supplementary Fig. 2) to maintain annual NPP > 0 (Methods: Simulations).

We find that liana $K_{w,max}$(req) is greater at lower diameters when total leaf area is constant and at lower Huber value (Fig. 2a, b) because the xylem supplies relatively more leaves with water under these conditions. This pattern indicates that the unique liana allometry influences its physiology, consistent with the structure of our model (Methods: Competition Model) and the theoretical model derived by Mencuccini et al.[43]; specifically, a lower Huber value, characteristic of lianas in comparison to trees[3,5], demands higher $K_{w,max}$(req) to supply leaves with a consistent source of water, thus maintaining positive NPP.

Second, liana $K_{w,max}$(req) is greater than tree $K_{w,max}$(req) except at large liana Huber values, at which point the liana's

sapwood-to-leaf area allometry approaches the tree's allometry. This result is consistent with our meta-analysis (Fig. 1) and with previous site-specific comparisons of liana and tree $K_{s,max}$[35,38,44]. The consistency of our model predictions, based on physical properties of xylem function, with observation suggests that the observed difference in $K_{s,max}$ in the literature represents a fundamental source of variation between woody growth forms in tropical forest biomes. This variation must be represented in the development of a liana growth form in vegetation models.

Finally, we find that climatic water stress influences $K_{w,max}$(req) (Fig. 2). The approximately twofold increase in liana $K_{w,max}$(req) in the dry forest compared with the moist forest (Fig. 2) suggests that liana $K_{w,max}$(req) is sensitive to changes in hydroclimate. Moreover, the ratio of liana $K_{w,max}$(req) to tree $K_{w,max}$(req) does not change as a function of hydroclimate (Fig. 2c), indicating that tree $K_{w,max}$(req) is similarly sensitive to hydroclimate. Therefore, we next investigated the magnitude of change in liana and tree $K_{w,max}$(req) over a hydroclimate gradient representative of tropical dry and tropical moist Neotropical forests. Furthermore, $K_{w,max}$(req) could be sensitive to low water supply (low $\Psi$), high water demand (high VPD), or a combination of the two hydroclimate variables. Because $\Psi$ and VPD naturally covary, we used our model to separate the sensitivity of liana and tree $K_{w,max}$(req) to $\Psi$ and VPD.

**Hydraulic trait-climate interactions.** Because of the natural covariance between $\Psi$ and VPD and the limited locations with reliable estimates of liana $K_{s,max}$, partitioning tropical forest vegetation sensitivity to the supply and demand of water has been a challenge[45]. Our approach was to address this challenge through model simulations. We interpolated annual $\Psi$ and VPD data ($\Psi$-VPD indices) between our tropical moist and tropical dry forest sites and used our model to find $K_{w,max}$(req) for each $\Psi$-VPD index for each growth form (Methods: Simulations). $K_{w,max}$(req) is more sensitive to increasing VPD than to decreasing $\Psi$, regardless of growth form; in fact, $K_{w,max}$(req) is sensitive to $\Psi$ only at the highest VPD indices (Fig. 3). This result suggests that neither trees nor lianas are limited by soil water supply under most conditions characteristic of American tropical forests; therefore, our simulations do not support the hypothesis that lianas experience a dry season growth advantage due to access to deep soil water reserves[8,25]. Rather, in agreement with recent field and common garden studies[46,47], our results imply that the maintenance of high $K_{s,max}$ relies more on lianas' ability to minimize water loss during the dry season (i.e. high water use efficiency).

Across $\Psi$-VPD indices, liana $K_{w,max}$(req) and tree $K_{w,max}$(req) display strikingly different sensitivities. Assuming a total leaf area of 200 m$^2$ and the "established" scenario, liana $K_{w,max}$(req) is on average ~24 times greater than tree $K_{w,max}$(req). Liana $K_{w,max}$(req) varied from 39 to 104 mol m$^{-1}$ s$^{-1}$ MPa$^{-1}$ under the wettest and driest hydroclimate scenarios, respectively. By contrast, tree $K_{w,max}$(req) only changed by 3 mol m$^{-1}$ s$^{-1}$ MPa$^{-1}$ over the same range of hydroclimate scenarios (Fig. 3). The greater absolute difference in liana and tree $K_{w,max}$(req) under drier hydroclimate is consistent with a recent empirical comparison of functional traits between growth forms in dry and wet tropical forests[48]. These results remain consistent under alternative competition and total leaf area scenarios (Supplementary Discussion: Competition model).

Because $K_{w,max}$(req) defines the hydraulic requirement for viability and VPD is predicted to increase in the future[13], our model suggests that lianas may reach a hydraulic physiological limit for viability sooner than trees, despite currently having a "dry season advantage" over trees[25]. To demonstrate this point,

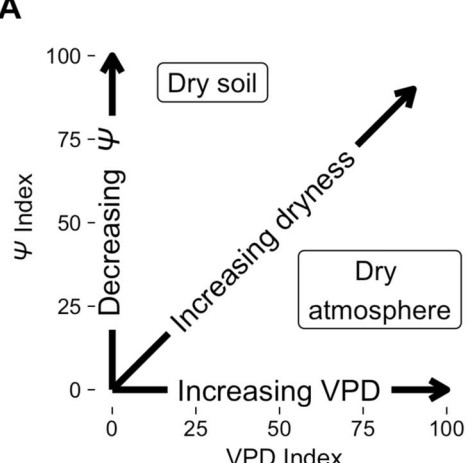

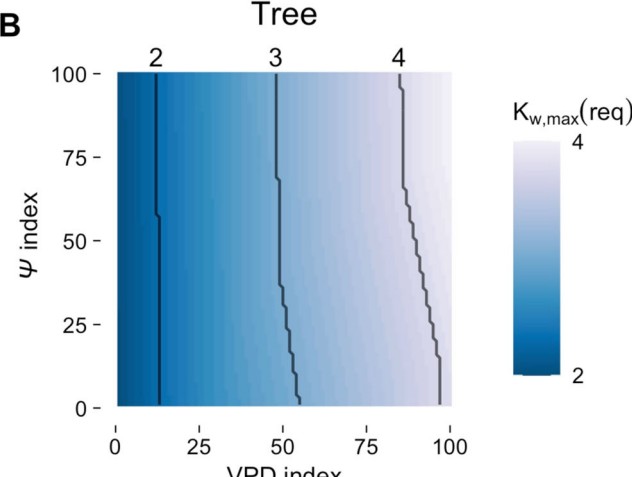

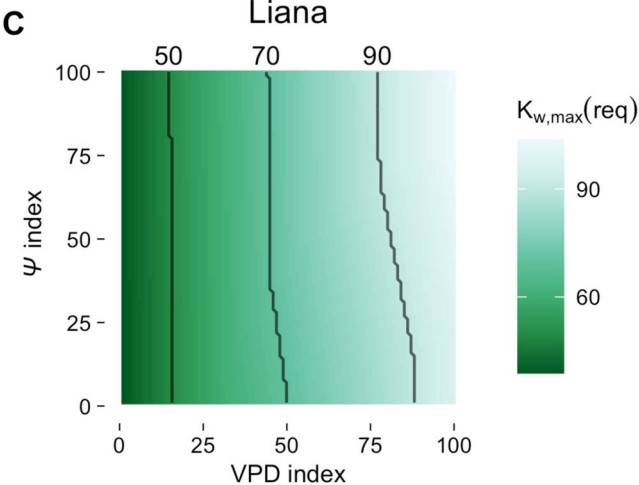

we extended our computation of $K_{w,max}$(req) to a future scenario in which VPD is double the present-day VPD at Horizontes, our tropical dry forest. This scenario is well within the range of those predicted by Coupled Model Intercomparison Project 5 (CMIP5) models under Representative Concentration Pathway (RCP) 4.5 for 2100 in the tropics[12]. Overall, the pattern of increasing

**Fig. 3 Required maximum whole-plant hydraulic conductivity ($K_{w,max}$(req)) as a function of vapor pressure deficit (VPD) and soil water potential ($\Psi$). A** Conceptual diagram showing how hydroclimate changes over the 2-dimensional space depicted in the other two panels. **B, C** $K_{w,max}$(req) (mol m$^{-1}$ s$^{-1}$ MPa$^{-1}$) over 10,000 combinations of VPD and $\Psi$ indices. Color (blue = tree, green = liana) represents $K_{w,max}$(req), with lighter color indicating greater $K_{w,max}$(req). Black lines are contours, which indicate the dominant axis of variation: vertical lines suggest $K_{w,max}$(req) is more sensitive to VPD and horizontal lines suggest $K_{w,max}$(req) is more sensitive to $\Psi$. All simulations were computed under the scenario of an established liana (40% of 200 m$^2$ total leaf area). **B** Tree $K_{w,max}$(req). **C** Liana $K_{w,max}$(req). Note different scales.

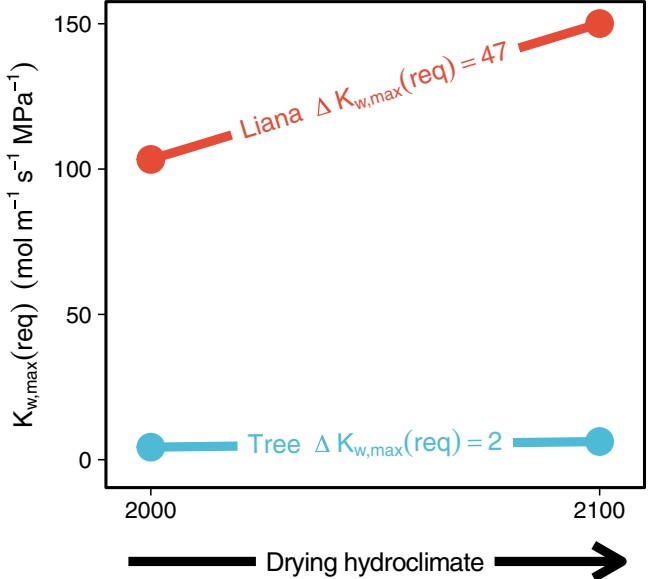

**Fig. 4 Liana required maximum whole plant hydraulic conductivity ($K_{w,max}$(req)) is more sensitive to drying hydroclimate than tree $K_{w,max}$(req).** Increase in liana and tree $K_{w,max}$(req) under present (2000) and future (2100) climate conditions at the tropical dry forest site (Horizontes, Costa Rica). $K_{w,max}$(req) is computed under the established liana scenario (60% tree leaf area, 40% liana leaf area of 200 m$^2$ total leaf area). Blue: tree $K_{w,max}$(req), red: liana $K_{w,max}$(req). Lines and labels depict the change in $K_{w,max}$(req) from present to 2100 for each growth form with units of mol m$^{-1}$ s$^{-1}$ MPa$^{-1}$.

$K_{w,max}$(req) continues for both trees and lianas, with liana $K_{w,max}$(req) increasing faster than tree $K_{w,max}$(req) as VPD increases, despite simultaneous increases in atmospheric carbon dioxide concentration (Supplementary Fig. 3). The increase in $K_{w,max}$(req) persists under different total leaf area and competition scenarios, as well as under an assumption of adapting (i.e., decreasing) $P_{50}$ (Supplementary Discussion: Competition model). However, the magnitude of the difference in $K_{w,max}$(req) between the present and 2100 is greater for lianas than trees (tree $\Delta K_{w,max}$(req) = 2 mol m$^{-1}$ s$^{-1}$ MPa$^{-1}$, liana $\Delta K_{w,max}$(req) = 47 mol m$^{-1}$ s$^{-1}$ MPa$^{-1}$; Fig. 4). Experimental and observational research has already attributed tree mortality to the effects of severe droughts and drying hydroclimate worsened by climate change and similar mortality events are expected in the future[49]. The greater sensitivity of liana $K_{w,max}$(req) than tree $K_{w,max}$(req) to drying hydroclimate in our simulations implies that lianas may undergo similar mortality events as $K_{w,max}$(req) becomes greater than maximum whole-plant conductivity, reinforcing our

prediction that a threshold for liana viability may be reached under 21$^{st}$ century climate change (Fig. 4, Supplementary Fig. 4).

**Adaptation to future hydroclimate.** In the future, liana and tree communities may physiologically adapt to drying hydroclimate by increasing cavitation resistance (i.e., lower average liana and tree $P_{50}$). Due to the simplicity of our model and the strong and uncertain correlation between $P_{50}$ and the slope of the percent loss of conductivity curve (Supplementary Fig. 5), we did not consider the possibility of $P_{50}$ adaptation in our future climate scenario simulations in order to vary as few parameters as possible. However, it is possible that greater cavitation resistance could result in lower $K_{w,max}$(req) via the hypothesized trade-off between xylem efficiency and safety[44].

To address the possibility that $K_{w,max}$(req) may be lower among lianas and trees in the future if $P_{50}$ adaptation occurs, we conducted additional simulations of liana and tree $K_{w,max}$(req) with lower $P_{50}$ parameterizations, corresponding to higher cavitation resistance (Methods: Model simulations). Our results indicate that $P_{50}$ adaptation has the potential to lower $K_{w,max}$(req) for both lianas and trees (Fig. 5). As $P_{50}$ decreases, $K_{w,max}$(req) decreases under both the drier and wetter site scenarios for the year 2100. Under the wetter hydroclimate scenario, when $P_{50} = -2.25$ MPa, tree $K_{w,max}$(req) = 1.84 mol m$^{-1}$ s$^{-1}$ MPa$^{-1}$ and liana $K_{w,max}$(req) = 42.2 mol m$^{-1}$ s$^{-1}$ MPa$^{-1}$ while when $P_{50} = -3$ MPa, tree $K_{w,max}$(req) = 1.30 mol m$^{-1}$ s$^{-1}$ MPa$^{-1}$ and liana $K_{w,max}$(req) = 29.3 mol m$^{-1}$ s$^{-1}$ MPa$^{-1}$ (compared to tree $K_{w,max}$(req) = 2.22 mol m$^{-1}$ s$^{-1}$ MPa$^{-1}$ and liana $K_{w,max}$(req) = 52.1 mol m$^{-1}$ s$^{-1}$ MPa$^{-1}$ with no $P_{50}$ adaptation). Under the drier hydroclimate scenario, when $P_{50} = -2.25$ MPa, tree $K_{w,max}$(req) = 5.09 mol m$^{-1}$s$^{-1}$ MPa$^{-1}$ and liana $K_{w,max}$(req) = 121 mol m$^{-1}$ s$^{-1}$ MPa$^{-1}$ and when $P_{50} = -3$ MPa, tree $K_{w,max}$(req) = 3.54 mol m$^{-1}$ s$^{-1}$ MPa$^{-1}$ and liana $K_{w,max}$(req) = 83.8 mol m$^{-1}$ s$^{-1}$ MPa$^{-1}$ (compared to tree $K_{w,max}$(req) = 6.25 mol m$^{-1}$ s$^{-1}$ MPa$^{-1}$ and liana $K_{w,max}$(req) = 150 mol m$^{-1}$ s$^{-1}$ MPa$^{-1}$ with no $P_{50}$ adaptation). This represents a significant decrease in $K_{w,max}$(req), particularly for lianas. However, $K_{w,max}$(req) remains greater for 2100 than at present for all scenarios even under the most extreme $P_{50}$ adaptation scenario we considered (present-day liana $K_{w,max}$(req) = 25.6 mol m$^{-1}$ s$^{-1}$ MPa$^{-1}$ and 71.3 mol m$^{-1}$ s$^{-1}$ MPa$^{-1}$ under wetter and drier hydroclimate scenarios, respectively and present-day tree $K_{w,max}$(req) = 1.14 mol m$^{-1}$ s$^{-1}$ MPa$^{-1}$ and 3.00 mol m$^{-1}$ s$^{-1}$ MPa$^{-1}$ under wetter and drier hydroclimate scenarios, respectively). This suggests that drying hydroclimate in the future is likely to impose a greater physiological demand on plant hydraulic architecture, particularly for lianas, regardless of the ability of the plant to experience $P_{50}$ adaptation.

Our model assumes that the hydraulic efficiency-safety trade-off occurs similarly in both trees and lianas, as evidenced by the decrease in $K_{w,max}$(req) under all scenarios when $P_{50}$ decreases. This hypothesis has received considerable empirical support for the tree growth form[35,50,51], but evidence for a trade-off among lianas is unsubstantiated. For example, van der Sande et al.[35]. found no trade-off between $K_{s,max}$ and $P_{50}$ for lianas. This suggests that liana $K_{w,max}$(req) may not benefit from decreasing $P_{50}$ under drier hydroclimate conditions. To more fully understand how hydroclimate and $P_{50}$ influence hydraulic physiological limits among lianas, future work should continue to investigate the relationship between hydraulic traits in lianas and more realistic models of liana hydraulic architecture should be developed for inclusion in dynamic vegetation models.

Furthermore, thus far, we have focused on the scenario of a threshold-like response of lianas to drying hydroclimate; that is, when $K_{w,max}$(req) surpasses realized maximum whole-plant

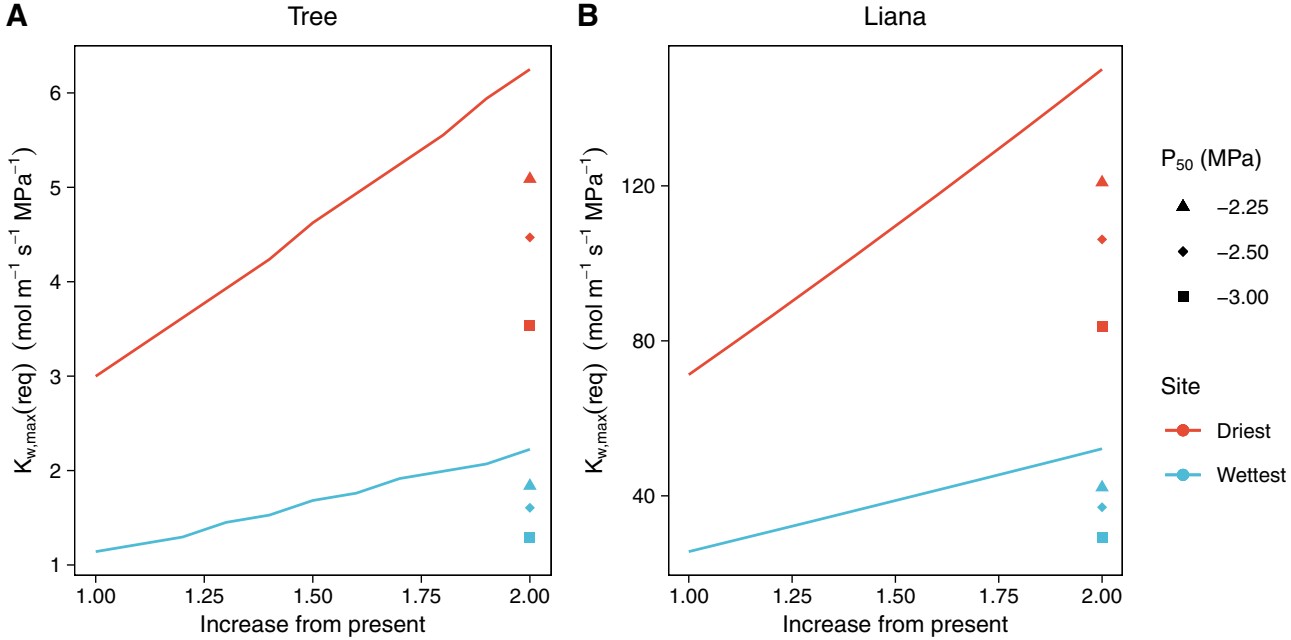

**Fig. 5 Effect of $P_{50}$ parameter value on projections of future required maximum whole-plant hydraulic conductivity ($K_{w,max}$(req)).** Change in $K_{w,max}$(req) as vapor pressure deficit (VPD) increases according to future projections for Central America. The *x*-axis is a multiplier of increase from the present. For example, 2.00 means VPD is doubled from the current hourly values for each month. The lines represent $K_{w,max}$(req) under potential future VPD conditions spanning 1x to 2x current VPD at the dry forest site, Horizontes, Costa Rica (red) and the moist forest site, Barro Colorado Island, Panama (BCI, blue). **A** tree $K_{w,max}$(req), **B** liana $K_{w,max}$(req). Symbols at 2.00 on the *x*-axis of each panel represent $K_{w,max}$(req) under various conditions of $P_{50}$ adaptation when VPD is doubled from present. Triangle: tree $P_{50}$ = liana $P_{50}$ = −2.25 MPa; diamond: tree $P_{50}$ = liana $P_{50}$ = −2.5 MPa; square: tree $P_{50}$ = liana $P_{50}$ = −3.0 MPa.

hydraulic conductivity, lianas will be unable to maintain a positive annual carbon balance, leading to higher mortality rates. More gradual mechanisms may also lead to increased liana mortality under a drier hydroclimate. For instance, physiological adaptations leading to a greater Huber value among lianas may decrease their competitive advantage with trees, thus leading to a more gradual decline in liana viability via greater competition with trees[52]. Such physiological adaptations could include a reduction in total leaf area to reduce water loss via transpiration or an increase in allocation to woody tissues to increase water storage. Alternatively, drought deciduousness among lianas could become more prevalent under drier conditions[8]. All of these adaptations would allow lianas to maintain a similar $K_{w,max}$(req) to that realized today, but would reduce net photosynthesis[52]. Nevertheless, our conclusions indicate that lianas are more susceptible than trees to drying hydroclimate and may experience higher mortality, whether via a threshold-like effect of increased $K_{w,max}$(req) or via a decrease in net photosynthesis in response to physiological adaptation to greater $K_{w,max}$(req).

In this study, we identified hydraulic conductivity as a critical trait that distinguishes lianas from trees, with lianas on average having a greater $K_{w,max}$(req). The difference in $K_{w,max}$(req) between lianas and trees is sensitive to Huber value and to VPD. Moving forward, the difference in liana and tree traits, particularly hydraulic traits, should be incorporated in more dynamic vegetation models[53,54]. The very few previous attempts to do so highlighted uncertainties in liana trait parameterization[54,55]. Our database of liana traits should significantly ameliorate this concern. Although important uncertainties remain with respect to liana belowground traits, belowground uncertainty pertains to trees and other plant functional types besides lianas.

We suggest that a climate threshold exists over which lianas will be unable to survive given the sensitivity of their hydraulic architecture to hydroclimate. If atmospheric VPD increases as projected by climate models, recent increases in liana abundance in the Americas[14–16] may be short-lived, with long-term consequences for forest community dynamics[56], C storage capacity[1,2,57], and the economic value of tropical forests[20,21,58]. Even if a climate threshold for liana viability is not realized, lianas may sustain significant reductions in population size via increased competition-driven mortality. In order to improve forecasts of these processes under climate change, dynamic vegetation models should include lianas parameterized with their distinguishing hydraulic traits.

## Methods

**TRY meta-analysis**. We used the TRY plant trait database[27] to identify traits that show systematic differences between the tree and liana growth forms, as a way to narrow the scope of the rest of the analysis. We chose traits to represent major tradeoffs within the "economic spectrum" framework, which places plants along a spectrum of strategies from acquisitive, fast return on investment to conservative, slow return on investment according to key functional trait values[30]. We narrowed traits to those that had observations for at least four tree and liana species. We then compiled our dataset using the following steps during November and December 2019. For each trait, we downloaded the dataset for all species available globally and averaged the observations of the trait to the species level to avoid statistical biases introduced in our growth form comparison due to a high density of observations in a few commercially valuable species. We matched the species ID number with the most frequently used growth form identifier using the TRY "growth form" trait and kept the species with the most frequent identifier of "tree," "liana," or "woody vine." We subsetted the data to keep only species with a majority of observations ascribed to the tree and liana growth forms (i.e., no herbaceous species, ferns, etc.), resulting in observations for 44,222 total species. Finally, we filtered the dataset of 44,222 species by hand to remove species misclassified as trees or lianas; species occurring entirely in temperate to boreal biomes; species from all gymnosperm lineages except the order *Gnetales*; and entries for taxonomic classifications broader than the genus level (e.g., taxonomic families). We found that hydraulic functional traits in the TRY database (i.e., $K_{s,max}$ and $P_{50}$) show systematic differences between growth forms (Supplementary Fig. 1; Supplementary Tables 3 and 4), while there is mixed evidence for differences in the acquisitiveness of trees and lianas in terms of stem anatomical traits (Supplementary Fig. 1; Supplementary Tables 3 and 4) and

leaf functional traits (Supplementary Fig. 6; Supplementary Tables 3 and 4), and no evidence of differences between tropical trees and lianas with respect to root functional traits (Supplementary Fig. 7; Supplementary Tables 3 and 4).

**Extended meta-analysis**. We conducted an additional literature search to supplement the hydraulic trait observations from the TRY database. The additional literature search served two purposes: (1) to fill a major gap identified during our TRY analysis in terms of liana trait observations, and (2) to address the methodological inconsistency of measuring $K_{s,max}$ and $P_{50}$ on liana branches shorter than the longest vessel, which incorrectly measures $K_{s,max}$ and $P_{50}$ without accounting for end wall resistivity[59,60].

We conducted a literature search using Web of Science and Google Scholar. We searched the following phrases in combination with "liana:" "hydraulic conductivity," "hydraulic trait," "hydraulic efficiency," and "hydraulic K." Of the literature we found, we kept only the studies that met the following criteria: (1) reported $K_{s,max}$ measurements for lianas, (2) measured $K_{s,max}$ instead of computing $K_{s,max}$ from xylem conduit dimensions, (3) measured $K_{s,max}$ on sunlit, terminal branches of mature individuals or saplings, and (4) measured $K_{s,max}$ on a branch longer than the longest vessel. We considered the authors to have used a branch length longer than maximum vessel length if the authors measured or reported maximum vessel length for the species and a longer branch was used. Because the best methodological practice for measuring $P_{50}$, especially in species with long vessels, is currently a matter of debate, we additionally removed all observations of $P_{50} > 0.75$. This filtering was performed to reduce the probability that falsely high (i.e., less negative) $P_{50}$ values were retained in our analysis because of improper measurement technique and is consistent with the $P_{50}$ filtering performed by Trugman et al.[61]. Improper measurement technique is a particular concern for lianas, whose wide and long vessels require cautious implementation of the traditional measurement techniques developed for trees. We note that retaining all liana $P_{50}$ observations (i.e., not filtering out observations > −0.75) results in a significant difference between trees and lianas (Mann–Whitney test statistic = 1029, $n_{tree} = 61$, $n_{liana} = 46$, $p < 0.05$). However, the effect size remains relatively small, indicating that even when retaining unrealistically high liana $P_{50}$ values, the difference between liana and tree $P_{50}$ is ecologically of only moderate significance (Glass' $\Delta = 0.47$). When possible, we manually inspected vulnerability curves of each species and removed strongly r-shaped curves, but corresponding hydraulic safety margins were not available for a quantitative determination. We applied the same criteria to the observations in the TRY database, combined the observations from TRY and from our additional literature search, and averaged the observations to the species level. This resulted in a total of 154 species with hydraulic trait observations matching our criteria, of which 51 species were lianas and 103 species were trees.

A list of the sources of our measurements is available in Supplementary Table 5[35,62–72].

**Statistical analysis**. For both the TRY analysis and the extended metaanalysis, we compared the tree and liana growth forms using two methods. First, we used two-sided Mann–Whitney $U$-tests, which test whether observations between groups are drawn from the same distribution. We used Mann–Whitney $U$-tests rather than t-tests because the distributions of most traits violate the normality assumption of t-tests. This approach is consistent with a recent pantropical meta-analysis comparing liana and tree functional trait distributions[73].

Second, we computed Glass' $\Delta$, a measure of effect size, which describes the magnitude of the difference between groups compared with the variation within the reference group[74,75]. The Glass' $\Delta$ was chosen rather than Cohen's d because the standard deviation of each group is substantially different for several traits, including hydraulic traits[74,75]. To avoid biasing our interpretation of the statistics by considering only one growth form as the reference group, we computed and present the test statistic and 95% confidence intervals resulting from using both the tree growth form (subscript "T") and liana growth form (subscript "L") as the reference group (Supplementary Table 2; Supplementary Table 4). Throughout the text, we present the statistics computed using the tree as the reference group for two reasons. First, we were interested in the degree to which lianas differ from the well-parameterized tree plant functional types in dynamic vegetation models. Second, because lianas are often parameterized using data from early successional tropical trees[55], we were interested in considering the degree to which the distribution of liana trait values differs from the distribution of tree trait values characterizing the plant functional types in which lianas are traditionally categorized.

All statistical analyses were conducted in the R statistical environment[76]. Mann–Whitney $U$-tests were conducted using the "stats" package and Glass' $\Delta$ statistics were computed using the "effectsize" package[77].

**Competition model**. We modified the single-tree model originally developed by Trugman et al.[39] to represent a single liana-tree pairing. The purpose of the original model developed by Trugman et al. is to calculate annual net primary production ($A_{net}$) of a single temperate tree under defined climatic conditions and morphological and physiological parameters, with $A_{net}$ becoming the input to a subsequent model describing tree drought recovery. Briefly, the model couples water transport using the Shinozaki pipe model[41] and the Ball-Berry model of stomatal conductance[42] and whole-plant photosynthesis using the Farquhar photosynthesis model[40]. The amount of water moving through the plant depends on soil water availability (soil water potential, $\Psi$); the hydraulic path length and xylem area of fine roots, stem, and petioles; and the water demand imposed on the tree by the atmosphere (vapor pressure deficit, VPD). Mathematically, this can be written with the following set of equations. First, the flow, $F$ (mmol s$^{-1}$), throughout a plant element is computed by integrating the hydraulic conductivity per unit of xylem area ($K$) from one end of the pipe continuum with water potential $\psi_1$ (MPa) to the other with water potential $\psi_2$, which can be expressed by the differences in the Kirchhoff transforms as

$$ F = \frac{a}{L} \int_{\psi_1}^{\psi_2} K(\psi) d\psi = \frac{a}{L}(\phi_2 - \phi_1) \tag{1} $$

where $a$ (m$^2$) is the xylem area of the element and $L$ (m) the pipe length. The element conductivity ($K$, mmol m$^{-1}$ s$^{-1}$ MPa$^{-1}$) decreases as stem water potential falls as a result of embolism. A logistic function of the form

$$ \frac{K_{max} * \exp(b1 * (\psi_{soil} - b2))}{\exp(b1 * (\psi_{soil} - b2)) + 1} \tag{2} $$

where b1 is the slope of the percent loss of conductivity (PLC) curve and b2 is $P_{50}$, is used to represent the loss of conductivity as water potential becomes more negative, and thus $\phi$ (mmol m$^{-1}$ s$^{-1}$) is a function of the maximum whole-plant hydraulic conductivity, $K_{max}$ (mmol m$^{-1}$ s$^{-1}$ MPa$^{-1}$). The assumptions of our pipe model (i.e., constant xylem area, $a$, with branching and path length, $L$, that is representative of the whole path from roots to leaves) allows us to approximate an individual tree or liana with an effective element conductivity for the entire path. This is in contrast to stem-specific hydraulic conductivity ($K_{s,max}$, mmol m$^{-1}$ s$^{-1}$ MPa$^{-1}$), which is commonly measured in the field on terminal branches and does not account for the tapering of vessel elements in branches. Therefore, $K_{max}$ is distinct from $K_{s,max}$.

If we neglect changes in water storage, $F$ is constant throughout the hydraulic continuum. Then, water flow from the roots to the stem is modeled as

$$ F = \frac{a_{root}}{L_{root}}(\phi_{soil} - \phi_{root}) = \frac{a_{stem}}{L_{stem}}(\phi_{root} - \phi_{stem}) \tag{3} $$

where $a_{root}$ and $a_{stem}$ are the cross-sectional xylem area of the root system and the cross-sectional area of the xylem, respectively, $L_{root}$ and $L_{stem}$ are the path length from the soil to the base of the stem and the tree height, respectively, and ($\phi_{soil} - \phi_{root}$) and ($\phi_{root} - \phi_{stem}$) are the integral of conductivity from the soil to the roots and from the roots to the stem, respectively, calculated from the Kirchhoff transform.

Flow from the stem to leaves is modeled as

$$ \frac{a_{stem}}{L_{stem}}(\phi_{root} - \phi_{stem}) = \frac{a_{petiole}}{L_{petiole}}\left(\phi_{stem} - \int_0^L \phi_{leaf}(l_a)\frac{dl_a}{L_a}\right) \tag{4} $$

where $a_{petiole}$ is the cross-sectional xylem area within a given petiole summed over the tree, $L_{petiole}$ is the length of the petiole, ($\phi_{stem} - \int_0^L \phi_{leaf}(l_a)\frac{dl_a}{L_a}$) is the integral of the conductivity from the stem to the petiole, $L_a$ (m$^2$ m$^{-2}$) is the leaf area index, $l_a$ (m$^2$) is the index of a given leaf layer, and $dl_a/L_a$ represents the xylem area per unit leaf. Assuming there is only one leaf layer and all photosynthesis is carbon limited only, this equation simplifies to

$$ \frac{a_{stem}}{L_{stem}}(\phi_{root} - \phi_{stem}) = \frac{a_{petiole}}{L_{petiole}}(\phi_{stem} - \phi_{leaf}) \tag{5} $$

where ($\phi_{stem} - \phi_{leaf}$) is the integral of the conductivity from the stem to the petiole under the assumption of one leaf layer. Flow from the leaf to the atmosphere is modeled as

$$ \frac{a_{petiole}}{L_{petiole}}(\phi_{stem} - \phi_{leaf}) = a_{leaf}g_s D \tag{6} $$

where $a_{leaf}$ is leaf area, $g_s$ (mmol m$^{-2}$ s$^{-1}$) is stomatal conductance, and $D$ (Pa) is VPD. Stomatal conductance, $g_s$, is modeled following ref. [67]. as

$$ g_s = A_n \frac{c_1}{(C_a - \Gamma)(1 + \frac{D}{D_0})}\beta(\psi_{leaf}) \tag{7} $$

In this equation, $C_a$ (ppm) is the atmospheric $CO_2$ concentration; $c_1$ (Pa), $D_0$ (Pa), and $\Gamma$ (ppm) are empirical constants from the Leuning model[78]; $A_n$ (kg C month$^{-1}$) is net photosynthesis; and $\psi_{leaf}$ is leaf water potential. The function $\beta(\psi_{leaf})$ serves to down-regulate photosynthesis under water stressed conditions and is determined by the carbon cost of sustaining negative water potential and loss of conductivity in the xylem. For simplicity, we assumed that $\beta(\psi_{leaf})$ varies linearly with the Kirchhoff transform as

$$ \beta(\psi_{leaf}) = \frac{\phi_{leaf}}{\phi_{max}} \tag{8} $$

where $\phi_{max}$ is the integral of maximum hydraulic conductivity of the xylem. $\beta(\psi_{leaf})$ varies between 1 (leaf at full hydration) and 0 (leaf under full water stress). The denominator $\phi_{max}$ is defined in terms of the maximum hydraulic conductivity

($K_{max}$) as follows:

$$\phi_{max} = \frac{K_{max} * \log(\exp(-b1 * b2) + 1)}{b1} \qquad (9)$$

where $K_{max}$ is the model equivalent of the maximum whole-plant hydraulic conductivity ($K_{w,max}$) and b1 (% MPa$^{-1}$) and b2 (MPa) are the slope of the percent loss of the conductivity curve and the pressure at which 50% of xylem function is lost, respectively. Here, $\beta$ broadly conforms to the solution to the Leuning model, but with a more mechanistic representation of soil moisture stress through soil water potential's effect on leaf water potential.

The method of solution is the same as in Trugman et al.[39]. In this way, computation of $A_{net}$ is related to three climatic variables ($\Psi$, VPD, and $CO_2$ concentration), dimensions of the water conducting tissue of the tree, and tree physiological parameters.

We modified the Trugman et al. model to include a tree-liana pair and to improve the realism of the relationship between climate and plant water flow. In contrast to the use of this model for computing $A_{net}$ as in Trugman et al., we use the model to define $K_{w,max}$(req), the required maximum whole-plant hydraulic conductivity, by iteratively finding the minimum $K_{max}$ (Eq. 5) to yield a positive $A_{net}$ on an annual timestep (Methods: Simulations). To emphasize the independence of the maximum hydraulic conductivity in the model ($K_{max}$) from plant branch-level measurements and differentiate this term in the model from $K_{s,max}$ (observed branch hydraulic conductivity), we designate this term maximum whole-plant hydraulic conductivity ($K_{w,max}$) hereafter. The hydraulic conductivity variables we consider in this manuscript ($K_{s,max}$, $K_{w,max}$, and $K_{w,max}$(req)) are defined in Supplementary Table 6.

We modified the model to account for the liana growth form in three ways: inclusion of liana-tree competition, development of liana-specific allometry, and development of a turnover routine. Our model assumes the liana and the tree are in direct competition for light and soil water. The liana-tree pair was assigned a total leaf area of 200 m$^2$ and we varied the proportion of the total leaf area given to each the tree and the liana (Methods: Simulations). Tree and liana leaves are distributed homogeneously through the canopy and the model assumes all leaves are sun leaves. Light competition is only dependent on the quantity of leaves apportioned to each growth form; the placement of the leaves is not considered. The growth forms compete for soil water by extending a fine root area proportional to leaf area into a single, homogeneous soil layer. There is assumed to be no parasitic effect of the liana on the tree.

Liana stem length does not depend on diameter at breast height (DBH), consistent with previous modeling efforts[55]. Instead, we assume liana length is as long as tree height, therefore making their canopies of equal height[55]. Liana stem length may be substantially longer than tree height[47]; our estimates of $K_{w,max}$(req) should be interpreted as conservative estimates. Liana DBH is then treated one of two ways. In Fig. 2, we investigate the simultaneous effects of allometry (i.e., Huber value) and hydroclimate on $K_{w,max}$(req). In this figure, we defined the total leaf area shared by the tree and the liana (200 m$^2$) and allowed liana DBH to vary between the minimum and maximum liana DBH (1.86 and 10.7 cm, respectively) observed during a field survey in Guanacaste, Costa Rica. We then computed Huber value by dividing the sapwood area (a function of DBH) by the total leaf area apportioned to each growth form. In all other model simulations, we assigned liana DBH according to the competition scenario: 2.65 cm for the "established" scenario (equal to the mean of the observations from Guanacaste, Costa Rica) and 2.00 cm for the "invasion" scenario (the minimum stem diameter for a canopy liana; see "Model parameterization").

We developed a turnover routine to account for differences in leaf and stem turnover between trees and lianas. The routine works as follows: during a given month, a small amount of stem is lost from an initial stem length at the beginning of the year (model parameter Lx), which corresponds with one-twelfth of the average annual stem turnover of the tree or liana. If net primary production (NPP) is negative for the month, all leaves are dropped (leaf area = 0) for the growth form and net primary production (NPP) is recalculated with leaf respiration = 0. If NPP is still negative after leaves are dropped, then stem turnover is increased to simulate a water stress response, which reduces stem respiration. This routine serves two purposes. First, the leaf turnover component allows us to account for the possibility of different phenological strategies between growth forms[79,80]. To the extent possible, we allow lianas to retain leaves during the dry season to account for the potential of a "dry season growth advantage," during which lianas maintain photosynthesis under drier conditions than trees[25]. Second, the stem turnover component represents the fact that lianas are documented to have more rapid woody turnover than trees[4].

The second part of our model modification is the downscaling of the model to an hourly step. The original model took as inputs VPD and $\Psi$ at a monthly timestep. However, this does not account for the strong subdaily variation in VPD. Therefore, we modified the hydroclimate drivers of the model to account for hourly variability in VPD: $\Psi$ remained a vector of monthly averages, while VPD became a matrix of hourly x monthly values. For use in the model, a moving average of VPD with the previous hour's VPD was calculated to smooth the effect of increasing VPD during the day and to account for our specification of 6:00-18:00 as daylight throughout the year.

We downscaled by computing respiration and gross primary production (GPP) for each hour of the day. GPP was set to 0 during the night (18:00-6:00) to produce

a 12-h light-dark cycle. We summed hourly respiration and GPP to produce daily and monthly values. Then, respiration and GPP entered the turnover routine. Finally, net primary production (NPP) was computed as NPP = GPP - respiration.

**Model parameterization**. The only model inputs that differed between the tree and liana growth forms were maximum whole-plant stem-specific hydraulic conductivity ($K_{w,max}$ mmol m$^{-1}$ s$^{-1}$ MPa$^{-1}$), DBH (cm), leaf area (m$^2$), turnover (% year$^{-1}$), and initial stem length (m) (Supplementary Table 7). We chose to keep $P_{50}$ and the slope of the percent loss of conductivity (PLC) curve (model parameters b2 and b1, respectively, Supplementary Table 7) the same between growth forms because (1) our meta-analysis suggested that the difference between growth forms in these traits is minimal compared to $K_{s,max}$ (Fig. 1), and (2) this decision minimized the number of parameters contributing to differences in required $K_{w,max}$ ($K_{w,max}$(req)) between growth forms.

We tested for correlations among the three traits within our meta-analysis. We found only weak correlations between $K_{s,max}$ and $P_{50}$ and between $K_{s,max}$ and slope of the PLC curve (both: $R^2 \approx 0.1$, $p < 0.05$, Supplementary Fig. 5), suggesting that fixed values for $P_{50}$ and slope of the PLC curve are appropriate for our analysis. Meanwhile, the correlation between $P_{50}$ and slope of the PLC curve is strong ($R^2 \approx 0.7$, $p < 0.05$, Supplementary Fig. 5), reinforcing the fact that assigning values for these parameters with a fixed relationship best represents plant physiology. We therefore pooled observations of slope of the PLC curve and $P_{50}$ from both growth forms in our meta-analysis to compute our estimates of b1 and b2.

DBH distributions and average DBH for each growth form were taken from surveys of lianas and trees in a second growth forest of Guanacaste, Costa Rica (Supplementary Fig. 8). For the scenario of an established liana in a tree canopy ("established scenario"), we assumed a liana DBH equal to the mean observed at Guanacaste, $\approx 2.65$ cm. For the scenario of a liana invading a tree canopy ("invasion scenario" considered in the Supplementary Discussion), we assumed a liana DBH of 2 cm[81]. In all simulations, tree DBH was assumed to be the average from the tree survey of Guanacaste ($\approx 18$ cm). For the established scenario, we assumed the liana occupied 40% of the total leaf area (80 m$^2$) and the tree occupied 60% of the total leaf area (120 m$^2$). For the invasion scenario, we assumed the liana occupied 10% of the 200 m$^2$ total leaf area (20 m$^2$) and the tree occupied 90% of the total leaf area (180 m$^2$).

For most traits, there was limited evidence for tree-liana differences (e.g., wood density, Glass' $\Delta < 1$) or there were insufficient data to parameterize the liana growth form (e.g., root:shoot ratio). Specific leaf area (SLA) was a special case. Although SLA was found to be significantly different between growth forms (Glass' $\Delta \approx 1$), we did not assign different values to lianas and trees because the TRY results are likely influenced by the low SLA of desert-dwelling and montane shrubs within the tree growth form. Values of the inputs and parameters that differ from the original model are provided in Supplementary Table 7[3,82–86]. All other parameters are the same as those used in the original model[39].

**Sensitivity analysis**. We conducted an extensive sensitivity analysis of our model to identify the parameters that are most influential to determining $K_{w,max}$(req). For each parameter in the model ($n = 25$), we computed $K_{w,max}$(req) with a 50% reduction and 50% increase from the default (mean) value while holding all other parameters at their default values. We then found the difference between $K_{w,max}$(req) from the 50% increase and 50% decrease in parameter value and divided the difference by the $K_{w,max}$(req) at the default parameter value; we report this computation as the "sensitivity." We computed the sensitivity of each parameter for two hydroclimate conditions, BCI and Horizontes, and for the two competition scenarios, established and invasion (with respect to the liana). When tree $K_{w,max}$(req) was computed, we held all liana parameters at their default values and vice versa. This amounted to over 400,000 additional annual model simulations. This sensitivity analysis informed the parameters that we used field collected data to constrain, including diameter at breast height (DBH), $P_{50}$, and hydraulic path length (Lx). Where constraining the parameters with field data was not possible, we conducted additional model simulations with alternative scenarios. For example, given that we found the model to be sensitive to the total leaf area, we ran additional simulations to create Fig. 3 and 4 under alternative total leaf areas, 150 m$^2$ and 400 m$^2$. The Supplementary Methods (Supplementary Method: Sensitivity analysis) offers more detailed results of our sensitivity analysis and how those results informed our modeling procedure.

**Climate data**. Our model computes NPP as a function of carbon dioxide concentration ([$CO_2$]), $\Psi$, and VPD. We set [$CO_2$] at 400 ppm, a low-end estimate for the 21st century, for all model simulations except in our predictions of 2100, in which [$CO_2$] = 550 ppm[87]. Our hydroclimate data come from two Neotropical forests with contrasting hydroclimate conditions, Horizontes, Costa Rica and Barro Colorado Island (BCI), Panama. $\Psi$ was determined from Medvigy et al.[88] (Horizontes) and Levy-Varon et al.[89] (BCI). In each case, $\Psi$ was estimated for multiple soil layers in the original dataset. However, because measurements were not taken at the same soil depths at each location and because our model assumes there is only one soil layer, we used $\Psi$ estimates from only the 15 cm depth, which was available for both sites, for all simulations. VPD data are from a reanalysis data product for Horizontes averaged over 2007-2018 (ref. [90]). For BCI, data are from

the Lutz Tower from 27 May 2002 to 5 June 2020 at 48 m canopy height. We computed VPD from relative humidity and air temperature data at both sites as follows:

$$SVP = \frac{610.78 * \exp\left(\frac{AT}{AT+238.3} * 17.2694\right)}{100} \quad (10)$$

$$VPD = \left(SVP * \left(1 - \frac{RH}{100}\right)\right) * 100 \quad (11)$$

where SVP is saturation vapor pressure (hPa), AT is air temperature (°C), RH is relative humidity (%), and VPD is in Pa. At both sites, VPD data were averaged across year and day of the month. Changes in monthly $\Psi$ and VPD for BCI and Horizontes are available in Supplementary Fig. 9.

**Simulations**. We used our model to simulate required conductivity ($K_{w,max}$(req)) by identifying the smallest value of whole-plant conductivity ($K_{w,max}$) at which NPP is positive under the given hydroclimate and liana-tree competition conditions. To compute $K_{w,max}$(req) we performed the following steps: (1) define the simulation inputs, including DBH and total leaf area fraction for each growth form, and hydroclimate (i.e., VPD and $\Psi$); (2) run the model for each month with the smallest value of $K_{w,max}$ available; (3) sum the monthly NPP computed by the model; (4) if NPP > 0, define $K_{w,max}$(req) as the current $K_{w,max}$; and (5) if NPP ≤ 0, select the next lowest value of $K_{w,max}$ and repeat the steps until NPP > 0, at which point $K_{w,max}$(req) is identified (Supplementary Fig. 2).

We emphasize that the model depends on $K_{w,max}$, whereas it is much more common to measure terminal branch $K_{s,max}$. Because of the uncertainty associated with scaling between $K_{w,max}$ and $K_{s,max}$, our estimates of $K_{w,max}$(req) should be compared to observed $K_{s,max}$ with caution. To reduce uncertainty in this parameter, we urge further measurements of $K_{w,max}$.

We first simulated $K_{w,max}$(req) under different hydroclimate scenarios, as shown in Fig. 2. The hydroclimate scenarios are tropical dry forest and tropical moist forest (Methods: Climate data). Instead of defining the liana DBH, we computed $K_{w,max}$(req) over a range of DBH values observed in our liana survey dataset from Horizontes, which allowed us to avoid assigning a fixed allometry to lianas in our initial simulations.

We similarly computed $K_{w,max}$(req) under a variety of VPD-$\Psi$ scenarios. The indices were computed by linearly interpolating the hydroclimate between the driest (Horizontes) and the wettest (BCI) sites for a length of 100 indices for both VPD and $\Psi$. For each combination of $\Psi$ and VPD (10,000 combinations), we computed $K_{w,max}$(req) using the method outlined above.

We extended our computation of $K_{w,max}$(req) for each growth form into the future under a gradient of increasing VPD conditions. Because of the high uncertainty surrounding the magnitude of increases in VPD over the next 100 years[12], we computed $K_{w,max}$(req) under a variety of VPD scenarios, ranging from 10% to 100% increase in VPD from the present at Horizontes (Supplementary Fig. 3). For the model simulations involving future VPD scenarios, we additionally changed the atmospheric [$CO_2$] to 550 ppm to reflect the dependence of climate change (i.e., increasing VPD) on increasing atmospheric [$CO_2$].

Finally, we investigated the potential influence of liana and tree physiological adaptation to drying hydroclimate via adapting $P_{50}$. Because of the strong empirical correlation between $P_{50}$ and the slope of the percent loss of conductivity curve (Slope), we simultaneously varied these two parameters in three additional scenarios, with hydroclimate conditions predicted for 2100 (i.e., 100% increase in VPD, no change in $\Psi$, [$CO_2$] = 550 ppm). We used the "established" competition scenario and assumed the same adaptation scenarios for both liana and tree $K_{w,max}$(req) simulations. The three scenarios are as follows: b1 = 0.92% MPa$^{-1}$, b2 = −2.25 MPa; b1 = 0.73% MPa$^{-1}$, b2 = −2.5 MPa; and b1 = 0.49% MPa$^{-1}$, b2 = −3 MPa. The most extreme liana $P_{50}$ observed in the literature we included in our extended meta-analysis is -2.99 MPa; therefore, our $P_{50}$ adaptation scenarios are consistent with the most drought-tolerant observations of present-day liana $P_{50}$.

**Reporting summary**. Further information on research design is available in the Nature Research Reporting Summary linked to this article.

## Data availability
The raw TRY data, processed TRY functional trait dataset, and our extended hydraulic functional trait meta-analysis have been deposited in the figshare repository at https://doi.org/10.6084/m9.figshare.c.5990986.v1. The values for parameters needed to run the model and the climate drives for the model are available on Github at https://github.com/amwillson/liana-tree-comp[91].

## Code availability
The model code and code to create each of the figures is available on GitHub (https://github.com/amwillson/liana-tree-comp)[91].

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

## Acknowledgements

We thank Helena Kleiner, Megan Vahsen, Jason McLachlan, Jody Peters, and Haley Kodak for suggestions on an early drafts of the manuscript. We are grateful for meteorological data obtained from Copernicus Climate Change Service information 2019. This material is based upon work supported by the U.S. Department of Energy, Office of Science, Office Biological and Environmental Research, under Award Numbers DESC0014363 and DESC0020344. A.M.W. received support from an NSF Graduate Research Fellowship, an Arthur J. Schmitt Fellowship, University of Notre Dame Environmental Research Center Graduate Fellowship and NSF Grant 1241874. A.T.T. acknowledges funding from the NSF Grants 2003205 and 2017949, the USDA National Institute of Food and Agriculture, Agricultural and Food Research Initiative Competitive Programme Grant No. 2018-67012-31496 and the University of California Laboratory Fees Research Program Award No. LFR-20-652467.

## Author contributions

D.M. provided the initial idea and methods for the project. A.M.W. and D.M. further developed the idea and methods. J.S.P. and C.M.S.-M. provided data. A.T.T. developed the original model. A.M.W. modified the model, ran the model simulations, conducted the analysis, and wrote the first draft of the manuscript with input from D.M. D.M., J.S.P., C.M.S.-M. and A.T.T. contributed to manuscript revisions.

## Competing interests

The authors declare no competing interests.
