## [Peer Review File · Nature Communications]

Reviewer comments, first round

Reviewer #1 (Remarks to the Author):

This study by Wilson et al. identifies important research and measurement needs of liana traits and anatomy and employs a mechanistic model for competition to evaluate liana viability in future climate conditions. The authors supplement liana measurements available in the TRY database with other data sources listed in the Extended Data Table 3 and thus identify where measurements are lacking, especially belowground. They also adapt a tree carbon allocation model used in Trugman et al. 2018 to conclude that liana viability might be threatened under future hydroclimate condition in the tropics.

This is an important study with many qualities. The meta-analysis of the extended dataset was clear, described differences between growth forms clearly, and accurately identified sample size limitations. I appreciate the authors' sharing of the data collection process and analysis on github for reproducibility. The science in this manuscript is very thoughtful; the authors have provided deep explanations of the different assertions made throughout. I often found my questions answered after further reading, especially in the "Supplementary Methods" and the "Supplementary Results and Discussion."

My main concern for this manuscript is the obscurity of what the model does, what are the inputs? What does it simulate? What processes does it consider? Throughout, the authors refer to agreements between model output and empirical observation to support model accuracy (e.g., lines 121-123 and lines 153-155). While the manuscript does refer to the Trugman et al (2018) paper for the model, I feel it important to at least give a brief recap of the model. Then, the supplementary section titled "competition model" will be more easily understood.

A consequence of the above is that it was hard for me to understand how why liana Kreq was so sensitive to some of the parameters mentioned in Supplementary figures 10 and 11. The authors do a great job of showing how the data constrained the model parameters to which Kreq was most sensitive. I suggest adding to this by expanding 1) upon model description and 2) the discussion on the relevant mechanisms making liana Kreq so sensitive to hydroclimatic changes (is the Huber value enough to explain the difference? Lines 127-134). These would help the reader further trust the modeling process.

Another thing to clarify: does the model consider possible liana P50 adaptation? Would this reduce future Kreq and change the conclusions of the study? Is a sensitivity analysis of Kreq around a decrease in P50 (more negative) possible?

Minor comments:

Figure 2: Why does the tree Kreq change when tree canopy cover fraction changes? I don't think this is obvious from the text.

Lines 351-353: Why was this specific threshold chosen? Is it possible it might cause the result in Fig. 1 that the P50s of trees and lianas are similar to be an artifact of data selection?

Extended Data Figure 1: Abscissa should be Kw not Ks

Reviewer #2 (Remarks to the Author):

Review of "Climate and hydraulic traits interact to set thresholds for liana viability"
Nature Communications

15-June-2021

Summary

Willson et al. use a meta analysis of liana functional traits to parameterize a model, the basis for which is a bold prediction: Future increases in VPD may push lianas over their survivability threshold across much of the neotropics. Such a prediction is not necessarily expected, given the secular rise in liana abundance across the Americas over the past decades, and the observation that liana abundance peaks in the most seasonal of tropical forests where drought stress is most pronounced. Therefore, if the authors conclusions are robust, such unintuitive predictions are indeed of great significance.

They have gone to great lengths to detail their approach and model used, in addition to an extensive sensitivity analysis – for that they should be commended. The model they used strikes an elegant balance between the minimal amount of complexity needed to address the problem while simplifying (with good justification wherever they do so) in order to keep it tractable. Nonetheless, I have some significant concerns, which if they can be addressed, will make this a suitable publication for this journal.

General

No doubt K_s is greater on average in lianas than trees; this is well supported by a literature review on the topic as well as the meta-analysis. The real question is by how much. I worry that the ~threefold difference in K_s in lianas is potentially exaggerated, since the effect size, which is based on the mean, may be affected by a (relatively) small number of liana species with very large K_s (Fig 1). Why is the Glass' effect size as stated in main text and Extended Data Table 2 (2.69) different from Suppl Results #1 (6.72)? One way to assess the role of (potential) outliers in K_s for lianas is to compute the effect size relative to lianas (which have much larger variance) rather than to trees.

More broadly, I am concerned that the broad-stroke statistical tests (Mann-Whitney) without regard to potential underlying geographical disparities between lianas and trees can affect the results. I certainly agree that meta-analysis is a powerful approach, the conclusions of which may legitimately contradict previous studies that were limited in scope to a single or handful of site (e.g., as authors state line 92). A more rigorous approach to assessing liana-tree trait differences would be to ensure an apples-to-apples comparison in terms of geographic extents; traits do correlate with climate and soils after all! This could be done any number of ways, e.g., repeating tests within regions, incorporating geographic region as a random effect within a mixed model, or even conducting a subset of tests dedicated to the all studies that compared/contrasted lianas and trees trait values. If sample sizes are too small/do not permit, the authors should at least acknowledge this as a possibility.

I would expect liana stems to have a lower construction cost (be less dense) than trees since they do not need to support their weight, so it is a bit surprising that wood density did not differ between growth forms, also because a difference in wood density would be expected if the two forms have different turnovers as you note here. Do you think sample size is great enough for this to be a robust conclusion? Construction (or maintenance) cost could comprise an essential part of a growth – mechanical safety tradeoff (Larjavaara & Muller Landau 2010). Differences in these costs between lianas and trees could also comprise a significant amount of total NPP and thus could be relevant here. I think it is worth at least noting this as an important unknown not explored in this approach.

I am getting contradictory messages about how the model implements hydraulic conductivity. Looking at Supplement of Trugman et al. (2018) reveals that the Kirchoff transform is used in this model, which entails integrating stem-specific hydraulic conductivity (an intensive quantity and dependent on Y) along the path. The Kirchoff transform circumvents the need for a whole-plant hydraulic conductivity as you describe it here (if I understand correctly). However, you seem to suggest (L479) that the model uses a single whole-plant value, and interpret it as such in Fig 2 and elsewhere, so I do not understand how you arrive at a K_w based on the model of Trugman et al. (2018). Second, of equal concern are unit issues. You cite Nardini and Salleo (2000) as a basis for assuming that whole-plant conductivity (K_w) is an order of magnitude less than branch-specific conductivity (K_s). But in that paper they are referring to conductance (note units are m^{-2} in the

denominator, not $m-1$, which signals that this is an extensive, not an intensive, quantity). If anything, a whole-plant K_w (of the units you give, m^{-1} , an intensive quantity), should be greater, not lesser, than branch K_s because conduits in stems get wider from branches to base. This, then, affects your inferred K_{req} and potentially the conclusions which follow. Perhaps I am missing something with all of this and there are no errors here, and if so, a good clarification in the text is needed. (Note, in some places you use K_s and K_w interchangeably, as in Ext Data Fig 1, there may be others, please check).

Specific Comments

L127-134: Yes, this is definitely spot-on. This was shown to be the case in a recent global meta-analysis of H_v (Mencuccini et al. 2019). While that paper emphasized a continuum in H_v - K_s scaling which included lianas, your meta-analysis can and should demonstrate that lianas sit at one end of that continuum; intuitively the growth form dichotomy suggests that there may be some degree of separation from the rest of the continuum?

Line 184: I would replace 'organism size' with 'hydraulic path length', as that more closely reflects the specific aspect of size related to this prediction. As you noted elsewhere, the use of tree height as an approximation to liana stem length will yield conservative predictions, given their almost certain longer length than tree height.

Fig 2 – I would plot this against H_v instead, since leaf area is held constant. This would effectively make the different canopy occupancy lines converge to a common line; they would just be at different extremes of H_v . There is a benefit to displaying the graph in terms of liana DBH, but it can be misleading, since what needs to be emphasized is the gradient in hydraulic supply per unit demand.

L392: I don't follow. Isn't DBH simply a free parameter that you varied independent of the competition scenario?

L449: These are easily excluded and thus this should be easy to test, no?

Extended Data Table 4: How can liana stem length be constant at 18.2 yet still be dynamic via stem turnover?

References:

Mencuccini, M., Rosas, T., Rowland, L., Choat, B., Cornelissen, H., Jansen, S., Kramer, K., Lapenis, A., Manzoni, S., Niinemets, Ü. and Reich, P.B., 2019. Leaf economics and plant hydraulics drive leaf: wood area ratios. *New Phytologist*, 224(4), pp.1544-1556.

Larjavaara M, and Muller-Landau HC (2010) PERSPECTIVE: Rethinking the value of high wood density. *Functional Ecology* 24:701-705

Response to Reviewers

In the following response to reviewers, the reviewers' comments remain in regular text. We add **our responses and explanations** below each comment in **blue**. We have copied and pasted **short new sections from the manuscript** in **green** offset by quotations. For longer revisions, we include reference numbers and refer the reader to the Appendices 1 & 2 at the end of this response. Changes in the **manuscript** are indicated **by purple text**. Changes to the Supplementary Discussion, Figures, and Tables are included in the response and their location is given by section title but not by line numbers or purple text because this formatting has been removed for final submission of this manuscript.

REVIEWER COMMENTS

Reviewer #1 (Remarks to the Author):

This study by Wilson et al. identifies important research and measurement needs of liana traits and anatomy and employs a mechanistic model for competition to evaluate liana viability in future climate conditions. The authors supplement liana measurements available in the TRY database with other data sources listed in the Extended Data Table 3 and thus identify where measurements are lacking, especially belowground. They also adapt a tree carbon allocation model used in Trugman et al. 2018 to conclude that liana viability might be threatened under future hydroclimate condition in the tropics.

This is an important study with many qualities. The meta-analysis of the extended dataset was clear, described differences between growth forms clearly, and accurately identified sample size limitations. I appreciate the authors' sharing of the data collection process and analysis on github for reproducibility. The science in this manuscript is very thoughtful; the authors have provided deep explanations of the different assertions made throughout. I often found my questions answered after further reading, especially in the "Supplementary Methods" and the "Supplementary Results and Discussion."

We thank the reviewer for their positive feedback. We are also very grateful for the thoughtful review that follows.

My main concern for this manuscript is the obscurity of what the model does, what are the inputs? What does it simulate? What processes does it consider? Throughout, the authors refer to agreements between model output and empirical observation to support model accuracy (e.g., lines 121-123 and lines 153-155). While the manuscript does refer to the Trugman et al (2018) paper for the model, I feel it important to at least give a brief recap of the model. Then, the supplementary section titled "competition model" will be more easily understood.

We thank the reviewer for pointing out this issue. We have added a new section to the Methods with an enhanced description of the model (Section "Competition model," lines 275-324). Because of the length of this addition, we have added the text to this response to reviewers in an Appendix below (Appendix 1).

A consequence of the above is that it was hard for me to understand how why liana Kreq was so sensitive to some of the parameters mentioned in Supplementary figures 10 and 11. The authors do a great job of showing how the data constrained the model parameters to which Kreq was most sensitive. I suggest adding to this by expanding 1) upon model description and 2) the discussion on the relevant mechanisms making liana Kreq so sensitive to hydroclimatic changes (is the Huber value enough to explain the difference? Lines 127-134). These would help the reader further trust the modeling process.

As suggested, we expanded on the model description on lines 275-324. Further, in the main text, we added a short discussion of the relationship between Huber value and conductivity with support from a recent meta-analysis brought to our attention by Reviewer 2 (lines 124-125, 127-128, 132-134). In addition, we expanded upon the results section of our sensitivity analysis in the supplement to more explicitly link the sensitivity analysis to model structure (Supplementary Section “Sensitivity Analysis”).

- **“Hydraulic traits influence viability” (main text):** Second, we find that both lianas and trees with lower Huber value (sapwood area per unit canopy area) have higher K_{req} because the xylem supplies relatively more leaves with water (Figure 2). This pattern indicates that the unique liana allometry influences its physiology, consistent with the structure of our model (Methods: Competition Model) and the theoretical model derived by Mencuccini et al.⁴²; specifically, a lower Huber value, characteristic of lianas in comparison to trees^{3,5}, demands higher K_{req} to supply leaves with a consistent source of water, thus maintaining positive NPP. Therefore, the observed and modeled difference in K_{req} between trees and lianas is the result of a physiological difference between growth forms that represents a fundamental source of variation between woody growth forms in tropical forest biomes. This conclusion is supported by a recent meta-analysis quantifying the empirical relationship between K_s and Huber value globally⁴².
- **“Sensitivity analysis” (Supplementary Discussion):** The parameters to which K_{req} are most sensitive are mainly those associated with the pipe model in our coupled modeling framework (Methods: Competition Model). Specifically, diameter at breast height (DBH), P_{50} (b2), tree height or the hydraulic path length (Lx), leaf area (AL), specific leaf area (SLA), and the biomass ratio of fine roots to leaves (q) determine the xylem path length (Lx), water to the leaves (DBH, AL, and SLA, q), and sensitivity to tension in the xylem (b2). The other parameter, rd, is a major component of total plant respiration, which is subtracted from photosynthesis in the computation of NPP from net photosynthesis (A_{net}). This indicates that our modeling framework emphasizes the influence of water transport on photosynthesis using well-established models of water transport and photosynthesis.

Another thing to clarify: does the model consider possible liana P50 adaptation? Would this reduce future Kreq and change the conclusions of the study?

Our model does not currently consider liana nor tree P_{50} adaptation and we thank the reviewer for drawing attention to this limitation to our modeling framework. To address the possibility of P_{50} adaptation in lianas, we have conducted additional simulations that allow both the tree and the liana to experience more negative P_{50} values (see our response to the comment directly below) and we have added text to the main text referencing this addition to the supplement (Supplementary Section “Hydraulic trait-climate interactions”). Given that liana K_{req} continues to be greater than tree K_{req} and that the change in liana K_{req} from the 2000 to 2100 is greater than the change in tree K_{req} over the same time period, we conclude that P_{50} adaptation does not change the conclusions we draw from our simulations. However, the magnitude of the change in

K_{req} from present-day to our scenario for 2100 is dependent on the value of P_{50} in the future simulations.

- **“Hydraulic trait-climate interactions” (main text):** The increase in K_{req} persists under different total canopy area and competition scenarios, as well as under an assumption of adapting (i.e., decreasing) P_{50} (Supplementary Discussion: Model results).

The choice to use static parameter values for P_{50} was made deliberately to avoid issues with parameterizing the slope of the percent loss of conductivity curve (parameter b1 in our model; Methods: Competition Model, Supplementary Table 4) with ecologically relevant values. This is a concern because of the strong correlation between b1 and b2 (P_{50} ; Supplementary Figure 4). Allowing P_{50} to vary would require concurrent variations in b1. Since there are few data available to constrain the range of possible combinations of b1 and b2, particularly for lianas, in the future (van der Sande et al. 2013), we chose to use static, empirically derived parameter values for these two parameters in our manuscript. Indeed, trait adaptation/acclimation is an area of active research with many unanswered questions, making this aspect difficult to address in our study by any other means than parameter sensitivity tests.

To our knowledge, the only publications measuring liana and tree P_{50} across a climatic gradient occur in tropical forests with contrasting hydroclimates and indicate that, when considering both lianas and trees together, P_{50} tends to be greater in wetter than drier tropical forests (Zhu et al. 2017; Medina-Vega et al. 2019). However, in the only study to compare P_{50} of the same liana species across multiple forest types (Zhu et al. 2017), the authors found that there is no difference in P_{50} of two liana species living in forests with contrasting water availability.

Is a sensitivity analysis of K_{req} around a decrease in P_{50} (more negative) possible?

Yes, it is possible. We conducted additional model simulations with decreased liana and tree P_{50} (parameter b2) in the future (using our climate scenario for the year 2100). In addition to decreasing P_{50} , we simultaneously scaled the slope of the percent loss of conductivity curve (parameter b1) using the linear relationship between these two parameter values we derived from our meta-analysis of both liana and tree observations (shown in Supplementary Figure 4). We chose to allow lianas and trees to experience the same P_{50} adaptation because we had parameterized lianas and trees identically for the parameters b1 and b2 and there is no evidence to support that lianas, but not trees, will experience P_{50} adaptation in the future. The parameterizations for these scenarios are as follows: b2 = -2.25 MPa, b1 = 0.92 % MPa⁻¹; b2 = -2.5 MPa, b1 = 0.73 % MPa⁻¹; and b2 = -3 MPa, b1 = 0.49 % MPa⁻¹. The most extreme liana P_{50} observation in our extended meta-analysis is -2.99 MPa, suggesting that these additional simulations on average capture the range of P_{50} values currently observed among tropical lianas. The methods for these additional simulations and the results are presented in “Supplementary Discussion: Model results”. A figure corresponding to the results has been added to the supplement (Supplementary Figure 16).

- In the future, we acknowledge the possibility that liana and tree communities may physiologically adapt to drying hydroclimate by increasing cavitation resistance (i.e., lower average liana and tree P_{50}). Due to the simplicity of our model and the strong and uncertain correlation between P_{50} and the slope of the percent loss of conductivity curve (Supplementary Figure 4), we did not consider the possibility of P_{50} adaptation in our future climate scenario simulations in order to vary as few parameters as possible. However, it is possible that greater cavitation resistance could result in lower K_{req} via the hypothesized trade-off between xylem efficiency and safety¹⁵. To address the possibility that K_{req} may be lower among lianas and trees in the future if P_{50} adaptation occurs, we conducted additional simulations of liana and tree K_{req} with lower P_{50} parameterizations, corresponding to higher cavitation resistance. Because of the strong empirical correlation between P_{50} and the slope of the percent loss of conductivity curve (Slope), we simultaneously varied these two parameters in three additional scenarios, with hydroclimate conditions predicted for 2100, as above (i.e., 100% increase in VPD, no change in $[\text{CO}_2] = 550$ ppm). We used the “established” competition scenario and assumed the same adaptation scenarios for both liana and tree K_{req} simulations. The three scenarios are as follows: b1 = 0.92 % MPa^{-1} , b2 = -2.25 MPa; b1 = 0.73 % MPa^{-1} , b2 = -2.5 MPa; and b1 = 0.49 % MPa^{-1} , b2 = -3 MPa. The most extreme liana P_{50} observed in the literature we included in our extended meta-analysis is -2.99 MPa; therefore, our P_{50} adaptation scenarios are consistent with the most drought-tolerant observations of present-day liana P_{50} . Results of our sensitivity analysis indicate that P_{50} adaptation has the potential to lower K_{req} for both lianas and trees (Supplementary Figure 16). As P_{50} decreases, K_{req} decreases under both the drier and wetter site scenarios for the year 2100. Under the wetter hydroclimate scenario, when $P_{50} = -2.25$ MPa, tree $K_{req} = 1.84 \text{ mol m}^{-1} \text{ s}^{-1} \text{ MPa}^{-1}$ and liana $K_{req} = 42.2 \text{ mol m}^{-1} \text{ s}^{-1} \text{ MPa}^{-1}$ while when $P_{50} = -3$ MPa, tree $K_{req} = 1.30 \text{ mol m}^{-1} \text{ s}^{-1} \text{ MPa}^{-1}$ and liana $K_{req} = 29.3 \text{ mol m}^{-1} \text{ s}^{-1} \text{ MPa}^{-1}$ (compared to tree $K_{req} = 2.22 \text{ mol m}^{-1} \text{ s}^{-1} \text{ MPa}^{-1}$ and liana $K_{req} = 52.1 \text{ mol m}^{-1} \text{ s}^{-1} \text{ MPa}^{-1}$ with no P_{50} adaptation). Under the drier hydroclimate scenario, when $P_{50} = -2.25$ MPa, tree $K_{req} = 5.09 \text{ mol m}^{-1} \text{ s}^{-1} \text{ MPa}^{-1}$ and liana $K_{req} = 121 \text{ mol m}^{-1} \text{ s}^{-1} \text{ MPa}^{-1}$ and when $P_{50} = -3$ MPa, tree $K_{req} = 3.54 \text{ mol m}^{-1} \text{ s}^{-1} \text{ MPa}^{-1}$ and liana $K_{req} = 83.8 \text{ mol m}^{-1} \text{ s}^{-1} \text{ MPa}^{-1}$ (compared to tree $K_{req} = 6.25 \text{ mol m}^{-1} \text{ s}^{-1} \text{ MPa}^{-1}$ and liana $K_{req} = 150 \text{ mol m}^{-1} \text{ s}^{-1} \text{ MPa}^{-1}$ with no P_{50} adaptation). This represents a significant decrease in K_{req} , particularly for lianas. However, K_{req} remains greater for 2100 than at present for all scenarios even under the most extreme P_{50} adaptation scenario we considered (present day liana $K_{req} = 25.6 \text{ mol m}^{-1} \text{ s}^{-1} \text{ MPa}^{-1}$ and $71.3 \text{ mol m}^{-1} \text{ s}^{-1} \text{ MPa}^{-1}$ under wetter and drier hydroclimate scenarios, respectively and present day tree $K_{req} = 1.14 \text{ mol m}^{-1} \text{ s}^{-1} \text{ MPa}^{-1}$ and $3.00 \text{ mol m}^{-1} \text{ s}^{-1} \text{ MPa}^{-1}$ under wetter and drier hydroclimate scenarios, respectively). This suggests that drying hydroclimate in the future is likely to impose a greater physiological demand on plant hydraulic architecture, particularly for lianas, regardless of the ability of the plant to experience P_{50} adaptation.

A low resolution version of the Supplementary Figure is below.

Figure Caption: Change in required whole-plant stem-specific hydraulic conductivity (K_{req}) as vapor pressure deficit (VPD) increases according to future projections for Central America. The x-axis is a multiplier of increase from the present. For example, 2.00 means VPD is doubled from the current hourly values for each month. The lines represent K_{req} under potential future VPD conditions spanning 1x to 2x current VPD at the dry forest site, Horizontes, Costa Rica (red) and the moist forest site, Barro Colorado Island, Panama (BCI, blue). The left panel shows tree K_{req} and the right panel shows liana K_{req} . Symbols at 2.00 on the x-axis of each panel represent K_{req} under various conditions of P_{50} adaptation when VPD is doubled from present. Triangle: tree P_{50} = liana P_{50} = -2.25 MPa; diamond: tree P_{50} = liana P_{50} = -2.5 MPa; square: tree P_{50} = liana P_{50} = -3.0 MPa.

Minor comments:

Figure 2: Why does the tree K_{req} change when tree canopy cover fraction changes? I don't think this is obvious from the text.

We thank the reviewer for pointing out that we did not address the change in tree K_{req} in this portion of the manuscript. We have added text to the discussion of Figure 2 in the main text (lines 124-125) to indicate that both liana and tree K_{req} are sensitive to changes in Huber value. To address a suggestion from Reviewer 2, we have also changed Figure 2 to show K_{req} as a function of Huber value as well as a function of DBH. We believe this change also aids the reader in understanding how change in tree leaf area (when DBH is held constant, i.e., a change in Huber value) results in a change in K_{req} .

- **“Hydraulic traits influence viability” (main text):** Second, we find that both lianas and trees with lower Huber value (sapwood area per unit canopy area) have higher K_{req} because the xylem supplies relatively more leaves with water (Figure 2).

A low resolution version of Figure 2 is available below.

Figure caption: **Allometry and climate affect hydraulic conductivity.** Required hydraulic conductivity (K_{req}) as a function of diameter at breast height (DBH, **A**) and Huber value (sapwood area [cm^2] per unit leaf area [m^2], **B**), and hydroclimate (tropical moist forest or tropical dry forest). Total canopy area = 200 m^2 . In both, solid lines represent liana K_{req} and colors represent the different combinations of hydroclimate scenario and fraction of the canopy occupied by each growth form. See legend for the fraction of the canopy occupied by the liana (the tree occupies the rest of the canopy) and the forest type corresponding to the climate data. **(A)** Dashed lines represent tree K_{req} at a reference scenario with tree DBH = 18.2 cm. **(B)** “X” marks tree K_{req} at the same reference scenario with DBH = 18.2 cm.

Lines 351-353: Why was this specific threshold chosen? Is it possible it might cause the result in Fig. 1 that the P50s of trees and lianas are similar to be an artifact of data selection?

We originally chose to filter liana P_{50} observations in this way because extremely high P_{50} values are unrealistic given how frequently tropical lianas experience such conditions naturally. Instead, it is likely that these observations incorrectly result from applying techniques for measuring P_{50} developed for trees to the liana growth form. Therefore, we followed the filtering technique developed by Trugman et al. (2020), where P_{50} values greater than -0.75 MPa are removed from the analysis, to address the same methodological concern.

To assess the robustness of our approach, we re-analyzed the P_{50} data including the extreme liana values and report the statistics in the methods section (section “Methods: Extended meta-analysis”, lines 235-244). We have chosen to continue to filter liana P_{50} in the main text because these values likely represent methodological error, as explained above.

- **“Extended meta-analysis” (Methods):** This filtering was performed to reduce the probability that falsely high (i.e., less negative) P_{50} values were retained in our analysis because of improper measurement technique and is consistent with the P_{50} filtering performed by Trugman et al.⁵⁶. Improper measurement technique is a particular concern for lianas, whose wide and long vessels require cautious implementation of the

traditional measurement techniques developed for trees. We note that retaining all liana P_{50} observations (i.e., not filtering out observations > -0.75) results in a significant difference between trees and lianas (Mann-Whitney test statistic = 1,029, $n_{\text{tree}} = 61$, $n_{\text{liana}} = 46$, $p < 0.05$). However, the effect size remains relatively small, indicating that even when retaining unrealistically high liana P_{50} values, the difference between liana and tree P_{50} is ecologically of only moderate significance (Glass' = 0.47).

Extended Data Figure 1: Abscissa should be Kw not Ks

We thank the reviewer for pointing out this typo in the figure. Extended Data Figure 1, now Supplementary Figure 1, has been replaced in the manuscript with the x-axis labeled “Kw” and with corresponding units (“mol m⁻¹ s⁻¹ MPa⁻¹”). A low resolution version of the updated figure is below.

Figure Caption: Conceptual diagram of how required whole-plant hydraulic conductivity (K_{req}) is defined for lianas. Net primary production (NPP) changes as a function of diameter at breast height (DBH, shown: 2 cm DBH (red solid line), 6 cm DBH (blue solid line)), length (shown: ≈ 14 m), canopy area (shown: 80 m^2), and whole-plant stem-specific hydraulic conductivity (K_w). At low K_w , NPP is negative because no photosynthesis occurs but respiration continues. As K_w increases, NPP increases and the rate at which NPP increases is influenced by stem allometry because the dimensions of the stem determine the xylem area supplying the canopy with water and the distance water must travel through the xylem to reach the canopy. K_{req} (dashed vertical black lines) is defined as the smallest value of K_w yielding positive NPP.

Reviewer #2 (Remarks to the Author):

Review of “Climate and hydraulic traits interact to set thresholds for liana viability”
 Nature Communications
 15-June-2021

Summary

Willson et al. use a meta analysis of liana functional traits to parameterize a model, the basis for which is a bold prediction: Future increases in VPD may push lianas over their survivability threshold across much of the neotropics. Such a prediction is not necessarily expected, given the secular rise in liana abundance across the Americas over the past decades, and the observation that liana abundance peaks in the most seasonal of tropical forests where drought stress is most pronounced. Therefore, if the authors conclusions are robust, such unintuitive predictions are indeed of great significance.

They have gone to great lengths to detail their approach and model used, in addition to an extensive sensitivity analysis – for that they should be commended. The model they used strikes an elegant balance between the minimal amount of complexity needed to address the problem while simplifying (with good justification wherever they do so) in order to keep it tractable. Nonetheless, I have some significant concerns, which if they can be addressed, will make this a suitable publication for this journal.

We thank the reviewer for this positive feedback, and also for the thoughtful criticisms below.

General

No doubt K_s is greater on average in lianas than trees; this is well supported by a literature review on the topic as well as the meta-analysis. The real question is by how much. I worry that the ~threefold difference in K_s in lianas is potentially exaggerated, since the effect size, which is based on the mean, may be affected by a (relatively) small number of liana species with very large K_s (Fig 1). Why is the Glass' effect size as stated in main text and Extended Data Table 2 (2.69) different from Suppl Results #1 (6.72)? One way to assess the role of (potential) outliers in K_s for lianas is to compute the effect size relative to lianas (which have much larger variance) rather than to trees.

The difference between the statistics reported in the main text and in the Supplementary Discussion is that the main text reports statistics from the extended meta-analysis, which includes some data from the TRY database and the additional literature on tree and liana hydraulic traits, while the Supplementary Discussion reports statistics from the TRY database alone, with observations not having been filtered using our criteria for hydraulic conductivity (Methods: Extended meta-analysis). We added language to the Supplementary Discussion to explain this discrepancy (section “Supplementary Discussion: TRY meta-analysis”).

- **“TRY meta-analysis” (Supplementary Discussion):** It is important to note that these statistics were computed on data not filtered using our more stringent requirements for hydraulic trait observations (Methods: Extended meta-analysis) and these statistics were computed only on observations from the TRY database, not including additional literature from our extended meta-analysis.

We acknowledge that the results may be influenced by the greater variation in liana K_s than tree K_s , particularly with the extremely high K_s observed for some liana species. We chose the Glass' Δ statistic to accommodate the unique structure (i.e., distribution and variability) of plant

functional trait data. The reasoning behind using this statistic has been expanded upon in the methods section of the manuscript (Section “Methods: Statistical analyses”; lines 265-271).

- **“Statistical analysis” (Methods):** Throughout the text, we present the statistics computed using the tree as the reference group for two reasons. First, we were interested in the degree to which lianas differ from the well-parameterized tree plant functional types in dynamic vegetation models. Second, because lianas have been parameterized using data from early successional tropical trees⁵⁰, we were interested in considering the degree to which the distribution of liana trait values differs from the distribution of tree trait values.

We recognize that this methodological choice has implications for the magnitude of the effect size computed. Therefore, we have added three columns to each of our tables describing effect size (Supplementary Figure 2, Supplementary Figure 8) that describe the effect size and confidence intervals when using the liana growth form as the reference group. This addition was also described in the methods section (Section “Methods: Statistical analysis”; lines 261-265). The magnitude of the effect size is different depending on whether the liana or tree growth form is used as the reference group as a result of the difference in variance of the observations corresponding to each growth form. More observations of liana functional traits could better constrain the magnitude of the difference in functional traits between growth forms by capturing the variability of functional trait values more comprehensively. However, the large effect sizes observed for some traits (e.g., K_s) with relatively sparse observations for lianas is evidence that a biologically significant difference in these traits between growth forms is already apparent. We included a discussion of how additional trait observations are useful for constraining these statistics in the Supplementary Discussion (Section “Supplementary Discussion: TRY meta-analysis”).

- **“Statistical analysis” (Methods):** To avoid biasing our interpretation of the statistics by considering only one growth form as the reference group, we computed and present the test statistic and 95% confidence intervals resulting from using both the tree growth form (subscript “T”) and liana growth form (subscript “L”) as the reference group (Supplementary Table 2; Supplementary Table 6).
- **“TRY meta-analysis” (Supplementary Discussion):** To begin to address this, we further pursued the significance of hydraulic trait differences between liana and tree growth forms with more empirical observations in our extended meta-analysis. We additionally suggest that future data collection efforts focus on increasing the number of liana species on which traits are measured to further constrain variation in liana functional trait distributions. The substantial dependence of the effect size statistic on the choice of reference group provided strong reasoning to further pursue the significance of hydraulic trait differences between liana and tree growth forms with more empirical observations in our extended meta-analysis.

Low resolution tables of the expanded statistical analysis are below.

Effect size for extended meta-analysis						
Trait	Glass' Δ_T	Lower CI_T	Upper CI_T	Glass' Δ_L	Lower CI_L	Upper CI_L
Stem-specific hydraulic conductivity ($\text{mol m}^{-1} \text{s}^{-1} \text{MPa}^{-1}$)	2.69	1.28	4.08	-0.55	-0.85	-0.25
P_{50} (MPa)	0.35	0.00	0.69	-0.47	-0.94	0.01
Slope of PLC curve ($\% \text{MPa}^{-1}$)	0.78	-0.35	1.87	-0.59	-1.43	0.29

Effect size for TRY traits						
Trait	Glass' Δ_T	Lower CI_T	Upper CI_T	Glass' Δ_L	Lower CI_L	Upper CI_L
Stem specific density (g cm^{-3})	-0.52	-0.87	-0.17	0.50	0.14	0.86
Vessel diameter (μm)	1.50	0.98	2.02	-0.72	-0.99	-0.44
Vessel density (1 mm^{-2})	-0.05	-0.24	0.15	0.04	-0.14	0.23
Stem specific hydraulic conductivity ($\text{mol m}^{-1} \text{s}^{-1} \text{MPa}^{-1}$)	6.72	0.76	12.68	-0.52	-1.01	-0.02
P_{50} (MPa)	1.04	0.81	1.28	-6.32	-11.29	-1.47
Leaf lifespan (months)	-0.32	-0.50	-0.14	0.93	0.34	1.51
Specific leaf area ($\text{mm}^2 \text{mg}^{-1}$)	1.06	0.67	1.45	-0.59	-0.83	-0.36
Area-based photosynthetic rate ($\text{mmol CO}_2 \text{m}^{-2} \text{s}^{-1}$)	0.20	0.08	0.33	-0.36	-0.59	-0.13
Mass-based leaf nitrogen (mg g^{-1})	0.11	0.00	0.22	-0.12	-0.24	0.00
Mass-based leaf phosphorus (mg g^{-1})	1.32	1.03	1.60	-0.73	-0.90	-0.55
Leaf area (cm^2)	-0.18	-0.23	-0.13	0.37	0.27	0.48
Specific root length (m g^{-1})	0.05	-0.55	0.65	-0.04	-0.56	0.48
Fine root diameter (mm)	0.47	-0.29	1.24	-0.34	-0.91	0.23
Mycorrhizal colonization (%)	0.08	-0.60	0.77	-0.10	-0.90	0.71
Rooting depth (m)	-0.11	-0.56	0.33	0.21	-0.64	1.04

Because the qualitative results (i.e., direction and significance of statistical tests) remain the same regardless of reference group choice, we continue to present the Glass' Δ using the tree growth form as the reference group in the text. However, we now discuss the uncertainty in the magnitude of the effect size in the Supplementary Discussion (sections "Supplementary Discussion: TRY meta-analysis" and "Supplementary Discussion: Extended meta-analysis").

- “TRY meta-analysis” (Supplementary Discussion):** When the liana growth form is used as the reference group (i.e., liana growth form trait standard deviation is used in the denominator) of the Glass' Δ effect size statistic, the magnitude of the effect size remains relatively similar for all traits in the TRY database considered except for the hydraulic traits (i.e., K_s and P_{50}). For K_s , Glass' $\Delta = 6.72$ ([0.76, 12.68] 95% CI) when the tree growth form is used as the reference group and Glass' $\Delta = -0.52$ ([-1.01, -0.02] 95% CI) when the liana growth form is used. In both cases, the liana has greater K_s on average than the tree growth form, but the magnitude of the difference is substantially different when different reference groups are used. Similarly, for P_{50} , Glass' $\Delta = 1.04$ ([0.81, 1.28] 95% CI) when the tree growth form is used as the reference group and Glass' $\Delta = -6.32$

([-11.29, -1.47] 95% CI) when the liana growth form is used. More data would be useful for constraining the intra-growth form variability of these functional traits to understand the magnitude of differences between tree and liana growth forms.

- **“Extended meta-analysis” (Supplementary Discussion):** We note that these conclusions persist regardless of which growth form is used as the reference group in the calculation of Glass’ Δ . For K_s , Glass’ Δ is smaller in magnitude when the liana growth form is used as the reference group, reflecting the higher variance within the liana growth form than the tree growth form, but lianas still show significantly higher K_s on average than trees (Glass’ Δ using liana growth form as reference = -0.55). Both P_{50} and Slope remain non-significant when using lianas as the reference group.

More broadly, I am concerned that the broad-stroke statistical tests (Mann-Whitney) without regard to potential underlying geographical disparities between lianas and trees can affect the results. I certainly agree that meta-analysis is a powerful approach, the conclusions of which may legitimately contradict previous studies that were limited in scope to a single or handful of site (e.g., as authors state line 92). A more rigorous approach to assessing liana-tree trait differences would be to ensure an apples-to-apples comparison in terms of geographic extents; traits do correlate with climate and soils after all! This could be done any number of ways, e.g., repeating tests within regions, incorporating geographic region as a random effect within a mixed model, or even conducting a subset of tests dedicated to the all studies that compared/contrasted lianas and trees trait values. If sample sizes are too small/do not permit, the authors should at least acknowledge this as a possibility.

We thank the reviewer for their careful consideration of the statistical approaches we took to our meta-analysis. We agree that geographic extent could be important, but we also note that several technical hurdles must be overcome to address this problem. First, there is the problem of sample size, which the reviewer mentioned. Second, we aggregated data from the TRY database to the species level, prohibiting a one-to-one matching of our averages with specific geographic locations. Third, the location metadata from the TRY database is incomplete, although one group addressed this problem by matching species to their climate niche envelope (Mencuccini et al. 2019).

In light of these considerations, we used the following approach to identify trends over geographic and climate space for tree and liana hydraulic traits. The literature with hydraulic trait observations for lianas and trees was extracted from our extended meta-analysis. We extracted metadata from the publications on observation locations and local climatology, including latitude, longitude, altitude, dry season length, and season during which measurements were taken on hydraulic traits. Because only one study reports trait values for the slope of the percent loss of conductivity curve, we did not consider this trait. We conducted simple linear regressions and t-tests between K_s and P_{50} and each of the geographic and climate variables above. Results of this analysis are available in Appendix 2 to this letter. We find that none of the geographic or climatic variables explains a meaningful amount of variation in either K_s or P_{50} . We have chosen to not include the results or figures from this analysis in the manuscript because of the

nonsignificance of the analysis. However, we did add a paragraph expanding upon this concern and some statistics from our analysis, and we provide the data used in this analysis and the R code used to conduct the analysis in our Github repository (“Supplementary Discussion: Extended meta-analysis”).

- **“Extended meta-analysis” (Supplementary Discussion):** In both our TRY meta-analysis and our extended meta-analysis, we recognize that the Mann-Whitney and Glass’ Δ tests may oversimplify our comparison of functional traits between growth forms. Geography and the climatic and edaphic characteristics of the location at which observations were made may influence functional traits, in addition to growth form. For the hydraulic traits considered in our extended meta-analysis (i.e., K_s and P_{50}), we conducted simple linear regressions and t-tests with various geographic (latitude, longitude, altitude) and climatic (dry season length, season during which measurements were made) variables extracted from the meta-data of the literature we compiled to address this concern. We found that no geographic or climatic variable strongly correlated with tree and liana observations combined and none of our variables of interest explained more than 15% of variation in K_s (R^2_{adj} of tree K_s with altitude = 0.15; not shown) and 26% of variation in P_{50} (R^2_{adj} of liana P_{50} with altitude = 0.26; not shown) when tree and liana observations were considered separately. We do not report the results of this supplementary analysis here, but the data and code used to analyze these data are available in our Github repository. Trends between functional traits and geographic, climatic, and edaphic variables may be particularly relevant for the TRY meta-analysis, where observations for trees and lianas may have frequently come from different locations. We suggest that future research consider these ecosystem variables in pantropical comparisons of liana and tree functional traits using the climatic niche envelopes of liana and tree species.

Additionally, our pantropical approach to comparing liana and tree functional traits is not without precedent. Recently, Mello et al. (2021) compared liana and tree functional trait values from a pantropical database using t-tests without accounting for underlying geography or climate.

- **“Statistical analysis” (Methods):** This approach is consistent with a recent pantropical meta-analysis comparing liana and tree functional trait distributions⁵⁷.

I would expect liana stems to have a lower construction cost (be less dense) than trees since they do not need to support their weight, so it is a bit surprising that wood density did not differ between growth forms, also because a difference in wood density would be expected if the two forms have different turnovers as you note here. Do you think sample size is great enough for this to be a robust conclusion? Construction (or maintenance) cost could comprise an essential part of a growth – mechanical safety tradeoff (Larjavaara & Muller Landau 2010). Differences in these costs between lianas and trees could also comprise a significant amount of total NPP and thus could be relevant here. I think it is worth at least noting this as an important unknown not explored in this approach.

From the perspective of the reviewer's rationale, we agree that it is surprising that lianas did not have lighter wood than trees. And we acknowledge that lianas are proportionally less well represented than trees in TRY. However, our results are consistent with several other recent analyses. Guzman et al. (2020) also found that wood density does not differ between a subsample of tree and liana species in a single tropical forest, while Mello et al. (2021) concluded that lianas and trees do not differ in terms of wood density pantropically.

These points are now discussed in our Supplementary Discussion ("Supplementary Discussion: TRY meta-analysis").

- **"TRY meta-analysis" (Supplementary Discussion):** While the sample size for liana stem specific density observations is relatively low compared to trees, our conclusion that stem specific density is comparable between trees and lianas is supported by both site-specific⁵ and pantropical⁶ liana-tree functional trait comparisons.

We appreciate the reviewer's comment that the difference in stem specific density between growth forms may become significant in our modeling framework as a result of the influence of stem density on total respiration (and thus NPP). However, this is not the case for three reasons. First, our model shows limited sensitivity to wood density (parameter ρ), with Sensitivity = 0.00-0.01 for all scenarios we consider in our sensitivity analysis (Supplementary Tables 7 & 8). Furthermore, in our model, stem density is related only to the xylem respiration term (parameter r_x). Our sensitivity analysis shows that our results are relatively insensitive to changes in the r_x parameter (Sensitivity = 0.00 for all scenarios considered; Supplementary Tables 7 & 8). Third, our model does not relate hydraulic parameters (e.g., b_2 (P_{50})) and stem density, which would imply a direct relationship between stem density and results of our model, so the use of a common value of stem density is consistent with the lack of a relationship between hydraulic transport and stem density in our model and not a commentary on underlying physiology. Therefore, we argue that the specific value of stem specific density used to parameterize our model should not substantially affect the conclusions drawn from our modeling efforts. We included a statement on the sensitivity of our model to stem specific density in the discussion of our sensitivity analysis ("Supplementary Discussion: Sensitivity analysis").

- **"Sensitivity analysis" (Supplementary Discussion):** Neither liana nor tree K_{req} is sensitive to stem specific density or xylem respiration, which is influenced by stem specific density, making our modeling results insensitive to uncertainty in the magnitude of the difference between growth forms for this functional trait.

Finally, we recognize that wood economics spectrum theory suggests a trade-off between wood density and turnover that should persist across growth forms. However, we believe our choice to parameterize turnover rates differently between growth forms despite finding no significant difference in wood density is reasonable. The relationship between wood density and construction and maintenance costs (thereby influencing turnover rates sensu Larjavaara & Muller-Landau 2010) pertains to tree trunks, or the main stem in lianas. Our turnover parameter specifically considers turnover of terminal branches. The relatively high turnover rate of liana

terminal branches is supported in the literature (Ichihashi & Tateno 2015). Furthermore, the theoretical trade-off discussed by Larjavaara & Muller-Landau (2010) only considers the tree growth form and the authors state that, given the difference in woody tissue purpose (i.e., lower mechanical strength requirement) for lianas, the relationship between wood density and construction and maintenance costs may be different for this growth form.

I am getting contradictory messages about how the model implements hydraulic conductivity. Looking at Supplement of Trugman et al. (2018) reveals that the Kirchoff transform is used in this model, which entails integrating stem-specific hydraulic conductivity (an intensive quantity and dependent on Y) along the path. The Kirchoff transform circumvents the need for a whole-plant hydraulic conductivity as you describe it here (if I understand correctly). However, you seem to suggest (L479) that the model uses a single whole-plant value, and interpret it as such in Fig 2 and elsewhere, so I do not understand how you arrive at a Kw based on the model of Trugman et al. (2018).

We thank the reviewer for their thorough examination of the model. In response to a suggestion from Reviewer 1, we have added a more detailed explanation of the original model developed by Trugman et al. (section “Methods: Competition model”; lines 275-324). We hope this addition serves to provide some context for how we define K_w , which is equivalent to K_{max} in the model description. The full model description is available in Appendix 1.

Second, of equal concern are unit issues. You cite Nardini and Salleo (2000) as a basis for assuming that whole-plant conductivity (Kw) is an order of magnitude less than branch-specific conductivity (Ks). But in that paper they are referring to conductance (note units are m-2 in the denominator, not m-1, which signals that this is an extensive, not an intensive, quantity). If anything, a whole-plant Kw (of the units you give, m-1, an intensive quantity), should be greater, not lesser, than branch Ks because conduits in stems get wider from branches to base. This, then, affects your inferred Kreq and potentially the conclusions which follow. Perhaps I am missing something with all of this and there are no errors here, and if so, a good clarification in the text is needed.

We thank the reviewer for their detailed review of the methodology used in this manuscript and in the paper written by Nardini & Salleo. We misinterpreted the units in Nardini & Salleo’s analysis and we recognize that this error makes the scaling factor between K_{req} (i.e., required conductivity integrated over the whole plant, K_w) and K_s (i.e., the observations from our meta-analysis) incorrect. To our knowledge, there is no existing literature on the relationship between K_w and K_s for tropical trees or lianas. Therefore, we have substantially modified how we present the significance of the change in liana K_{req} in the future. First, we have replaced Figure 4 in the original manuscript with an updated figure that does not rely on a comparison between K_{req} and observations of K_s . Additionally, we modified the main text of the manuscript to reference the updated version of Figure 4. We now emphasize the magnitude of the difference in liana K_{req} between our present-day simulations (with hydroclimate consistent with the early 21st century) and our future scenario (with VPD and atmospheric CO₂ concentration consistent with 2100) (Section “Hydraulic trait-climate interactions”; lines 172-174).

Furthermore, we now argue that the magnitude of the change in K_{req} results in a physiologically stressful and potentially infeasible level of hydraulic adaptation for lianas in the future (Section “Hydraulic trait-climate interactions”; lines 177-182).

We believe this revision will result in a higher impact manuscript than our original manuscript for two reasons. First, eliminating the reliance on an uncertain scaling factor between K_{req} and K_s leads to higher certainty in the conclusions we draw from our simulations. Second, our simulations now clearly demonstrate the degree to which liana K_{req} is more sensitive to drying hydroclimate than tree K_{req} , suggesting the strong potential for a decline in liana viability in the future as K_{req} exceeds observed whole-plant conductivity. This point is particularly salient given that mortality as a result of hydraulic failure and associated carbon starvation among trees has already been observed as a result of drying hydroclimate and these mortality events occurred with minimal change in tree K_{req} according to our model simulations. This discussion is included in the manuscript (Section “Hydraulic trait-climate interactions”; lines 173-183).

- **“Hydraulic trait-climate interactions” (main text):** Overall, the pattern of increasing K_{req} continues for both trees and lianas, with liana K_{req} increasing faster than tree K_{req} as VPD increases, despite simultaneous increases in atmospheric carbon dioxide concentration (Supplementary Figure 2). The increase in K_{req} persists under different total canopy area and competition scenarios, as well as under an assumption of adapting (i.e., decreasing) P_{50} (Supplementary Discussion: Model results). However, the magnitude of the difference in K_{req} between the present and 2100 is greater for lianas than trees (tree $\Delta K_{req} = 2 \text{ mol m}^{-1} \text{ s}^{-1} \text{ MPa}^{-1}$, liana $\Delta K_{req} = 47 \text{ mol m}^{-1} \text{ s}^{-1} \text{ MPa}^{-1}$; Figure 4). Experimental and observational research has already attributed tree mortality to the effects of severe droughts and drying hydroclimate worsened by climate change and similar mortality events are expected in the future⁴⁷. The greater sensitivity of liana K_{req} than tree K_{req} to drying hydroclimate in our simulations implies that lianas may undergo similar mortality events as K_{req} becomes greater than whole-plant conductivity, reinforcing our prediction that a threshold for liana viability may be reached under 21st century climate change (Figure 4, Supplementary Figure 3).

A low resolution version of Figure 4 is below.

Figure Caption: Increase in liana and tree K_{req} under present (2000) and future (2100) climate conditions at the tropical dry forest site (Horizontes, Costa Rica). K_{req} is computed under the established liana scenario (60% tree canopy occupancy, 40% liana canopy occupancy of 200 m² total canopy area). Blue: tree K_{req} , red: liana K_{req} . Labels indicate the change in K_{req} from present to 2100 for each growth form with units of mol m⁻¹ s⁻¹ MPa⁻¹.

In addition to the changes in the results, we changed the Methods to align with our revised version of Figure 4 in three places. First, we revised our discussion of the relationship between whole-plant conductivity (K_w) and branch specific conductivity (K_s) to urge caution in interpreting our modeling results in light of K_s observations (section “Methods: Simulations”; lines 438-439). Second, we removed two sentences from the end of the same section (section “Methods: Simulations”) that explained how we compared K_s and K_w in the original version of Figure 4. Finally, we removed a sentence describing the relationship between K_w and K_s from the main text (section “Hydraulic traits influence viability”). In this way, our interpretation of the simulations no longer relies on scaling between K_w and K_s .

- “Simulations” (Methods):** We emphasize that the model depends on K_w , whereas it is much more common to measure terminal branch K_s . **Because of the uncertainty associated with scaling between K_w and K_s , our estimates of K_{req} should be compared to observed K_s with caution.** To reduce uncertainty in this parameter, we urge further measurements of K_w .

We additionally changed the format of Supplementary Figures 3, 14, and 15 to be consistent with the format of Figure 4 in the main text. Correspondingly, we altered the text associated with these figures in the Supplementary Discussion. Specifically, we compare the results of the invasion scenario assuming a 200 m² total canopy area (our baseline assumption) in “Supplementary Discussion: Model results”.

- **“Model results” (Supplementary Discussion):** In the main text, we suggest that lianas may reach a hydraulic trait threshold for viability under the established scenario by 2100, under the prediction of a 100% increase in VPD, while the tree growth form, as a whole, appears less affected than lianas. Under the invasion scenario, these conclusions persist (Supplementary Figure 3). Trees remain relatively insensitive to drying hydroclimate in terms of K_{req} ($\Delta K_{req} = 3 \text{ mol m}^{-1} \text{ s}^{-1} \text{ MPa}^{-1}$). Meanwhile, lianas experience a change in K_{req} approximately seven times greater ($\Delta K_{req} = 21 \text{ mol m}^{-1} \text{ s}^{-1} \text{ MPa}^{-1}$). The magnitude of the change in liana K_{req} is lower under the invasion scenario, which is intuitive because of the lower liana canopy area assumed (i.e., fewer leaves with which to supply water). The main difference we note between the established and invasion scenarios is that liana K_{req} at both time periods under the invasion scenario is substantially lower than under the established scenario. Specifically, at present, liana $K_{req} = 103 \text{ mol m}^{-1} \text{ s}^{-1} \text{ MPa}^{-1}$ under the established scenario and $47 \text{ mol m}^{-1} \text{ s}^{-1} \text{ MPa}^{-1}$ under the invasion scenario, while in 2100, liana $K_{req} = 150 \text{ mol m}^{-1} \text{ s}^{-1} \text{ MPa}^{-1}$ under the established scenario and $68 \text{ mol m}^{-1} \text{ s}^{-1} \text{ MPa}^{-1}$ under the invasion scenario (Figure 4; Supplementary Figure 3). We conclude that the threshold for liana survival will be reached for larger lianas (i.e., lianas with larger canopy area and lower Huber value) before smaller lianas (i.e., lianas with smaller canopy area and higher Huber value).

The supplementary figures are available below, along with their associated figure captions.

Supplementary Figure 3: Increase in liana and tree K_{req} under present (2000) and future (2100) climate conditions at the tropical dry forest site (Horizontes, Costa Rica). K_{req} is computed under the invasion scenario (10% liana canopy occupancy, 2 cm liana DBH, 90% tree canopy occupancy, ≈ 18.2 cm tree DBH). Total canopy area = 200 m^2 . Blue: tree K_{req} , red: liana K_{req} . Labels indicate the change in K_{req} from present to 2100 for each growth form with units of $\text{mol m}^{-1} \text{s}^{-1} \text{MPa}^{-1}$.

Supplementary Figure 14: Increase in liana and tree K_{req} under present (2000) and future (2100) climate conditions at the tropical dry forest site (Horizontes, Costa Rica) with differing total canopy areas. In all cases, K_{req} is computed under the established liana scenario (60% tree canopy occupancy, 40% liana canopy occupancy, 2.65 cm liana DBH, 18.2 cm tree DBH). Blue: tree K_{req} , red: liana K_{req} . Labels indicate the change in K_{req} from present to 2100 for each growth form with units of $\text{mol m}^{-1} \text{s}^{-1} \text{MPa}^{-1}$. Left: total canopy area = 150 m², right: total canopy area = 400 m².

Supplementary Figure 15: Increase in liana and tree K_{req} under present (2000) and future (2100) climate conditions at the tropical dry forest site (Horizontes, Costa Rica) with differing total canopy areas. In all cases, K_{req} is computed under the invasion liana scenario (90% tree canopy

occupancy, 10% liana canopy occupancy, 2 cm liana DBH, 18.2 cm tree DBH). Blue: tree K_{req} , red: liana K_{req} . Labels indicate the change in K_{req} from present to 2100 for each growth form with units of $\text{mol m}^{-1} \text{s}^{-1} \text{MPa}^{-1}$. Left: total canopy area = 150 m^2 , right: total canopy area = 400 m^2 .

(Note, in some places you use K_s and K_w interchangeably, as in Ext Data Fig 1, there may be others, please check).

We thank the reviewer for pointing out this typo. The axis label was changed in Extended Data Figure 1. A low resolution version of Extended Data Figure 1 is below.

Specific Comments

L127-134: Yes, this is definitely spot-on. This was shown to be the case in a recent global meta-analysis of H_v (Mencuccini et al. 2019). While that paper emphasized a continuum in H_v - K_s scaling which included lianas, your meta-analysis can and should demonstrate that lianas sit at one end of that continuum; intuitively the growth form dichotomy suggests that there may be some degree of separation from the rest of the continuum?

We appreciate the reviewer's positive comment and the additional citation provided. We suggest that, on average, lianas and trees represent a dichotomy that can be represented in global vegetation models using plant functional types. To more closely relate our results to the literature, we have added references to both the model (i.e., equations 1 & 2 of Mencuccini et al.) and empirical results describing the relationship between K_s and H_v from Mencuccini et al (2019) (section "Hydraulic traits influence viability", lines 127-128, 132-134).

- “Hydraulic traits influence viability” (main text):** This pattern indicates that the unique liana allometry influences its physiology, consistent with the structure of our model (Methods: Competition Model) and the theoretical model derived by Mencuccini et al.⁴²; specifically, a lower Huber value, characteristic of lianas in comparison to trees^{3,5}, demands higher K_{req} to supply leaves with a consistent source of water, thus maintaining positive NPP. Therefore, the observed and modeled difference in K_{req} between trees and

lianas is the result of a physiological difference between growth forms that represents a fundamental source of variation between woody growth forms in tropical forest biomes. This conclusion is supported by a recent meta-analysis quantifying the empirical relationship between K_s and Huber value globally⁴².

Regarding the relative separation between tree and liana growth forms in terms of Huber value and K_s , we suggest here that a reasonable starting place for parameterizing lianas in dynamic global vegetation models is to parameterize lianas differently from co-occurring trees (e.g., early successional tropical forest trees [de Porcia e Brugnara et al. 2019]). In this way, lianas will be better represented in global modeling efforts. However, we do not suggest that lianas represent a homogenous group or that lianas trait values have no overlap with tree trait values. Instead, consistent with recent research such as Coppieters (2021), we recognize that lianas should eventually be represented by multiple plant functional types in vegetation models and that liana trait values would exist along a continuum as tree trait values do.

Line 184: I would replace ‘organism size’ with ‘hydraulic path length’, as that more closely reflects the specific aspect of size related to this prediction. As you noted elsewhere, the use of tree height as an approximation to liana stem length will yield conservative predictions, given their almost certain longer length than tree height.

We thank the reviewer for this suggestion and have changed the text accordingly (line 186).

- **“Hydraulic trait-climate interactions” (main text):** The difference in K_{req} between lianas and trees is dependent on hydraulic path length and on VPD.

Fig 2 – I would plot this against Hv instead, since leaf area is held constant. This would effectively make the different canopy occupancy lines converge to a common line; they would just be at different extremes of Hv. There is a benefit to displaying the graph in terms of liana DBH, but it can be misleading, since what needs to be emphasized is the gradient in hydraulic supply per unit demand.

We thank the reviewer for this suggestion. Figure 2 has been revised to be a multi-panel plot with DBH and Huber value on the x-axis of different panels and the log of hydraulic conductivity on the y-axis. As the reviewer points out, the conclusions drawn from the figure do not change, but the new figure emphasizes the point that Huber value is the driver behind changes in K_s , along with differences in hydroclimate, as shown between the drier and wetter tropical forest scenarios in this figure. The caption to this figure has also been changed to reflect the multi-panel format and the new plot with Huber value on the x-axis. Finally, we modified the text corresponding to our discussion of Figure 2 slightly (section “Hydraulic traits influence viability”, lines 124-125, 127-128, 132-134) to align with our new x-axis in Figure 2.

- **“Hydraulic traits influence viability” (main text):** Second, we find that both lianas and trees with lower Huber value (sapwood area per unit canopy area) have higher K_{req} because the xylem supplies relatively more leaves with water (Figure 2). This pattern

indicates that the unique liana allometry influences its physiology, consistent with the structure of our model (Methods: Competition Model) and the theoretical model derived by Mencuccini et al.⁴²; specifically, a lower Huber value, characteristic of lianas in comparison to trees^{3,5}, demands higher K_{req} to supply leaves with a consistent source of water, thus maintaining positive NPP.

A low resolution version of the updated Figure 2 is below.

Figure caption: Allometry and climate affect hydraulic conductivity. Required hydraulic conductivity (K_{req}) as a function of diameter at breast height (DBH, left) and Huber value (sapwood area [cm^2] per unit leaf area [m^2], right), and hydroclimate (tropical moist forest or tropical dry forest). Total canopy area = 200 m^2 . In both, solid lines represent liana K_{req} and colors represent the different combinations of hydroclimate scenario and fraction of the canopy occupied by each growth form. See legend for the fraction of the canopy occupied by the liana (the tree occupies the rest of the canopy) and the forest type corresponding to the climate data. (A) dashed lines represent tree K_{req} at a reference scenario with tree DBH = 18.2 cm. (B) “X” marks tree K_{req} at the same reference scenario with DBH = 18.2 cm.

L392: I don't follow. Isn't DBH simply a free parameter that you varied independent of the competition scenario?

We thank the reviewer for drawing our attention to the ambiguity in this part of our methods section. For Figure 2, we do not assume a particular liana DBH; for all other simulations presented in this paper, we assume a liana DBH of 2.00 cm or 2.65 cm, depending on the competition scenario. We have added text to the Methods to address this question (section “Methods: Competition model”, lines 339-346).

- **“Competition model” (methods):** DBH is then treated one of two ways. In Figure 2, where we investigate the simultaneous effects of allometry (i.e., Huber value) and hydroclimate on K_{req} , we allowed DBH to vary between the minimum and maximum liana

DBH (1.86 and 10.7 cm, respectively) observed during a field survey in Guanacaste, Costa Rica and computed Huber value using the relationship between DBH and sapwood area and leaf area. In all other model simulations, we assigned liana DBH according to the competition scenario: 2.65 cm for the “established” scenario (equal to the mean of the observations from Guanacaste, Costa Rica) and 2.00 cm for the “invasion” scenario (the minimum stem diameter for a canopy liana; see “Model parameterization”).

L449: These are easily excluded and thus this should be easy to test, no?

Unfortunately, the geographical metadata given by the TRY database is lacking and precludes a systematic removal of species falling within ecoregions that co-occur longitudinally with tropical forests. That is, because mountainous and desert regions occur at the same latitudes as tropical and sub-tropical rainforests, particularly in Australasia, species occurring exclusively in mountain and desert ecosystems cannot be rapidly filtered out of the dataset using occurrence maps. We carefully reviewed the occurrence maps of every species for which an observation for any of our traits of interest was available via TRY (n = 44,222); to additionally review the biomes in which each species occurs would have been time prohibitive. We recognize this as a limitation of our use of the TRY database and therefore discuss this limitation in the Supplementary Discussion, as the reviewer points out.

Extended Data Table 4: How can liana stem length be constant at 18.2 yet still be dynamic via stem turnover?

We thank the reviewer for pointing out the ambiguity in the name of this parameter. Here, we consider stem length to be the “starting” stem length for the year. Each subsequent month, some stem length is lost, as described in the turnover routine. We have redefined this model parameter from “stem length” to “initial stem length” to address this ambiguity. Consequently, we have added this language to the Methods (sections “Methods: Competition model” and “Methods: Model parameterization”; line 349 and line 376, respectively) and we changed the parameter definition in Supplementary Table 4.

- **“Competition model” (methods):** We developed a turnover routine to account for differences in leaf and stem turnover between trees and lianas. The routine works as follows: during a given month, a small amount of stem is lost **from an initial stem length at the beginning of the year (model parameter L_x)**, which corresponds with one-twelfth of the average annual stem turnover of the tree or liana.
- **“Model parameterization” (methods):** The only model inputs that differed between the tree and liana growth forms were whole-plant stem-specific hydraulic conductivity (K_w), DBH, leaf area, turnover, and **initial** stem length (Supplementary Table 4).

A low resolution version of the section of Supplementary Table 4 containing the revision is below.

Changed Model Parameters					
Name	Definition	Value	Units	Source	Tree or Liana Function
ax	Functional xylem cross-sectional area	$\text{Min}(\text{tot area}, (2.41 * (\text{tree dbh}/2)^{1.97} * 0.0001))$	m ²	Reyes-García et al. 2012	T, L
b1	Slope of PLC curve	1.79	% MPa ⁻¹	Meta-analysis	T, L
b2	P ₅₀	-1.91	MPa	Meta-analysis	T, L
b2Ht	DBH to height allometric constant	0.455		Smith-Martin et al. (unpublished)	T
Ca	Atmospheric [CO ₂]	400	ppm	Low estimate for 21 st century	T, L
dbh	Diameter at breast height	Varied	cm	Smith-Martin et al. (unpublished)	T, L
dbh2h1	DBH to height allometric constant	3.06		Smith-Martin et al. (unpublished)	T
frac.liana.al	Fraction of the total canopy occupied by the liana	Invading liana: 0.1; Mature liana: 0.4		Competition scenarios	L
frac.tree.al	Fraction of the total canopy occupied by the tree	Invading liana: 0.9; Mature liana: 0.6		Competition scenarios	T
Kmax	Maximum whole-plant hydraulic conductivity	Varied	mmol m ⁻¹ s ⁻¹ MPa ⁻¹	Meta-analysis	T, L
leaf.biom	Leaf biomass	$(1 / (\text{SLA}/S)) * \text{al}$	Kg		T, L
Lx	Initial stem length	18.2	M	DBH-height allometry	L
Lx_lost	Stem length lost due to turnover	$Lx * \text{stem turn}$	M		T, L

References (from reviewer 2):

Mencuccini, M., Rosas, T., Rowland, L., Choat, B., Cornelissen, H., Jansen, S., Kramer, K., Lapenis, A., Manzoni, S., Niinemets, Ü. and Reich, P.B., 2019. Leaf economics and plant hydraulics drive leaf: wood area ratios. *New Phytologist*, 224(4), pp.1544-1556.

Larjavaara M, and Muller-Landau HC (2010) PERSPECTIVE: Rethinking the value of high wood density. *Functional Ecology* 24:701-705

References:

- Coppieters, K. The impact of liana hydraulic properties on their functioning: A modeling study in Horizontes, Costa Rica. (Ghent University, Ghent, Belgium, 2021).
- Di Porcia e Brugnera, M. *et al.* Modeling the impact of liana infestation on the demography and carbon cycle of tropical forests. *Glob. Change Biol.* **25**, 3767–3780 (2019).
- Guzman, M. E. D. *et al.* Hydraulic traits of Neotropical canopy liana and tree species across a broad range of wood density: implications for predicting drought mortality with models. *Tree Physiol.* **41**, 24-34 (2020).
- Ichihashi, R. & Tatenno, M. Biomass allocation and long-term growth patterns of temperate lianas in comparison to trees. *New Phytol.* **207**, 604-612 (2015).
- Larjavaara, M. & Muller-Landau, H. C. Rethinking the value of high wood density. *Funct. Ecol.* **24**, 701-705 (2010).
- Medina-Vega, J. A., Bongers, F., Poorter, L., Schnitzer, S. A. & Sterck, F. J. Lianas have more acquisitive traits than trees in a dry but not in a wet forest. *J. Ecol.* **109**, 2367-2384 (2021).
- Mello, F. N. A., Estrada-Villegas, S., DeFilippis, D. M. & Schnitzer, S. A. Can functional traits explain plant coexistence? A case study with tropical lianas and trees. *Diversity* **12**, 397 (2020).
- Mencuccini, M. *et al.* Leaf economics and plant hydraulics drive leaf : wood area ratios. *New Phytol.* **224**, 1544-1556 (2019).
- Trugman, A. T., Anderegg, L. D. L., Shaw, J. D. & Anderegg, W. R. L. Trait velocities reveal that

mortality has driven widespread coordinated shifts in forest hydraulic trait composition.

Proc. Natl. Acad. Sci. U.S.A. **117**, 8532–8538 (2020).

Van der Sande, M. T., Poorter, L., Schnitzer, S. A. & Markesteijn, L. Are lianas more drought tolerant than trees? A test for the role of hydraulic architecture and other stem and leaf traits. *Oecologia* **172**, 961–972 (2013).

Zhu, S.-D., Chen, Y.-J. & Cao, K.-F. Different hydraulic traits of woody plants from tropical forests with contrasting soil water availability. *Tree Physiol.* **37**, 1469-1477 (2017).

Appendix 1: Model description

Below is the text added to the methods section of the manuscript describing the model we used in our simulations (section “Methods: Competition model”; lines 275-324).

We modified the single-tree model originally developed by Trugman et al.³⁶ to represent a single liana-tree pairing. The purpose of the original model developed by Trugman et al. is to calculate annual net primary production (A_{net}) of a single temperate tree under defined climatic conditions and morphological and physiological parameters, with A_{net} becoming the input to a subsequent model describing tree drought recovery. Briefly, the model couples water transport using the Shinozaki pipe model³⁸ and the Ball-Berry model of stomatal conductance³⁹ and whole-plant photosynthesis using the Farquhar photosynthesis model³⁷. The amount of water moving through the plant depends on soil water availability (soil water potential, Ψ); the hydraulic path length and xylem area of fine roots, stem, and petioles; and the water demand imposed on the tree by the atmosphere (vapor pressure deficit, VPD). Mathematically, this can be written with the following set of equations. First, the flow, F , throughout a plant element is computed by integrating the hydraulic conductivity per unit of xylem area (K) from one end of the pipe continuum with water potential ψ_1 to the other with water potential ψ_2 , which can be expressed by the differences in the Kirchhoff transforms as

$$F = \frac{a}{L} \int_{\psi_1}^{\psi_2} K(\Psi) d\Psi = \frac{a}{L} (\phi_2 - \phi_1) \quad (1)$$

where a is the xylem area of the element and L the pipe length. The element conductivity (K) decreases as stem water potential falls as a result of embolism. A logistic function is used to represent the loss of conductivity as water potential becomes more negative, and thus ϕ is proportional to the maximum hydraulic conductivity, K_{max} .

If we neglect changes in water storage, F is constant throughout the hydraulic continuum. Then, water flow from the roots to the stem is modeled as

$$F = \frac{a_{root}}{L_{root}} (\phi_{soil} - \phi_{root}) = \frac{a_{stem}}{L_{stem}} (\phi_{root} - \phi_{stem}) \quad (2)$$

where a_{root} and a_{stem} are the surface area of the tree roots and cross-sectional area of the xylem, respectively, L_{root} and L_{stem} are the path length from the soil to the base of the stem and the tree height, respectively, and ϕ_{soil} , ϕ_{root} , and ϕ_{stem} are the integral of the conductivity for the soil, roots and stem, respectively, calculated from the Kirchhoff transform. Flow from the stem to the leaves is modeled as

$$\frac{a_{stem}}{L_{stem}} (\phi_{root} - \phi_{stem}) = \frac{a_{petiole}}{L_{petiole}} (\phi_{stem} - \int_0^L \phi_{leaf}(l_a) \frac{dl_a}{L_a}) \quad (3)$$

where a_{petiole} is the cross-sectional xylem area within a given petiole summed over the tree, L_{petiole} is the length of the petiole, ϕ_{leaf} is the integral of the conductivity for the petiole, L_a is the leaf area index, l_a is the index of a given leaf layer, and dl_a/L_a represents the xylem area per unit leaf. Assuming there is only one leaf layer and all photosynthesis is carbon limited only, this equation simplifies to

$$\frac{a_{\text{stem}}}{L_{\text{stem}}}(\phi_{\text{root}} - \phi_{\text{stem}}) = \frac{a_{\text{petiole}}}{L_{\text{petiole}}}(\phi_{\text{stem}} - \phi_{\text{leaf}}) \quad (4)$$

Flow from the leaf to the atmosphere is modeled as

$$\frac{a_{\text{petiole}}}{L_{\text{petiole}}}(\phi_{\text{stem}} - \phi_{\text{leaf}}) = a_{\text{leaf}} g_s D \quad (5)$$

where a_{leaf} is leaf area, g_s is stomatal conductance, and D is VPD. Stomatal conductance, g_s , is modeled following ref. 62 as

$$g_s = A_n \frac{c_1}{(C_a - \Gamma)(1 + \frac{D}{D_0})} \beta(\Psi_{\text{leaf}}) \quad (6)$$

In this equation, C_a is the atmospheric CO_2 concentration; c_1 , D_0 , and Γ are empirical constants from the Leuning model⁶²; A_n is net photosynthesis; and Ψ_{leaf} is leaf water potential. The function $\beta(\Psi_{\text{leaf}})$ serves to down-regulate photosynthesis under water stressed conditions and is determined by the carbon cost of sustaining negative water potential and loss of conductivity in the xylem. For simplicity we assumed that $\beta(\Psi_{\text{leaf}})$ varies linearly with the Kirchhoff transform as

$$\beta(\Psi_{\text{leaf}}) = \frac{\phi_{\text{leaf}}}{\phi_{\text{max}}} \quad (7)$$

where ϕ_{max} is the integral of maximum hydraulic conductivity of the xylem. B varies between 1 (leaf at full hydration) and 0 (leaf under full water stress). Here, $\beta(\Psi_{\text{leaf}})$ broadly conforms to the solution of the Leuning model, but with a more mechanistic representation of soil moisture stress through soil water potential's effect on leaf water potential.

The method of solution is the same as in Trugman et al.³⁶. In this way, computation of A_{net} is related to three climatic variables (Ψ , VPD, and CO_2 concentration), dimensions of the water conducting tissue of the tree, and tree physiological parameters.

We modified the Trugman et al. model to include a tree-liana pair and to improve the realism of the relationship between climate and plant water flow. In contrast to the use of this model for computing A_{net} as in Trugman et al., we use the model to define K_{req} , the required

whole-plant hydraulic conductivity, by iteratively finding the minimum K_{\max} (Equation 4) to yield a positive A_{net} on an annual timestep (Methods: Simulations).

Appendix 2: geographic and climatic correlations with hydraulic traits

Results of our analysis of variation in hydraulic traits as a function of geographic location and climate, as recommended by Reviewer 2, is below. We have chosen to present this analysis as a series of figures with tree and liana observations combined. Adjusted R-squared for lianas and trees combined is shown on the plot. Adjusted R-squared for each growth form separately is listed below each figure.

Appendix Figure 1: simple linear regression between latitude and K_s . Liana $R_{adj}^2 = -0.02$; tree $R_{adj}^2 = 0.06$.

Appendix Figure 2: simple linear regression between longitude and K_s . Liana $R^2_{adj} = 0.02$; tree $R^2_{adj} = 0.12$.

Appendix Figure 3: simple linear regression between altitude and K_s . Liana $R^2_{adj} \cong 0$; tree $R^2_{adj} = 0.15$.

Appendix Figure 4: simple linear regression between dry season length and K_s . Liana $R^2_{adj} = -0.02$; tree $R^2_{adj} = 0.03$.

Appendix Figure 5: simple linear regression and ANOVA between the season during which measurements were conducted and K_s . “Both” indicates that reported values were an average of measurements made during the wet and dry seasons. ANOVA tests for trees and lianas individually were also significant.

Appendix Figure 6: simple linear regression between latitude and P_{50} . Note that despite the appearance that only two latitudes were considered, multiple values are present in each grouping around 10° and 20°. Liana $R^2_{adj} = 0.19$; tree $R^2_{adj} = -0.01$.

Appendix Figure 7: simple linear regression between longitude and P₅₀. Similar to latitude, multiple longitudes are present in each of the major groupings. Liana $R^2_{adj} = 0.20$; tree $R^2_{adj} = -0.01$.

Appendix Figure 8: simple linear regression between altitude and P_{50} . Liana $R^2_{adj} = 0.26$, tree $R^2_{adj} = -0.01$.

Appendix Figure 9: simple linear regression between dry season length and P_{50} . Liana $R^2_{adj} = 0.23$; tree $R^2_{adj} = -0.01$.

Appendix Figure 10: t-test between season during which measurements were taken and P₅₀. Liana p < 0.01, tree p > 0.1.

Reviewer comments, second round

Reviewer #2 (Remarks to the Author):

Re-review of "Climate and hydraulic traits interact to set thresholds for liana viability"

Summary

The authors have done a great job addressing the comments and I'm encouraged that they used this review process to make such demonstrable improvements to an already well-thought and compelling analysis. Most notably, the essential elements of the model are, with two key exceptions (see main concerns below) sufficiently described. With a lot more clarity about the model, I still have some significant concerns. I still think this is a great analysis and worthy of publication as long as these can be addressed.

2 main concerns

Units and definitions: First, I have a comment about whether whole-plant conductivity is even a useful or valid concept (see General Comments below). Second, while the authors have defined K_{req} , nowhere do they explicitly define whole-plant conductivity, K_w . The closest I can find is L292-293 where " ϕ is proportional to K_{max} " and later (L322-323) that K_{req} is found by iteratively finding the minimum K_{max} . So it seems that they are defining K_w either in terms of the flux potential ϕ (perhaps divided by Area times length) or in terms of K_{max} . But neither of these are 'whole-plant conductivity'. Third, if what the authors are reporting for K_{req} is the former, then this is really a flux potential in Figures 2A,B, 3B,C, and 4. This is because the kirchoff transform integrates over water potential (MPa no longer in the denominator), not a flux per unit water potential, which would be some form of conductance or conductivity. Finally, at least in the model, there is no "uncertainty with scaling between K_w and K_s " (L438-439) – this is fully defined.

Functional form of the $K(Y)$ curve. First of all, the authors refer to Table S2 of Trugman et al. (2018) to view the form of the logistic curve employed, but Table S2 only reports the definite integral for the whole $K(Y)$ curve, which is the max kirchoff value (ϕ_{max}). I had to go to code https://github.com/amwillson/liana-tree-comp/blob/master/NPP_models.R#L79 to see what the indefinite integral is, which is the form needed to estimate ϕ at a given water potential. That equation, when differentiated with respect to ψ , does not appear to have the desired properties of a $K(Y)$ curve at full saturation ($Y = 0$); in other words, $K(Y=0)$ does not appear to equal K_{max} but rather $K_{max}/\exp(b_1*b_2)$. I checked quickly, so could be making a mistake; this should be checked.

General Comments on Interpretation

I don't think it makes much sense to talk about whole-plant conductivity, only whole plant conductance. Conductivity only makes sense when considering any plant segment that can be approximated as a pipe (easily defined cross-sectional area and length). Conductance accounts for all of the specific geometries encountered when integrating over the organism. That said, the authors' emphasis on a whole-plant value is good, because it implies that any future trait filtering under drier climates will ultimately select based on a whole-plant value, not just on branch-specific conductivity.

But the authors do not recognize (or at least do not emphasize) this in the manuscript. While filtering may occur based on tissue-specific conductivity, all of the other mechanisms contributing to the whole-plant value may be selected on as well (Huber value, external and internal branching structures/allometry, root structures and fungal mutualisms). Considering huber value for instance, has implications for interpretation and discussion. It could be that in the future, rather than lianas not being viable in a strict climate envelope-trait filtering sense, they simply are less leafy (to maintain K_{req}) via phenotypic plasticity. This in turn reduces their competitive advantage with trees, and eventually leads to their demise as opposed to a catastrophic threshold/tipping point (e.g., as suggested in lines 180-182). In light of these considerations, a limited view that implies selection will act on K_s alone (cf lines 66-68, 165, 184-185) is, in my view, not warranted. On the other hand, there are other places in the text where explanations are sufficiently broad (e.g., Lines 192-193).

L118: should highlight here the Huber value too

L184-5: and H_v . The two go hand-in hand in my view. Alternatively, if you want to highlight the

whole plant conductance (not conductivity), that incorporates both traits.

L186: I'm sure path length plays a role, but more relevant based on your model and analysis is that K_{req} depends on K_s , HV, and VPD.

L249: It would be useful to know the end result of combining TRY with the extended meta-analysis: how many species total, of which how many are lianas?

p7 Suppl: A final meta-analysis sensitivity analysis I think would settle most concerns about potentially co-mingling geographic disparity and liana vs tree differences. Can you simply repeat your Mann-Whitney tests and Glass effect size estimates for K_s excluding trees that fall outside of the geographic range of the 51 liana species that you ended up with? Your geographic/climatic variable analysis was done on lianas and trees separately (and yes, it adds confidence to your conclusions), but that doesn't necessarily rule out the possibility that disparate geographic ranges of trees and lianas could partially explain the difference. I'm not necessarily requiring it (i.e., it may not be possible if there are substantial gaps in tree K_s data collocated with the liana K_s data), but it's just another way to add confidence a result that is so central to the message of this paper.

L284ff: State the units of your variables (included for some but not all)

L296ff: Please re-read / check. Some things aren't quite right. How can this be flow from soil to the stem base but include tree height as a path length?

L325-335: How is total leaf area represented? Supp Table 4 says $total$ is 200 but says it is calculated as $100 m^2 * 2 m^2 m^{-2}$ which to me implies that $total$ is not canopy area but rather total leaf area with canopy area = $100 m^2$ and LAI of $2 m^2 m^{-2}$. Is this correct? It seems canopy area (L28 and elsewhere) is confused with total leaf area as this is quite confusing for readers.

L342-343: Rephrase. I think you're using an allometric relationship between DBH and sapwood area but then keeping leaf area fixed. It sounds like you have an allometric relationship between DBH and leaf area.

L376: Kw units

L376-8: I still don't understand what Kw is and how it's different from K_s - 'stem-specific hydraulic conductivity (units $mol m^{-1} m^{-2} MPa^{-1}$). Similarly, how do you use the measurements of K_s to constrain Kw?

Lin 440: You still have not defined Kw?

L338: Liana DBH

L343-346: Why is the minimum (2 cm) and mean (2.65 cm) DBH for lianas so similar? Is the size distribution really that right-skewed? I

Supp Table 4: What is difference between b_2Ht and dbh_2h_1 ? Should one of these be biomass as in Trugman et al 2018? I am searching to understand how leaf area is treated.

Figures

Figure 2: The way this figure is presented at least for me requires a fair bit of time to digest. I think two main sources of confusion are: 1) the x-axis is strictly reserved for variation in lianas but not trees, but the axis title does not indicate this and 2) one has to read the figure caption to understand that solid lines are lianas and dashed lines / Xs are trees. It may improve clarity to restructure the presentation in such a way as to make it seem the main messages of this figure are: 1) Liana k_{req} increases with increasing leafiness (lower HV) and with a drier climate, and 2) Liana k_{req} exceeds tree k_{req} by many factors over the majority of simulations, given the observed size distributions. Given that invasion scenario is also tantamount to variation in liana HV, how critical is it to have the different invasion scenarios represented? It may be worth re-conceiving a figure that best conveys these messages while eliminating redundancy.

Figure 3B: Need units in the legend for K_{req} , and shouldn't it be $\log(K_{req})$?

I could not find the new Suppl Fig S15 (sensitivity to future adjustment in P50) referred to in the response to reviewer #1.

In the following response to reviewers, the reviewers' comments are *italicized*. We add **our responses and explanations** below each comment in **blue**. We have copied and pasted **short new sections from the manuscript** in **green** offset by quotations. For longer revisions, we include reference numbers and refer the reader to the Appendices 1 & 2 at the end of this response. Changes in the **manuscript** are indicated **by purple text**. Changes to the Supplementary Discussion, Figures, and Tables are included in the response and their location is given by section title but not by line numbers or purple text because this formatting has been removed for final submission of this manuscript.

REVIEWER COMMENTS

reviewer #2 (Remarks to the Author):

Re-review of "Climate and hydraulic traits interact to set thresholds for liana viability"
Nature Communications

Summary

The authors have done a great job addressing the comments and I'm encouraged that they used this review process to make such demonstrable improvements to an already well-thought and compelling analysis. Most notably, the essential elements of the model are, with two key exceptions (see main concerns below) sufficiently described. With a lot more clarity about the model, I still have some significant concerns. I still think this is a great analysis and worthy of publication as long as these can be addressed.

We thank the reviewer for their positive feedback and recognizing the improvements made to the original manuscript.

2 main concerns

Units and definitions: First, I have a comment about whether whole-plant conductivity is even a useful or valid concept (see General Comments below).

Our conceptualization of whole-plant conductivity is derived from the "maximum hydraulic conductivity" term (K_{\max}) in our photosynthesis model. We have chosen to use the term "whole-plant hydraulic conductivity" to emphasize that this term is not specific to any single plant organ (e.g., branch hydraulic conductivity). We believe that maximum conductivity, and by association whole-plant conductivity, is a useful concept because this term is frequently a parameter in dynamic vegetation models. For example, one of the hydraulic parameters of Ecosystem Demography 2 is maximum hydraulic conductivity and this parameter, like ours, is not specific to a single plant organ (Xu et al. 2016). Reporting on a whole-plant hydraulic conductivity is therefore consistent with our manuscript's purpose to improve liana representation in dynamic vegetation models. We hope this explanation clarifies our intention for the reviewer; we address more specific concerns from the reviewer below.

To improve clarity of the term K_w (and K_{req}), we have re-defined the terms throughout the paper as “maximum whole plant hydraulic conductivity” (K_w) and “required maximum whole plant hydraulic conductivity” (K_{req}) throughout the manuscript.

Second, while the authors have defined K_{req} , nowhere do they explicitly define whole-plant conductivity, K_w . The closest I can find is L292-293 where “ ϕ is proportional to K_{max} ” and later (L322-323) that K_{req} is found by iteratively finding the minimum K_{max} . So it seems that they are defining K_w either in terms of the flux potential ϕ (perhaps divided by Area times length) or in terms of K_{max} . But neither of these are ‘whole-plant conductivity’.

We appreciate that the reviewer has pointed out that we did not sufficiently define whole plant conductivity (K_w) and we have incorporated a more direct definition of this term in the manuscript (Methods: Competition Model; lines 351-356) and we also added a Supplementary Table of the definitions of the conductivity terms (K_s , K_w , and K_{req}) to the manuscript to assist the reader in differentiating between these terms (Supplementary Table X). K_w is the maximum hydraulic conductivity (K_{max}), which is used to model water flow through the plant and the effects of water stress on photosynthesis under water stressed conditions. The same function, K , is used for all plant organs. Therefore, K_{max} is the maximum whole-plant hydraulic conductivity. We have chosen to designate this value as whole-plant conductivity (K_w) because this emphasizes that the model parameter is not specific to a given plant organ, as is the case with stem-specific branch hydraulic conductivity, K_s , which is more commonly measured in the field (Pérez-Harguindeguy et al. 2016).

Finally, upon reading the reviewer’s comments, we have noticed and corrected an error in our language describing the model: ϕ is *not* proportional to K_{max} , but rather ϕ is a function of K_{max} . This change is reflected in the manuscript (Methods: Competition Model; line 312).

- **“Competition model” (Methods):** We modified the Trugman et al. model to include a tree-liana pair and to improve the realism of the relationship between climate and plant water flow. In contrast to the use of this model for computing A_{net} as in Trugman et al., we use the model to define K_{req} , the required **maximum** whole-plant hydraulic conductivity, by iteratively finding the minimum K_{max} (Equation 4) to yield a positive A_{net} on an annual timestep (Methods: Simulations). **To emphasize the independence of the maximum hydraulic conductivity in the model (K_{max}) from plant branch-level measurements and differentiate this term in the model from K_s (observed branch hydraulic conductivity), we designate this term maximum whole-plant hydraulic conductivity (K_w) hereafter.** The hydraulic conductivity variables we consider in this manuscript (K_s , K_w , and K_{req}) are defined in Supplementary Table 7.
- **“Competition model” (Methods):** A logistic function is used to represent the loss of conductivity as water potential becomes more negative, and thus ϕ ($\text{mmol m}^{-1} \text{s}^{-1}$) is a **function** of the maximum hydraulic conductivity, K_{max} .

A copy of Supplementary Table 7 is available below.

Parameter	Definition	Observed or modeled
K_s	Stem-specific hydraulic conductivity. Measured on terminal branches.	Observed
K_w	Maximum whole-plant specific hydraulic conductivity. Equivalent to model parameter K_{max} . Does not apply to a specific plant organ.	Modeled
K_{req}	Required maximum whole-plant hydraulic conductivity. The K_w required to maintain positive annual net primary production.	Modeled

Supplementary Table 7: Definitions of the three hydraulic conductivity terms used throughout the manuscript. The “Parameter” column indicates to which term the row pertains. The “Definition” column provides a definition and description of the term. The “Observed or Modeled” column indicates whether the term applies to quantities that are observed (i.e., come from measurement) or are modeled (i.e., model parameters).

Third, if what the authors are reporting for K_{req} is the former, then this is really a flux potential in Figures 2A,B, 3B,C, and 4. This is because the kirchoff transform integrates over water potential (MPa no longer in the denominator), not a flux per unit water potential, which would be some form of conductance or conductivity.

As explained above, whole-plant hydraulic conductivity (K_w) is defined as the maximum hydraulic conductivity parameter (K_{max}) in our photosynthesis model. Water potential is not integrated out of K_{max} . We have added an additional equation to the model description in the methods to help the reader track the use of K_w in the model and show its independence from the Kirchoff transform (Methods: Competition Model; lines 337-341). The method by which we define K_{req} , defining the minimum maximum hydraulic conductivity that yields a positive A_{net} (the output of the model) is described in the methods (Methods: Simulations; lines 463-468) and may be useful for understanding the definition of K_{req} .

- **“Competition model” (Methods):** The function $\beta(\psi_{leaf})$ serves to down-regulate photosynthesis under water stressed conditions and is determined by the carbon cost of sustaining negative water potential and loss of conductivity in the xylem. For simplicity we assumed that $\beta(\psi_{leaf})$ varies linearly with the Kirchoff transform as

$$\beta(\psi_{leaf}) = \frac{\phi_{leaf}}{\phi_{max}}$$

where ϕ_{max} is the integral of maximum hydraulic conductivity of the xylem. $\beta(\psi_{leaf})$ varies between 1 (leaf at full hydration) and 0 (leaf under full water stress). The denominator ϕ_{max} is defined in terms of the maximum hydraulic conductivity (K_{max}) as follows:

$$\phi_{max} = \frac{K_{max} * \log(\exp(-b1*b2) + 1)}{b1}$$

where K_{max} is equivalent to the maximum whole-plant hydraulic conductivity (K_w , $\text{mmol m}^{-1} \text{s}^{-1} \text{MPa}^{-1}$) and $b1$ (% MPa^{-1}) and $b2$ (MPa) are the slope of the percent loss of the conductivity curve and the pressure at which 50% of xylem function is lost, respectively.

Finally, at least in the model, there is no “uncertainty with scaling between K_w and K_s ” (L438-439) – this is fully defined.

While there is a relationship between K_w and K_s defined in our photosynthesis model because of the scaling between the whole plant and each individual organ, we argue that this scaling is an assumption of the model. Without sufficient data on this scaling, we must make an assumption, but we argue that more data on the scaling would benefit the modeling community by providing a more empirically driven relationship between these two important variables.

*Functional form of the $K(Y)$ curve. First of all, the authors refer to Table S2 of Trugman et al. (2018) to view the form of the logistic curve employed, but Table S2 only reports the definite integral for the whole $K(Y)$ curve, which is the max kirchoff value (ϕ_{max}). I had to go to code https://github.com/amwillson/liana-tree-comp/blob/master/NPP_models.R#L79 to see what the indefinite integral is, which is the form needed to estimate ϕ at a given water potential. That equation, when differentiated with respect to ψ , does not appear to have the desired properties of a $K(Y)$ curve at full saturation ($Y = 0$); in other words, $K(Y=0)$ does not appear to equal K_{max} but rather $K_{max}/\exp(b1*b2)$. I checked quickly, so could be making a mistake; this should be checked.*

Thank you for your attention to our code. The reviewer can integrate the following logistic function from $-\infty$ to ψ_{soil}

$$\frac{K_{max} * \exp(b1*(\psi_{soil} - b2))}{\exp(b1*(\psi_{soil} - b2)) + 1}$$

Evaluating this function at $\psi_{soil} = 0$, we find

$$\frac{K_{max} * \exp(-b1*b2)}{\exp(-b1*b2) + 1}$$

The maximum realized model K is not exactly K_{max} but the difference is less than 2% for the model's default values of $b1$ and $b2$ (2 and -2), which we regard as a tolerable deviation.

General Comments on Interpretation

I don't think it makes much sense to talk about whole-plant conductivity, only whole plant conductance. Conductivity only makes sense when considering any plant segment that can be approximated as a pipe (easily defined cross-sectional area and length). Conductance accounts for all of the specific geometries encountered when integrating over the organism.

We have chosen to consider whole-plant conductivity, as it relates to maximum conductivity, because this parameter is frequently used in dynamic vegetation models, as discussed above. Second, in the context of our photosynthesis model, the geometries of the organism are additionally easily defined. This is because the path length of each organ (stem, fine roots, and petioles) is defined. The stem length is defined either allometrically from diameter at breast height (tree) or as an input to the model (liana), while fine root and petiole length are inputs to the model. Total functional xylem area is also defined as a function of diameter at breast height.

That said, the authors' emphasis on a whole-plant value is good, because it implies that any future trait filtering under drier climates will ultimately select based on a whole-plant value, not just on branch-specific conductivity. But the authors do not recognize (or at least do not emphasize) this in the manuscript. While filtering may occur based on tissue-specific conductivity, all of the other mechanisms contributing to the whole-plant value may be selected on as well (Huber value, external and internal branching structures/allometry, root structures and fungal mutualisms). Considering huber value for instance, has implications for interpretation and discussion. It could be that in the future, rather than lianas not being viable in a strict climate envelope-trait filtering sense, they simply are less leafy (to maintain K_{req}) via phenotypic plasticity. This in turn reduces their competitive advantage with trees, and eventually leads to their demise as opposed to a catastrophic threshold/tipping point (e.g., as suggested in lines 180-182). In light of these considerations, a limited view that implies selection will act on K_s alone (cf lines 66-68, 165, 184-185) is, in my view, not warranted. On the other hand, there are other places in the text where explanations are sufficiently broad (e.g., Lines 192-193).

We certainly agree that alternative mechanisms by which lianas could be affected by changing hydroclimate are possible. We have thus taken this opportunity to reconsider how we have presented our prediction of future liana mortality. We have added a paragraph that discusses a competition-based alternative to the threshold-like scenario that we have emphasized thus far to the discussion in the main text (Main text: Hydraulic trait-climate interactions; lines 185-199).

Additionally, we have made some minor modifications to the main text to accommodate this shift in focus. First, we added the phrase “on average” to our summary statement that liana K_{req} is greater than tree K_{req} (Main text: Hydraulic trait-climate interactions; line 201). Second, we modified the end of our discussion, which summarizes our findings, to include alternative mechanisms of liana mortality under changing climate (Main text: Hydraulic trait-climate interactions; lines 212-214).

These additions to the main text complement our discussion of the potential for liana physiological adaptation under a changing climate in the Supplement. In the section “Model results: Future hydroclimate” of the Supplementary Discussion, we discuss the greater sensitivity of larger lianas to drying hydroclimate than smaller lianas, potentially shifting the size distribution of lianas to smaller individuals in the future. In the section “Model results: Physiological adaptation under future hydroclimate” of the Supplementary Discussion, we

discuss the possibility of adaptation of other hydraulic traits to drying hydroclimate, thus maintaining a similar K_{req} to what we estimate for the present day.

Additionally, we emphasize that our main conclusions, that liana K_{req} is greater than tree K_{req} and is more sensitive to drying hydroclimate than tree K_{req} , are robust to changes in allometry, including Huber value. According to our sensitivity analysis, liana K_{req} remains generally greater than tree K_{req} over different competition scenarios (where different fractions of total leaf area are apportioned to each growth form) and over different scenarios of total leaf area, both of which affect Huber value (e.g. Supplementary Figure 11). Second, the absolute magnitude of change in liana K_{req} as hydroclimate dries is consistently greater than the change in tree K_{req} over the same change in hydroclimate, regardless of total leaf area or competition scenario (e.g., Supplementary Figures 14 & 15). While we do not argue that other mechanisms of liana mortality as a result of drying hydroclimate are possible in the future, we believe our core argument, that hydraulic conductivity is an important parameter differentiating trees and lianas in tropical forests, is robust to these possibilities.

- **“Hydraulic trait-climate interactions” (Main text):** Thus far, we have focused on the scenario of a threshold-like response of lianas to drying hydroclimate; that is, when K_{req} surpasses realized maximum whole-plant hydraulic conductivity, lianas will be unable to maintain a positive annual carbon balance, leading to higher mortality rates. More gradual mechanisms may also lead to increased liana mortality under a drier hydroclimate. For instance, physiological adaptations leading to a greater Huber value among lianas may decrease their competitive advantage with trees, thus leading to a more gradual decline in liana viability via greater competition with trees⁴⁸. Such physiological adaptations could include a reduction in total leaf area to reduce water loss via transpiration or an increase in allocation to woody tissues to increase water storage. Alternatively, drought deciduousness among lianas could become more prevalent under drier conditions⁸. All of these adaptations would allow lianas to maintain a similar K_{req} to that realized today, but would reduce net photosynthesis⁴⁸. Nevertheless, our conclusions indicate that lianas are more susceptible than trees to drying hydroclimate and may experience higher mortality, whether via a threshold-like effect of increased K_{req} or via a decrease in net photosynthesis in response to physiological adaptation to greater K_{req} .
- **“Hydraulic trait-climate interactions” (Main text):** In this study, we identified hydraulic conductivity as a critical trait distinguishing lianas from trees, with lianas **on average** having a greater K_{req} .
- **“Hydraulic trait-climate interactions” (Main text):** We suggest that a climate threshold exists over which lianas will be unable to survive given the sensitivity of their hydraulic architecture to hydroclimate. If atmospheric VPD increases as projected by climate models, recent increases in liana abundance in the Americas¹⁴⁻¹⁶ may be shortlived, with long-term consequences for forest community dynamics⁵¹, C storage capacity^{1,2,52}, and the economic value of tropical forests^{20,21,53}. **Even if a climate threshold for liana viability is not realized, lianas may sustain significant reductions in population size via increased competition-driven mortality.** In order to improve forecasts of these processes under

climate change, dynamic vegetation models should include lianas parameterized with their distinguishing hydraulic traits.

L118: should highlight here the Huber value too

We thank the reviewer for this suggestion. We have added text to imply that the dependence of K_{req} on diameter at breast height is a result of the change in Huber value at constant leaf area (Main text: Hydraulic traits influence viability; lines 117-119). This text was also changed to support our interpretation of the new format for Figure 2.

- **“Hydraulic traits influence viability” (Main text):** We find that liana K_{req} is greater at lower diameters when total leaf area is constant and at lower Huber value (Figure 2a-b) because the xylem supplies relatively more leaves with water under these conditions.

L184-5: and Hv. The two go hand-in hand in my view. Alternatively, if you want to highlight the whole plant conductance (not conductivity), that incorporates both traits.

We appreciate that the reviewer has encouraged us to emphasize the interdependence of hydraulic conductivity and Huber value in our modeling experiments. We have modified the second sentence of this paragraph to address this comment and the one below by explaining the difference in K_{req} between lianas and trees as a function of Huber value (not path length) and VPD (Main text: Hydraulic trait-climate interactions; line 202).

- **“Hydraulic trait-climate interactions” (Main text):** In this study, we identified hydraulic conductivity as a critical trait distinguishing lianas from trees, with lianas having a greater K_{req} . The difference in K_{req} between lianas and trees is sensitive to Huber value and to VPD.

L186: I'm sure path length plays a role, but more relevant based on your model and analysis is that K_{req} depends on K_s , HV , and VPD.

We agree with the reviewer that the phrasing of this sentence was inappropriate. We have modified the sentence to state that the difference in K_{req} between lianas and trees is a result of differences in Huber value, not hydraulic path length (Main text: Hydraulic trait-climate interactions; line 202).

- **“Hydraulic trait-climate interactions” (Main text):** In this study, we identified hydraulic conductivity as a critical trait distinguishing lianas from trees, with lianas having a greater K_{req} . The difference in K_{req} between lianas and trees is sensitive to Huber value and to VPD.

L249: It would be useful to know the end result of combining TRY with the extended meta-analysis: how many species total, of which how many are lianas?

We thank the reviewer for this suggestion. We added the total number of species, number of tree species, and number of lianas species to this paragraph (Methods: Extended meta-analysis; lines 267-268).

- **“Extended meta-analysis” (Methods):** We applied the same criteria to the observations in the TRY database, combined the observations from TRY and from our additional literature search, and averaged the observations to the species level. This resulted in a total of 154 species with hydraulic trait observations matching our criteria, of which 51 species were lianas and 103 species were trees.

p7 Suppl: A final meta-analysis sensitivity analysis I think would settle most concerns about potentially co-mingling geographic disparity and liana vs tree differences. Can you simply repeat your Mann-Whitney tests and Glass effect size estimates for K_s excluding trees that fall outside of the geographic range of the 51 liana species that you ended up with? Your geographic/climatic variable analysis was done on lianas and trees separately (and yes, it adds confidence to your conclusions), but that doesn't necessarily rule out the possibility that disparate geographic ranges of trees and lianas could partially explain the difference. I'm not necessarily requiring it (i.e., it may not be possible if there are substantial gaps in tree K_s data collocated with the liana K_s data), but it's just another way to add confidence a result that is so central to the message of this paper.

We appreciate the reviewer's concern about the sensitivity of the conclusions we draw from our meta-analysis to geography. Upon conducting further analysis with a subset of our data, we conclude that our conclusions are robust to the specific geographic extent of liana and tree growth forms considered. We subset our full meta-analysis dataset to include only publications that reported hydraulic trait measurements for co-located trees and lianas. Publications only reporting hydraulic trait measurements for lianas or trees were excluded. We then re-computed the Mann-Whitney and Glass' Delta tests with this subset of data. We find that the conclusions do not change: K_s is significantly different between lianas and trees, while P_{50} and slope of the PLC curve remain statistically non-significant. Tables in the same format as those reported in the Supplement to our manuscript are below, but the tables found below are not included in the manuscript. We have instead added a statement to the supplement with our conclusions from this additional analysis (Supplementary Discussion: Extended meta-analysis).

- **“Extended meta-analysis” (Supplementary Discussion):** For the hydraulic traits considered in our extended meta-analysis (i.e., K_s and P_{50}), we conducted simple linear regressions and t-tests with various geographic (latitude, longitude, altitude) and climatic (dry season length, season during which measurements were made) variables extracted from the meta-data of the literature we compiled to address this concern. We found that no geographic or climatic variable strongly correlated with tree and liana observations combined and none of our variables of interest explained more than 15% of variation in K_s (R^2_{adj} of tree K_s with altitude = 0.15; not shown) and 26% of variation in P_{50} (R^2_{adj} of liana P_{50} with altitude = 0.26; not shown) when tree and liana observations were considered separately. We additionally conducted Mann-Whitney U-tests and computed

Glass' Δ for a subset of the data that included only publications publishing hydraulic trait observations for collocated trees and lianas (i.e., all publications reporting observations for only trees or lianas were removed). The subset included a total of 65 tree species and 49 liana species. The results of both the Mann-Whitney U-test and the Glass' Δ indicate that K_s is significantly different between trees and lianas (Glass' $\Delta = 2.29$, MannWhitney test statistic = 1,055, $n_{tree} = 65$, $n_{liana} = 49$, $p < 0.01$), while P_{50} and Slope remain non-significant (P_{50} : Glass' $\Delta = 0.323$, Mann-Whitney test statistic = 980.0, $n_{tree} = 60$, $n_{liana} = 39$, $p > 0.05$; Slope: Glass' $\Delta = 0.778$, Mann-Whitney test statistic = 33.0, $n_{tree} = 13$, $n_{liana} = 8$, $p > 0.05$). We do not report the results of these supplementary analyses here, but the data and code used to analyze these data are available in our Github repository.

Below are the tables of the Mann-Whitney and Glass' Δ tests for the subset of data.

Mann-Whitney U Tests for extended meta-analysis						
Trait	μ_{tree}	μ_{liana}	n_{tree}	n_{liana}	Test Statistic	p-value
Stem-specific hydraulic conductivity (mol/m/s/MPa)	322.16	946.44	65	49	1,054	2.08×10^{-3}
P_{50} (MPa)	-2.03	-1.74	60	39	980	1.75×10^{-1}
Slope of PLC curve (%/MPa)	1.52	2.23	13	8	33	1.85×10^{-1}

Effect size for extended meta-analysis			
Trait	Glass' Δ	Lower CI	Upper CI
Stem-specific hydraulic conductivity (mol/m/s/MPa)	2.29	0.94	3.62
P_{50} (MPa)	0.32	-0.03	0.67
Slope of PLC curve (%/MPa)	0.78	-0.35	1.87

L284ff: State the units of your variables (included for some but not all)

We thank the reviewer for addressing this omission. We have updated the model description to include units for each variable (Methods: Competition model; lines 305-340). The updated text is available as Appendix 1 at the end of this document.

L296ff: Please re-read / check. Some things aren't quite right. How can this be flow from soil to the stem base but include tree height as a path length?

The equation to which the reviewer is referring has flow (F) equal to both the flow from the soil to the roots and flow from the roots to the stem to show the equivalence of these terms. To ease the interpretation of our model, we have split equation 2 into 2 equations (Methods: Competition model; lines 315-320).

- **“Competition model” (Methods):** Then, water flow from the roots to the stem is modeled as

$$F = \frac{a_{root}}{L_{root}} (\phi_{soil} - \phi_{root}) \quad (2)$$

where a_{root} (m^2) is the surface area of the tree roots, L_{root} (m) is the path length from the soil to the base of the stem, and ϕ_{soil} ($mmol\ m^{-1}\ s^{-1}$) and ϕ_{root} ($mmol\ m^{-1}\ s^{-1}$) are the integral of the conductivity for the soil and roots, respectively, calculated from the Kirchhoff transform. Flow from the roots to the stem is modeled as

$$\frac{a_{root}}{L_{root}} (\phi_{soil} - \phi_{root}) = \frac{a_{stem}}{L_{stem}} (\phi_{root} - \phi_{stem}) \quad (3)$$

where a_{stem} (m^2) is the cross-sectional area of the xylem, L_{stem} (m) is the tree height, and ϕ_{stem} ($mmol\ m^{-1}\ s^{-1}$) is the integral of the conductivity for the stem.

*L325-335: How is total leaf area represented? Supp Table 4 says tot.al is 200 but says it is calculated as 100 m2 * 2 m2 m-2 which to me implies that tot.al is not canopy area but rather total leaf area with canopy area = 100 m2 and LAI of 2 m2 m-2. Is this correct? It seems canopy area (L28 and elsewhere) is confused with total leaf area as this is quite confusing for readers.*

Total leaf area and canopy area are equivalent in our photosynthesis model. This is because we assume all leaves are in one leaf layer (line 324). To aid readers in understanding the assumptions of our model, we have made the language throughout the text more consistent by replacing the term “canopy area” with the term “total leaf area.”

L342-343: Rephrase. I think you’re using an allometric relationship between DBH and sapwood area but then keeping leaf area fixed. It sounds like you have an allometric relationship between DBH and leaf area.

We thank the reviewer for pointing out that this wording is not intuitive. We have changed the sentence as is shown below to improve reader comprehension (Methods: Competition model; lines 372-376).

- **“Competition model” (Methods):** In Figure 2, we investigate the simultaneous effects of allometry (i.e., Huber value) and hydroclimate on K_{req} . In this figure, we defined the total leaf area shared by the tree and the liana ($200\ m^2$) and allowed liana DBH to vary between the minimum and maximum liana DBH (1.86 and 10.7 cm, respectively) observed during a field survey in Guanacaste, Costa Rica. We then computed Huber value by dividing the sapwood area (a function of DBH) by the total leaf area apportioned to each growth form.

L376: Kw units

We thank the reviewer for pointing out this omission. The units for K_w , as well as the other parameters mentioned, have been added to the text (Methods: Model parameterization; lines 409-410).

- “Model parameterization” (Methods): The only model inputs that differed between the tree and liana growth forms were **maximum** whole-plant stem-specific hydraulic conductivity (K_w , $\text{mmol m}^{-1} \text{s}^{-1} \text{MPa}^{-1}$), DBH (cm), leaf area (m^2), turnover ($\% \text{ year}^{-1}$), and initial stem length (m) (Supplementary Table 4).

L376-8: I still don't understand what K_w is and how it's different from K_s - 'stem-specific hydraulic conductivity (units $\text{mol m}^{-1} \text{m}^{-2} \text{MPa}^{-1}$). Similarly, how do you use the measurements of K_s to constrain K_w ?

K_w (maximum whole-plant hydraulic conductivity) is a model parameter that is not specific to any plant organ. K_s (stem-specific hydraulic conductivity) is an observed quantity that is measured on terminal branches. We hope the text added to define K_w in the model description (Methods: Competition model; lines 351-356) and the new Supplementary Table 7 make the difference between these conductivities more clear. The measurements of K_s no longer constrain the possible values of the model input parameter K_w due to the uncertainty in the scaling between these terms. The boundaries of the possible values of K_{req} are $(\min(K_s)/10, \max(K_s))$; however, these boundaries were chosen because they represent the range of possible K_{req} values derived from our model (i.e., a smaller K_{req} would not be observed if the boundary conditions were decreased and a larger K_{req} would not be observed if the boundary conditions were increased). To avoid confusion about the relationship between K_w and K_s , we have removed two pertinent phrases from the text (Methods: Model parameterization; line 410 and Methods: Simulations; lines 465-466).

- **“Competition model” (Methods):** We modified the Trugman et al. model to include a tree-liana pair and to improve the realism of the relationship between climate and plant water flow. In contrast to the use of this model for computing A_{net} as in Trugman et al., we invert the model to define K_{req} , the required **maximum** whole-plant hydraulic conductivity, by iteratively finding the minimum K_{max} (Equation 4) to yield a positive A_{net} on an annual timestep (Methods: Simulations). **To emphasize the independence of the maximum hydraulic conductivity (K_{max}) from plant branch-level measurements and differentiate this term in the model from K_s (observed branch hydraulic conductivity), we designate this term maximum whole-plant hydraulic conductivity (K_w) hereafter. The hydraulic conductivity variables we consider in this manuscript (K_s , K_w , and K_{req}) are defined in Supplementary Table 7.**

A copy of Supplementary Table 7 is available below.

Parameter	Definition	Observed or modeled
K_s	Stem-specific hydraulic	Observed

	conductivity . Measured on terminal branches.	
K_w	Maximum whole-plant specific hydraulic conductivity . Equivalent to model parameter K_{max} . Does not apply to a specific plant organ.	Modeled
K_{req}	Required maximum whole-plant hydraulic conductivity . The K_w required to maintain positive annual net primary production.	Modeled

Lin 440: You still have not defined K_w ?

We thank the reviewer for pointing out this omission. We hope the explanation of K_w given in the model description serves to address this comment (Methods: Competition model; lines 351-356).

- “Competition model” (Methods):** We modified the Trugman et al. model to include a tree-liana pair and to improve the realism of the relationship between climate and plant water flow. In contrast to the use of this model for computing A_{net} as in Trugman et al., we use the model to define K_{req} , the required **maximum** whole-plant hydraulic conductivity, by iteratively finding the minimum K_{max} (Equation 4) to yield a positive A_{net} on an annual timestep (Methods: Simulations). **To emphasize the independence of the maximum hydraulic conductivity (K_{max}) from plant branch-level measurements and differentiate this term in the model from K_s (observed branch hydraulic conductivity), we designate this term maximum whole-plant hydraulic conductivity (K_w) hereafter. The hydraulic conductivity variables we consider in this manuscript (K_s , K_w , and K_{req}) are defined in Supplementary Table 7.**

L338: Liana DBH

We thank the reviewer for making this clarifying suggestion. The word “liana” has been added to this sentence (Methods: Competition model; line 370).

- “Competition model” (Methods):** Liana DBH is then treated one of two ways.

L343-346: Why is the minimum (2 cm) and mean (2.65 cm) DBH for lianas so similar? Is the size distribution really that right-skewed? I

According to our field survey from Guanacaste, Costa Rica, canopy liana DBH is highly skewed. This is shown in Supplementary Figure 5, which depicts the frequency of tree and liana DBHs in Guanacaste, Costa Rica, a region with relatively high liana proliferation.

A copy of Supplementary Figure 5 is below.

Supplementary Figure 5: Distribution of diameters at breast height (DBHs). (A) tree DBHs from a second-growth forest plot in Guanacaste, Costa Rica from Smith-Martin et al. 2020¹⁸. (B) liana DBHs from Guanacaste, Costa Rica from Smith-Martin et al. (unpublished).

Supp Table 4: What is difference between b2Ht and dbh2h1? Should one of these be biomass as in Trugman et al 2018? I am searching to understand how leaf area is treated.

The parameters b2Ht and dbh2h1 are both related to the relationship between DBH and height. Height is defined for trees as $\text{height} = \text{dbh2h1} * \text{DBH}^{\text{b2Ht}}$. This has been added to Supplementary Table 4 in the “Definition” column.

Leaf area is an input to the model. The total leaf area is defined for both the liana and the tree combined (typically, 200 m²). Then, the fraction of the total leaf area that is apportioned to each the liana and the tree is defined in accordance with the competition scenario (invasion scenario = 10% liana leaf area, 90% tree leaf area; established scenario = 40% liana leaf area, 60% tree leaf area). This is described in the Methods section “Competition model” (lines 360-363) and subsection “Model parameterization” (lines 430-433).

Figures

Figure 2: The way this figure is presented at least for me requires a fair bit of time to digest. I think two main sources of confusion are: 1) the x-axis is strictly reserved for variation in lianas but not trees, but the axis title does not indicate this and 2) one has to read the figure caption to

understand that solid lines are lianas and dashed lines / Xs are trees. It may improve clarity to restructure the presentation in such a way as to make it seem the main messages of this figure are: 1) Liana K_{req} increases with increasing leafiness (lower HV) and with a drier climate, and 2) Liana K_{req} exceeds tree K_{req} by many factors over the majority of simulations, given the observed size distributions. Given that invasion scenario is also tantamount to variation in liana HV, how critical is it to have the different invasion scenarios represented? It may be worth re-conceiving a figure that best conveys these messages while eliminating redundancy.

We thank the reviewer for highlighting their difficulty interpreting this figure. We have reformatted the figure to improve clarity. First, panels A and B now state that the x-axis is *liana* DBH and *liana* Huber value. Second, panels A and B no longer include the reference tree scenarios; instead, only liana $\log(K_{req})$ is plotted in these panels. Third, in order to still compare liana and tree K_{req} under different climate and allometry scenarios, we have added a panel C, which plots the ratio of liana K_{req} to tree K_{req} as a function of liana Huber value. In panel C, tree K_{req} is still computed from the reference tree scenario which is independent of the x-axis (i.e., one Huber value per line, not variable Huber value). Finally, the figure now includes only one leaf area scenario for each growth form (the tree occupies 60% of the total leaf area, the liana occupies 40% of the total leaf area), which is consistent with the format of our Figures 3 & 4 of the main text. A copy of this figure is available below.

To accommodate the new format of this figure, we have updated the main text of the manuscript. First, we modified how we described our model simulations in the main text to exclude information about the competition scenarios (Main text: Hydraulic traits influence viability; line 113). This is because we no longer discuss the “invasion” competition scenario in the main text; this only appears in the methods and the supplementary discussion. Second, we emphasize *liana* allometry in our discussion of Figure 2, rather than including a discussion of both trees and lianas. We then separately discuss the relationship between tree and liana K_{req} shown in panel C (Main text: Hydraulic traits influence viability; lines 134-136). This required us to re-order the paragraphs in the section “Hydraulic traits influence viability,” in addition to making adjustments to the text. To make the changes as clear to the reviewer as possible, we have included the entire section in Appendix 2.

We made some additional small edits to the main text in order to remove language pertaining to the different “competition scenarios.” This included removing phrases about competition scenarios from the “Hydraulic traits influence viability” section of the main text (e.g., line 113). Additionally, we added an explanation of where to find the “invasion” scenario to the Methods (Methods: Model parameterization; line 428). Finally, we edited the text in the methods to reflect removing the “invasion” leaf area scenario from Figure 2 (Methods: Simulations; lines 473-477).

- **“Model parameterization” (Methods):** For the scenario of a liana invading a tree canopy (“invasion scenario” considered in the Supplementary Discussion), we assumed a liana DBH of 2 cm⁶⁶.

Below is a copy of Figure 2.

Figure 2: **Allometry and climate affect hydraulic conductivity.** Required hydraulic conductivity (K_{req}) as a function of diameter at breast height (DBH, **A**) and Huber value (sapwood area [cm²] per unit leaf area [m²], **B**, **C**), and hydroclimate (tropical moist forest or tropical dry forest). Total leaf area = 200 m², 60% tree leaf area, 40% liana leaf area. In all three panels, colors represent the different hydroclimate scenarios (tropical dry or tropical moist forest). **(A)** Liana $\log(K_{req})$ as a function of liana DBH. **(B)** Liana $\log(K_{req})$ as a function of liana Huber value. **(C)** The ratio of liana K_{req} to tree K_{req} as a function of liana Huber value. Tree K_{req} was computed at a reference scenario where tree DBH = 18.2 cm.

Figure 3B: Need units in the legend for K_{req} , and shouldn't it be $\log(K_{req})$?

We thank the reviewer for pointing out this omission. The units have been added to the legend for Figure 3. The units are not $\log(K_{req})$. The logarithm was only used in Figure 2 to better show the shape of the curve across a variety of DBH/Huber value scenarios.

- **“Figure 3” (Main text):** Figure 3: **Required hydraulic conductivity (K_{req}) as a function of vapor pressure deficit (VPD) and soil water potential (Ψ).** (A) Conceptual diagram showing how hydroclimate changes over the 2-dimensional space depicted in the other two panels. (B and C) K_{req} (mol m⁻¹ s⁻¹ MPa⁻¹) over 10,000 combinations of VPD and Ψ indices.

I could not find the new Suppl Fig S15 (sensitivity to future adjustment in P50) referred to in the response to reviewer #1.

The figure the reviewer is looking for is Supplementary Figure 16.

References

- Xu, X., Medvigy, D., Powers, J. S., Becknell, J. M. & Guan, K. 2016. Diversity in plant hydraulic traits explains seasonal and inter-annual variations and vegetation dynamics in seasonally dry tropical forests. *New Phytol.* **212**, 80-95. <https://doi.org/10.1111/nph.14009>.
- Pérez-Harguindeguy, N. *et al.* 2013. New handbook for standardised measurement of plant functional traits worldwide. *Aust. J. Bot.* **61**, 167-234. <http://dx.doi.org/10.1071/BT12225>.

Appendix 1: Model description

Below is the section of the methods titled “Competition model.” Additions to this section are in green text, as above.

We modified the singletree model originally developed by Trugman et al.³⁶ to represent a single lianatree pairing. The purpose of the original model developed by Trugman et al. is to calculate annual net primary production (A_{net}) of a single temperate tree under defined climatic conditions and morphological and physiological parameters, with A_{net} becoming the input to a subsequent model describing tree drought recovery. Briefly, the model couples water transport using the Shinozaki pipe model³⁸ and the Ball-Berry model of stomatal conductance³⁹ and **maximum** whole-plant photosynthesis using the Farquhar photosynthesis model³⁷. The amount of water moving through the plant depends on soil water availability (soil water potential, Ψ); the hydraulic path length and xylem area of fine roots, stem, and petioles; and the water demand imposed on the tree by the atmosphere (vapor pressure deficit, VPD). Mathematically, this can be written with the following set of equations. First, the flow, F (**mmol s⁻¹**), throughout a plant element is computed by integrating the hydraulic conductivity per unit of xylem area (K) from one end of the pipe continuum with water potential ψ_1 (**MPa**) to the other with water potential ψ_2 , which can be expressed by the differences in the Kirchhoff transforms as

$$F = \frac{a}{L} \int_{\psi_1}^{\psi_2} K(\psi) d\psi = \frac{a}{L} (\phi_2 - \phi_1) \quad (1)$$

where a (**m²**) is the xylem area of the element and L (**m**) the pipe length. The element conductivity (K , **mmol m⁻¹ s⁻¹ MPa⁻¹**) decreases as stem water potential falls as a result of embolism. A logistic function is used to represent the loss of conductivity as water potential becomes more negative, and thus ϕ (**mmol m⁻¹ s⁻¹**) is **a function of** the maximum hydraulic conductivity, K_{max} .

If we neglect changes in water storage, F is constant throughout the hydraulic continuum. Then, water flow from the roots to the stem is modeled as

$$F = \frac{a_{root}}{L_{root}} (\phi_{soil} - \phi_{root}) \quad (2)$$

where a_{root} is the surface area of the tree roots, L_{root} is the path length from the soil to the base of the stem, and ϕ_{soil} and ϕ_{root} are the integral of the conductivity for the soil and roots, respectively, calculated from the Kirchhoff transform. Flow from the roots to the stem is modeled as

$$\frac{a_{root}}{L_{root}} (\phi_{soil} - \phi_{root}) = \frac{a_{stem}}{L_{stem}} (\phi_{root} - \phi_{stem}) \quad (3)$$

where a_{stem} is the cross-sectional area of the xylem, L_{stem} is the tree height, and ϕ_{stem} is the integral of the conductivity for the stem. Flow from the stem to leaves is modeled as

$$\frac{a_{stem}}{L_{stem}} (\phi_{root} - \phi_{stem}) = \frac{a_{petiole}}{L_{petiole}} (\phi_{stem} - \int_0^L (\phi_{leaf}(l) \frac{dl_a}{L_a}) \quad (4)$$

where $a_{petiole}$ is the cross-sectional xylem area within a given petiole summed over the tree, $L_{petiole}$ is the length of the petiole, ϕ_{leaf} is the integral of the conductivity for the petiole, L_a ($m^2 m^{-2}$) is the leaf area index, l_a (m^2) is the index of a given leaf layer, and dl_a/L_a represents the xylem area per unit leaf. Assuming there is only one leaf layer and all photosynthesis is carbon limited only, this equation simplifies to

$$\frac{a_{stem}}{L_{stem}} (\phi_{root} - \phi_{stem}) = \frac{a_{petiole}}{L_{petiole}} (\phi_{stem} - \phi_{leaf}) \quad (5)$$

Flow from the leaf to the atmosphere is modeled as

$$\frac{a_{petiole}}{L_{petiole}} (\phi_{stem} - \phi_{leaf}) = a_{leaf} g_s D \quad (6)$$

where a_{leaf} is leaf area, g_s ($mmol m^{-2} s^{-1}$) is stomatal conductance, and D (Pa) is VPD. Stomatal conductance, g_s , is modeled following ref. 63 as

$$g_s = A_n \frac{c_1}{(C_a - \Gamma)(1 + \frac{D}{D_0})} \beta(\psi_{leaf}) \quad (7)$$

In this equation, C_a (ppm) is the atmospheric CO_2 concentration; c_1 (Pa), D_0 (Pa), and Γ (ppm) are empirical constants from the Leuning model⁶³; A_n (kg C month⁻¹) is net photosynthesis; and ψ_{leaf} is leaf water potential. The function $\beta(\psi_{leaf})$ serves to down-regulate photosynthesis under water stressed conditions and is determined by the carbon cost of sustaining negative water potential and loss of conductivity in the xylem. For simplicity, we assumed that $\beta(\psi_{leaf})$ varies linearly with the Kirchoff transform as

$$\beta(\psi_{leaf}) = \frac{\phi_{leaf}}{\phi_{max}} \quad (8)$$

where ϕ_{max} is the integral of maximum hydraulic conductivity of the xylem. B varies between 1 (leaf at full hydration) and 0 (leaf under full water stress). The denominator ϕ_{max} is defined in terms of the maximum hydraulic conductivity (K_{max}) as follows:

$$\phi_{max} = \frac{K_{max} * \log(\exp(-b1*b2) + 1)}{b1} \quad (9)$$

where K_{max} is equivalent to the maximum whole-plant hydraulic conductivity (K_w) and $b1$ (% MPa^{-1}) and $b2$ (MPa) are the slope of the percent loss of the conductivity curve and the pressure at which 50% of xylem function is lost, respectively. Here, β broadly conforms to the solution to the Leuning model, but with a more mechanistic representation of soil moisture stress through soil water potential's effect on leaf water potential.

The method of solution is the same as in Trugman et al.³⁶. In this way, computation of A_{net} is related to three climatic variables (ψ , VPD, and CO_2 concentration), dimensions of the water conducting tissue of the tree, and tree physiological parameters.

Appendix 2: “Hydraulic traits influence viability”

Below is the section of the main text titled “Hydraulic traits influence viability.” Additions to this section are in green text, as above.

To evaluate how K_s influences liana-tree competition, we parameterized a model³⁶ coupling Farquhar photosynthesis³⁷, Shinozaki water transport³⁸, and Ball-Berry stomatal conductance³⁹ to estimate annual net primary production (NPP) for a liana-tree pair sharing a single canopy (Methods: Competition Model). We restricted growth form-specific parameterization to whole-plant hydraulic conductivity, allometry, and woody turnover rate (Methods: Parameterization). We conducted an extensive sensitivity analysis to ensure that parameters for which tropical data are sparse would not strongly influence simulation outcomes (Methods: Sensitivity analysis, Supplementary Discussion: Sensitivity analysis).

We forced the model with average monthly soil water potential (Ψ) and average hourly vapor pressure deficit (VPD) characteristic of Central American sites representing contrasting hydroclimates: Barro Colorado Island, Panama (“tropical moist forest”) and Horizontes, Costa Rica (“tropical dry forest”) (Methods: Climate Data). All other parameters remained constant between runs. For each scenario, we identified the minimum **maximum** whole-plant hydraulic conductivity required (K_{req} , Supplementary Figure 1) to maintain annual NPP > 0 (Methods: Simulations).

We find that liana K_{req} is greater at lower diameters when total leaf area is constant and at lower Huber value (Figure 2a-b) because the xylem supplies relatively more leaves with water under these conditions. This pattern indicates that the unique liana allometry influences its physiology, consistent with the structure of our model (Methods: Competition Model) and the theoretical model derived by Mencuccini et al.⁴²; specifically, a lower Huber value, characteristic of lianas in comparison to trees^{3,5}, demands higher K_{req} to supply leaves with a consistent source of water, thus maintaining positive NPP.

Second, liana K_{req} is greater than tree K_{req} except at large liana Huber values, at which point the liana's sapwood-to-leaf area allometry approaches the tree's allometry. This result is consistent with our meta-analysis (Figure 1) and with previous site-specific comparisons of liana and tree K_s ^{35,40,41}. The consistency of our model predictions, based on physical properties of xylem function, with observation suggests that the observed difference in K_s in the literature represents a fundamental source of variation between woody growth forms in tropical forest biomes. This variation must be represented in the development of a liana growth form in vegetation models.

Finally, we find that climatic water stress influences K_{req} (Figure 2). The approximately twofold increase in liana K_{req} in the dry forest compared with the moist forest (Figure 2) suggests that liana K_{req} is sensitive to changes in hydroclimate. Moreover, the ratio of liana K_{req} to tree K_{req} does not change as a function of hydroclimate (Figure 2c), indicating that tree K_{req} is similarly sensitive to hydroclimate. Therefore, we next investigated the magnitude of change in liana and tree K_{req} over a hydroclimate gradient representative of tropical dry and tropical moist Neotropical forests. Furthermore, K_{req} could be sensitive to low water supply (low Ψ), high water demand (high VPD), or a combination of the two hydroclimate variables. Because Ψ and VPD naturally covary, we used our model to separate the sensitivity of liana and tree K_{req} to Ψ and VPD.

Reviewer comments, third round

Reviewer #2 (Remarks to the Author):

Re-Re-review of "Climate and hydraulic traits interact to set thresholds for liana viability"

The model description detail is now sufficient (with some minor corrections, see below) and allows the readers to fully understand what is going on. I'm satisfied with the authors responses and now fully understand (via the equations) what 'whole-plant hydraulic conductivity' is (I suggest some additional clarity, see below). The figures are also greatly improved. I had a few final minor corrections below.

L311: Give here the logistic equation for $K(Y)$ which you stated in your response to reviewers and define your b_1 and b_2 terms.

L312-313: I still think this can be made more clear in light of discussion on this point; I suggest amending and adding: "...is a function of the maximum *whole-plant* hydraulic conductivity, K_{max} (units). K_{max} is analogous to stem-specific hydraulic conductivity (K_s , same units) measured in branches, but here represents a whole-plant value because the assumptions of our pipe model (constant xylem area a with branching and path length L that is representative of the whole path from roots to leaves) allow us to approximate a tree or liana with an effective element conductivity for this entire path. As such, K_{max} is distinct from K_s ."

Supp Table 4: Now, if K_{max} is not something that exists in the literature (only K_s is measured), then how can you say in this table that you used meta-analysis to parameterize K_{max} ?

Thanks for clarifying distinction between K_s , K_{max} , K_w , and K_{req} . For the two terms which are identical (K_w and K_{max}), why not simply use one term? Personally, I think a convention like $K_{s,max}$, $K_{w,max}$, and $K_{w,max(req)}$ would make the distinction between your meta-analysis (first term) and model (second two terms) clearer, while also being explicit that all terms indicate 'maximum' (i.e., at $Y = 0$).

L316: In light of the pipe model, I think A_{root} here is not the surface area of roots but rather the sum of xylem cross-sectional area across all root branching levels.

L315-318: This series of equations (2), (3) I find misleading/confusing in light of the pipe model. For equation 2, there is an inconsistency – the water (flux) potentials are soil and root, so the path length should not be from root tip to stem base, which is on the order of meters, but rather the path length associated the distance water travels from bulk soil to root xylem (soil-root epidermis-cortex-xylem), which is on the order of millimeters. You are not representing this level of detail, so I think what's actually being done from soil to stem base in the model is represented by a single equation. Root water potential in your model is intermediate to soil and stem water potential but does not need to be explicitly represented:

$A_{root}/L_{root} * (\phi_{i,soil} - \phi_{i,stem})$. (revised Eqn 2)
Where A_{root} and L_{root} are as you have defined them.

And then it follows that, the stem to leaf equation $A_{stem}/L_{stem} * (\phi_{i,stem} - \phi_{i,leaf})$ can be equated with the previous revised Eqn 2 as per continuity. Correct the ϕ values in your Eqns 4 and 5 accordingly.

Also $\phi_{i,X}$ is not an 'integral of conductivity for X' – rather more accurately the Kirchoff transform is always stated in terms of a difference; e.g. $(\phi_{i,Y} - \phi_{i,X})$ is 'the integral of conductivity from X to Y' (check that the integral is going in the right direction for your various soil, stem, leaf terms).

In the following response to reviewers, the reviewers' comments are *italicized*. We add **our responses and explanations** below each comment in **blue**. We have copied and pasted **short new sections from the manuscript** in **green** offset by quotations. Changes in the **manuscript** are indicated **by purple text**. Changes to the Supplementary Discussion, Figures, and Tables are included in the response and their location is given by section title but not by line numbers or purple text because this formatting has been removed for final submission of this manuscript.

REVIEWER COMMENTS

Reviewer #2 (Remarks to the Author):

Re-Re-review of "Climate and hydraulic traits interact to set thresholds for liana viability"

The model description detail is now sufficient (with some minor corrections, see below) and allows the readers to fully understand what is going on. I'm satisfied with the authors responses and now fully understand (via the equations) what 'whole-plant hydraulic conductivity' is (I suggest some additional clarity, see below). The figures are also greatly improved. I had a few final minor corrections below.

We thank the reviewer for their constructive feedback throughout the review process and for their careful consideration of the improvements we have made to the manuscript.

L311: Give here the logistic equation for K(Y) which you stated in your response to reviewers and define your b1 and b2 terms.

We included the equation for K(Y) and the physiological definitions for b1 and b2 in the manuscript (Methods: Competition model; LINES 379-380).

- **"Competition model" (Methods):** The element conductivity (K , $\text{mmol m}^{-1} \text{s}^{-1} \text{MPa}^{-1}$) decreases as stem water potential falls as a result of embolism. A logistic function of the form

$$\frac{K_{max} * \exp(b1*(\psi_{soil} - b2))}{\exp(b1*(\psi_{soil} - b2)) + 1} \quad (2)$$

where $b1$ is the slope of the percent loss of conductivity (PLC) curve and $b2$ is P_{50} , is used to represent the loss of conductivity as water potential becomes more negative, and thus ϕ ($\text{mmol m}^{-1} \text{s}^{-1}$) is a function of the maximum whole-plant hydraulic conductivity, K_{max} .

*L312-313: I still think this can be made more clear in light of discussion on this point; I suggest amending and adding: "...is a function of the maximum *whole-plant* hydraulic conductivity,*

*K_{max} (units). K_{max} is analogous to stem-specific hydraulic conductivity (K_s, same units) measured in branches, but here represents a whole-plant value because the assumptions of our pipe model (constant xylem area *a* with branching and path length *L* that is representative of the whole path from roots to leaves) allow us to approximate a tree or liana with an effective element conductivity for this entire path. As such, K_{max} is distinct from K_s.”*

We thank the reviewer for their suggestion on how to better communicate the distinction between K_{max} and K_s. We incorporated a version of the text the reviewer suggested to the methods (Methods: Competition model; LINES 383-388). We chose to remove the language that K_{max} and K_s are “analogous” that the reviewer used to continue to emphasize that these variables are distinct and should not be confused.

- **“Competition model” (Methods):** The element conductivity (*K*, mmol m⁻¹ s⁻¹ MPa⁻¹) decreases as stem water potential falls as a result of embolism. A logistic function of the form

$$\frac{K_{max} * \exp(b1 * (\psi_{soil} - b2))}{\exp(b1 * (\psi_{soil} - b2)) + 1} \quad (2)$$

where *b1* is the slope of the percent loss of conductivity (PLC) curve and *b2* is P₅₀, is used to represent the loss of conductivity as water potential becomes more negative, and thus *ϕ* (mmol m⁻¹ s⁻¹) is a function of the maximum whole-plant hydraulic conductivity, K_{max} (mmol m⁻¹ s⁻¹ MPa⁻¹). The assumptions of our pipe model (i.e., constant xylem area, *a*, with branching and path length, *L*, that is representative of the whole path from roots to leaves) allows us to approximate an individual tree or liana with an effective element conductivity for the entire path. This is in contrast to stem-specific hydraulic conductivity (K_{s,max}, mmol m⁻¹ s⁻¹ MPa⁻¹), which is commonly measured in the field on terminal branches and does not account for the tapering of vessel elements in branches. Therefore, K_{max} is distinct from K_{s,max}.

Supp Table 4: Now, if K_{max} is not something that exists in the literature (only K_s is measured), then how can you say in this table that you used meta-analysis to parameterize K_{max}?

We thank the reviewer for bringing this typo to our attention. Originally, we defined the boundaries of our values of K_{max} from the literature. During the revision process, we eliminated the dependence of K_{max} on K_s or any other literature-defined variable. We revised the entry in the “Source” column of Supplementary Table 4 to read “Response variable” to indicate that we treat K_{max} as the response variable, since we are deriving K_{w,max}(req) from this parameter in all of our simulations. An updated version of Supplementary Table 4 is presented below.

Changed Model Parameters

Name	Definition	Value	Units	Source	Tree or Liana Function
ax	Functional xylem cross-sectional area	Min(tot.area, (2.41 * (dbh/2) ^{1.97} * 0.0001))	m ²	Reyes-García et al. 2012	T, L
b1	Slope of PLC curve	1.79	% MPa ⁻¹	Meta-analysis	T, L
b2	P ₅₀	-1.91	MPa	Meta-analysis	T, L
b2Ht	DBH to height allometric constant (height = dbh ^{2h1} * DBH ^{b2Ht})	0.455		Smith-Martin et al. (unpublished)	T
Ca	Atmospheric [CO ₂]	400	ppm	Low estimate for 21 st century	T, L
dbh	Diameter at breast height	Varied	cm	Smith-Martin et al. (unpublished)	T, L
dbh2h1	DBH to height allometric constant (height = dbh ^{2h1} * DBH ^{b2Ht})	3.06		Smith-Martin et al. (unpublished)	T

frac.liana.al	Fraction of the total leaf area occupied by the liana	Invading liana: 0.1; Mature liana: 0.4		Competition scenarios	L
frac.tree.al	Fraction of the total leaf area occupied by the tree	Invading liana: 0.9; Mature liana: 0.6		Competition scenarios	T
Kmax	Maximum whole-plant hydraulic conductivity	Varied	$\text{mmol m}^{-1} \text{s}^{-1} \text{MPa}^{-1}$	Response variable	T, L
leaf.biom	Leaf biomass	$(1 / (\text{SLA}/S)) * \text{al}$	Kg		T, L
Lx	Initial stem length	18.2	M	DBH-height allometry	L
Lx_lost	Stem length lost due to turnover	$Lx * \text{stem.turn}$	M		T, L
Lx_turn	Stem length left after turnover	$Lx - Lx_lost$	M		T, L
rho	Wood density	420	kg m^{-3}	Trugman et al. 2018; Putz 1990; Putz & Milton 1982	T, L
stem.biom	Total stem biomass	$\text{tot.area} * \text{rho} * Lx / 2$	kg	Trugman et al. 2018	T, L
stem.turn	Stem turnover	10	$\% \text{ year}^{-1}$	Ichihashi & Tateno 2015 & Powers (personal observation)	L

stem.turn	Stem turnover	2	% year ⁻¹	Vilanova et al. 2018 & Lewis et al. 2004	T
tot.al	Total leaf area	200	m ²	100 m ² * 2 m ² m ⁻²	T, L
tot.area	Total stem cross sectional area	((*dbh ²) / 4) * 0.0001	cm ²	Geometric relationship	T, L

Thanks for clarifying distinction between K_s , K_{max} , K_w , and K_{req} . For the two terms which are identical (K_w and K_{max}), why not simply use one term? Personally, I think a convention like $K_{s,max}$, $K_{w,max}$, and $K_{w,max(req)}$ would make the distinction between your meta-analysis (first term) and model (second two terms) clearer, while also being explicit that all terms indicate ‘maximum’ (i.e., at $Y = 0$).

We agree with the reviewer that revising the terminology used to represent K_s , K_w , and K_{req} to $K_{s,max}$, $K_{w,max}$, and $K_{w,max(req)}$ will help the reader track the use of these terms throughout the manuscript. Accordingly, we have replaced the old terms (K_s , K_w , and K_{req}) with the new terms ($K_{s,max}$, $K_{w,max}$, $K_{w,max(req)}$) throughout the text, figures, tables, and supplement.

The terms K_w and K_{max} are distinct because K_{max} is the term originally used in the model (i.e., the one used in the original model description from Trugman et al. (2018)). Therefore, we have decided to continue to differentiate these terms in the methods when first introducing the model variables.

L316: In light of the pipe model, I think a_{root} here is not the surface area of roots but rather the sum of xylem cross-sectional area across all root branching levels.

The definition of a_{root} in our models description was changed to reflect this suggestion (Methods: Competition model; LINES 390-394).

- **“Competition model” (Methods):** Then, water flow from the roots to the stem is modeled as

$$F = \frac{a_{root}}{L_{root}} (\phi_{soil} - \phi_{root}) = \frac{a_{stem}}{L_{stem}} (\phi_{root} - \phi_{stem}) \quad (3)$$

where a_{root} and a_{stem} are the cross-sectional xylem area of the root system and the cross-sectional area of the xylem, respectively, L_{root} and L_{stem} are the path length from the soil to the base of the stem and the tree height, respectively, and $(\phi_{soil} - \phi_{root})$ and $(\phi_{root} - \phi_{stem})$ are

the integral of conductivity from the soil to the roots and from the roots to the stem, respectively, calculated from the Kirchhoff transform.

L315-318: This series of equations (2), (3) I find misleading/confusing in light of the pipe model. For equation 2, there is an inconsistency – the water (flux) potentials are soil and root, so the path length should not be from root tip to stem base, which is on the order of meters, but rather the path length associated the distance water travels from bulk soil to root xylem (soil-root epidermis-cortex-xylem), which is on the order of millimeters. You are not representing this level of detail, so I think what’s actually being done from soil to stem base in the model is represented by a single equation. Root water potential in your model is intermediate to soil and stem water potential but does not need to be explicitly represented:

A_{root}/L_{root}(phi,soil – phi,stem). (revised Eqn 2)
Where A_{root} and L_{root} are as you have defined them.*

And then it follows that, the stem to leaf equation A_{stem}/L_{stem}(phi,stem – phi,leaf) can be equated with the previous revised Eqn 2 as per continuity. Correct the phi values in your Eqns 4 and 5 accordingly.*

We believe this concern comes from our separation of equations 2 and 3 in a previous round of revision. According to the original model description in Trugman et al. (2018), we have recombined these equations to facilitate the reader’s understanding of our use of the pipe model, consistent with the model’s original description (Methods: Competition model; LINES 390-394).

- **“Competition model” (Methods):** If we neglect changes in water storage, F is constant throughout the hydraulic continuum. Then, water flow from the roots to the stem is modeled as

$$F = \frac{a_{root}}{L_{root}} (\phi_{soil} - \phi_{root}) = \frac{a_{stem}}{L_{stem}} (\phi_{root} - \phi_{stem}) \quad (3)$$

a_{root} and a_{stem} are the cross-sectional xylem area of the root system and the cross-sectional area of the xylem, respectively, L_{root} and L_{stem} are the path length from the soil to the base of the stem and the tree height, respectively, and $(\phi_{soil} - \phi_{root})$ and $(\phi_{root} - \phi_{stem})$ are the integral of conductivity from the soil to the roots and from the roots to the stem, respectively, calculated from the Kirchhoff transform. Flow from the stem to leaves is modeled as

$$\frac{a_{stem}}{L_{stem}} (\phi_{root} - \phi_{stem}) = \frac{a_{petiole}}{L_{petiole}} (\phi_{stem} - \int_0^L \phi_{leaf}(l_a) dl_a) \quad (4)$$

where $a_{petiole}$ is the cross-sectional xylem area within a given petiole summed over the tree, $L_{petiole}$ is the length of the petiole, ϕ_{leaf} is the integral of the conductivity for the petiole, L_a ($m^2 m^{-2}$) is the leaf area index, l_a (m^2) is the index of a given leaf layer, and

dl_a/L_a represents the xylem area per unit leaf. Assuming there is only one leaf layer and all photosynthesis is carbon limited only, this equation simplifies to

$$\frac{a_{stem}}{L_{stem}} (\phi_{root} - \phi_{stem}) = \frac{a_{petiole}}{L_{petiole}} (\phi_{stem} - \phi_{leaf}) \quad (5)$$

Also ϕ_{X} is not an ‘integral of conductivity for X’ – rather more accurately the Kirchoff transform is always stated in terms of a difference; e.g. $(\phi_{Y} - \phi_{X})$ is ‘the integral of conductivity from X to Y’ (check that the integral is going in the right direction for your various soil, stem, leaf terms).

We thank the reviewer for bringing this comment to our attention. We have modified the model description accordingly. In addition, we included separate descriptions of the terms

$(\phi_{stem} - \int_0^L \phi_{leaf}(l_a) \frac{dl_a}{L_a})$ (in equation 4) and $(\phi_{stem} - \phi_{leaf})$ (in equation 5) to accommodate this revision (Method: Competition model; LINES 393, 398, & 402).

- **“Competition model” (Methods):** If we neglect changes in water storage, F is constant throughout the hydraulic continuum. Then, water flow from the roots to the stem is modeled as

$$F = \frac{a_{root}}{L_{root}} (\phi_{soil} - \phi_{root}) = \frac{a_{stem}}{L_{stem}} (\phi_{root} - \phi_{stem}) \quad (3)$$

where a_{root} and a_{stem} are the cross-sectional xylem area of the root system and the cross-sectional area of the xylem, respectively, L_{root} and L_{stem} are the path length from the soil to the base of the stem and the tree height, respectively, and $(\phi_{soil} - \phi_{root})$ and $(\phi_{root} - \phi_{stem})$ are the integral of conductivity from the soil to the roots and from the roots to the stem, respectively, calculated from the Kirchoff transform. Flow from the stem to leaves is modeled as

$$\frac{a_{stem}}{L_{stem}} (\phi_{root} - \phi_{stem}) = \frac{a_{petiole}}{L_{petiole}} (\phi_{stem} - \int_0^L \phi_{leaf}(l_a) \frac{dl_a}{L_a}) \quad (4)$$

where $a_{petiole}$ is the cross-sectional xylem area within a given petiole summed over the tree, $L_{petiole}$ is the length of the petiole, $(\phi_{stem} - \int_0^L \phi_{leaf}(l_a) \frac{dl_a}{L_a})$ is the integral of the conductivity from the stem to the petiole, L_a ($m^2 \cdot m^{-2}$) is the leaf area index, l_a (m^2) is the index of a given leaf layer, and dl_a/L_a represents the xylem area per unit leaf. Assuming there is only one leaf layer and all photosynthesis is carbon limited only, this equation simplifies to

$$\frac{a_{stem}}{L_{stem}} (\phi_{root} - \phi_{stem}) = \frac{a_{petiole}}{L_{petiole}} (\phi_{stem} - \phi_{leaf}) \quad (5)$$

where $(\phi_{stem} - \phi_{leaf})$ is the integral of the conductivity from the stem to the petiole under the assumption of one leaf layer.

References

Trugman, A. T. *et al.* Tree carbon allocation explains forest drought-kill and recovery patterns. *Ecol. Lett.* 21, 1552–1560 (2018).